



# Holocene thinning of Darwin and Hatherton glaciers, Antarctica, and implications for grounding-line retreat in the Ross Sea

**Trevor R. Hillebrand**[1,a], **John O. Stone**[1], **Michelle Koutnik**[1], **Courtney King**[2], **Howard Conway**[1], **Brenda Hall**[2], **Keir Nichols**[3], **Brent Goehring**[3], and **Mette K. Gillespie**[4]

[1]Department of Earth and Space Sciences, University of Washington, Seattle, WA 98195, USA
[2]School of Earth and Climate Science and Climate Change Institute, University of Maine, Orono, ME 04469, USA
[3]Department of Earth and Environmental Sciences, Tulane University, New Orleans, LA 70118, USA
[4]Faculty of Engineering and Science, Western Norway University of Applied Sciences, Sogndal, 6856, Norway
[a]now at: Fluid Dynamics and Solid Mechanics Group, Los Alamos National Laboratory, Los Alamos, NM 87545, USA

**Correspondence:** Trevor R. Hillebrand (trhille@lanl.gov)

**Abstract.** Chronologies of glacier deposits in the Transantarctic Mountains provide important constraints on grounding-line retreat during the last deglaciation in the Ross Sea. However, between Beardmore Glacier and Ross Island – a distance of some 600 km – the existing chronologies are generally sparse and far from the modern grounding line, leaving the past dynamics of this vast region largely unconstrained. We present exposure ages of glacial deposits at three locations alongside the Darwin–Hatherton Glacier System – including within 10 km of the modern grounding line – that record several hundred meters of Late Pleistocene to Early Holocene thickening relative to present. As the ice sheet grounding line in the Ross Sea retreated, Hatherton Glacier thinned steadily from about 9 until about 3 ka. Our data are equivocal about the maximum thickness and Mid-Holocene to Early Holocene history at the mouth of Darwin Glacier, allowing for two conflicting deglaciation scenarios: (1) $\sim 500$ m of thinning from 9 to 3 ka, similar to Hatherton Glacier, or (2) $\sim 950$ m of thinning, with a rapid pulse of $\sim 600$ m thinning at around 5 ka. We test these two scenarios using a 1.5-dimensional flowband model, forced by ice thickness changes at the mouth of Darwin Glacier and evaluated by fit to the chronology of deposits at Hatherton Glacier. The constraints from Hatherton Glacier are consistent with the interpretation that the mouth of Darwin Glacier thinned steadily by $\sim 500$ m from 9 to 3 ka. Rapid pulses of thinning at the mouth of Darwin Glacier are ruled out by the data at Hatherton Glacier. This contrasts with some of the available records from the mouths of other outlet glaciers in the Transantarctic Mountains, many of which thinned by hundreds of meters over roughly a 1000-year period in the Early Holocene. The deglaciation histories of Darwin and Hatherton glaciers are best matched by a steady decrease in catchment area through the Holocene, suggesting that Byrd and/or Mulock glaciers may have captured roughly half of the catchment area of Darwin and Hatherton glaciers during the last deglaciation. An ensemble of three-dimensional ice sheet model simulations suggest that Darwin and Hatherton glaciers are strongly buttressed by convergent flow with ice from neighboring Byrd and Mulock glaciers, and by lateral drag past Minna Bluff, which could have led to a pattern of retreat distinct from other glaciers throughout the Transantarctic Mountains.

## 1 Introduction

### 1.1 The last deglaciation in the Ross Embayment

Grounded ice filled the Ross Sea of Antarctica (Fig. 1) during the last glaciation ($\sim 20$–13 ka), after which the ice sheet grounding line retreated more than 1200 km along the western side of the embayment, reaching its current position $\sim 2$ ka (Anderson et al., 2014). Conway et al. (1999) contrasted this with the persistence of grounded ice over Roosevelt Island until the Late Holocene and proposed that

grounding-line recession followed the pattern of a swinging gate, with its hinge in the eastern Ross Sea. Initial radiocarbon ages from deposits alongside the Darwin–Hatherton Glacier System (DHGS) suggested that the glacier system reached its modern configuration > 6.8 ka (Bockheim et al., 1989), which provided one of the key constraints for this model of grounding-line retreat.

More recent data show that the ∼ 800 km long section of the Transantarctic Mountains (TAM) front between Terra Nova Bay and Shackleton Glacier may have deglaciated almost simultaneously in the Early to Mid-Holocene, followed by slow recession into the Late Holocene (Spector et al., 2017). Lower Shackleton and Beardmore glaciers had reached their modern configurations by 7.4 ka, roughly contemporaneous with dramatic thinning at Mackay, Mawson, and David glaciers (Jones et al., 2015, 2020; Stutz et al., 2020) and ungrounding of ice in McMurdo Sound (Hall et al., 2004). The pattern of deglaciation was complex, as the grounding line temporarily stabilized on pinning points and retreated in deep troughs (Dowdeswell et al., 2008; Halberstadt et al., 2016; Prothro et al., 2020). After ∼ 7 ka glacier thinning in the southern Ross Embayment slowed as the grounding line neared its modern position between Scott and Reedy glaciers around 3 ka (Spector et al., 2017), likely during a readvance after the West Antarctic Ice Sheet grounding line retreated ≥ 200 km inland of its modern position (Kingslake et al., 2018). This suggests that the swinging gate model of grounding-line retreat is likely too simple.

King et al. (2020) showed that significant thinning of Hatherton Glacier continued for several thousand years after the Bockheim et al. (1989) chronology, calling the accuracy of the previous inferred age of grounding-line arrival at the DHGS (> 6.8 ka) into question (Conway et al., 1999). Darwin and Hatherton glaciers thus provide the only high-resolution constraint on last glacial maximum (LGM[1]) to present ice thickness between the Minna Bluff to McMurdo Sound region and Beardmore Glacier, a distance of ∼ 600 km. However, these data were collected alongside the tributary Hatherton Glacier 50 km from the modern grounding line of Darwin Glacier; it is not clear precisely how this chronology constrains the timing of grounding-line arrival. This leaves a first-order question unanswered: how and when did the grounding line reach its modern position in the region between Minna Bluff and Beardmore Glacier?

The region around Darwin Glacier – extending from Byrd Glacier to Minna Bluff – exerts a strong control on the dynamics of the Ross Ice Shelf. High discharge of cold ice from Byrd Glacier today dominates the regional pattern of deformation of the ice shelf by reducing extensional strain rates and increasing side shear stress that reduces ice-shelf flow

---

[1]In this paper, the term LGM will apply to the timing of maximum ice extent and thickness at the location in question, and should not be taken to mean the global or Antarctic LGM unless specifically stated.

(LeDoux et al., 2017; Hughes et al., 2017). Darwin, Mulock, and Skelton glaciers add to this effect by exerting pressure on Byrd outflow in opposition to West Antarctic ice. While the DHGS is extremely small compared to Byrd and Mulock, it is the only one of the three to have well-documented and extensive glacial deposits, although a comprehensive record of its fluctuations has not yet been achieved. The history of ice dynamics in the Byrd–Darwin–Mulock region may be key to understanding the dynamics of grounding-line retreat in the Ross Embayment during the last deglaciation and possibly to the restabilization of the grounding line of the West Antarctic Ice Sheet (Kingslake et al., 2018).

## 1.2 Physiographic setting of Darwin and Hatherton glaciers

Darwin Glacier and its major tributary Hatherton Glacier are outlets of the East Antarctic Ice Sheet (EAIS) that flow through the TAM into the modern Ross Ice Shelf. In contrast to the neighboring fast-flowing and much larger Byrd and Mulock glaciers, ice-flow velocities for the DHGS do not exceed $110 \, \mathrm{m \, yr^{-1}}$, and everywhere the velocity of Hatherton Glacier is $< 12 \, \mathrm{m \, yr^{-1}}$ (Rignot et al., 2011; Gillespie et al., 2017). The catchment for the DHGS is small ($8060 \, \mathrm{km^2}$) due to both high bedrock topography preventing flow into their canyons and the proximity of the much larger Byrd and Mulock glaciers, whose catchments effectively cut off the DHGS catchment (Swithinbank et al., 1988; Gillespie et al., 2017). While Byrd and Mulock glaciers currently contribute about 22 and $5 \, \mathrm{Gt \, yr^{-1}}$ to the Ross Ice Shelf, respectively (Stearns, 2011), Darwin Glacier contributes just $0.24 \pm 0.05 \, \mathrm{Gt \, yr^{-1}}$ (Gillespie et al., 2017). The DHGS is among the smallest of the TAM outlet glaciers where geochronologic constraints are available but is comparable in size to Skelton, Mackay, Tucker, and Aviator glaciers, with considerably larger visible ablation areas (Bindschadler et al., 2008). The modern grounding line of Darwin Glacier sits on a forward sloping bed ∼ 925 m below sea level (Gillespie et al., 2017).

The surface mass balance (SMB) of the DHGS is spatially complex, with persistent blue-ice areas and seasonal surface melt (Brown and Scambos, 2004; Gillespie et al., 2017). Net ablation is evident alongside the glaciers, which are bordered by the largest area of exposed rock in the TAM outside of the Dry Valleys–McMurdo Sound region, exceeding $800 \, \mathrm{km^2}$. The boundary layer meteorology is dominated by strong, dry, and cold downslope winds between April and September, with mean monthly air temperatures around $-25 \, ^\circ\mathrm{C}$ (Noonan et al., 2015). In December and January, relatively humid winds dominate, and mean monthly temperatures rise to $-4 \, ^\circ\mathrm{C}$. Most ablation likely occurs during the winter months, due to removal of surface snow by strong downslope winds, although surface meltwater is visible in the summer (Kingslake et al., 2017). Seasonal ablation and localized melting have produced numerous lakes and ponds that are found at the edges of Darwin and Hatherton glaciers,

The Cryosphere, 15, 1–26, 2021

https://doi.org/10.5194/tc-15-1-2021

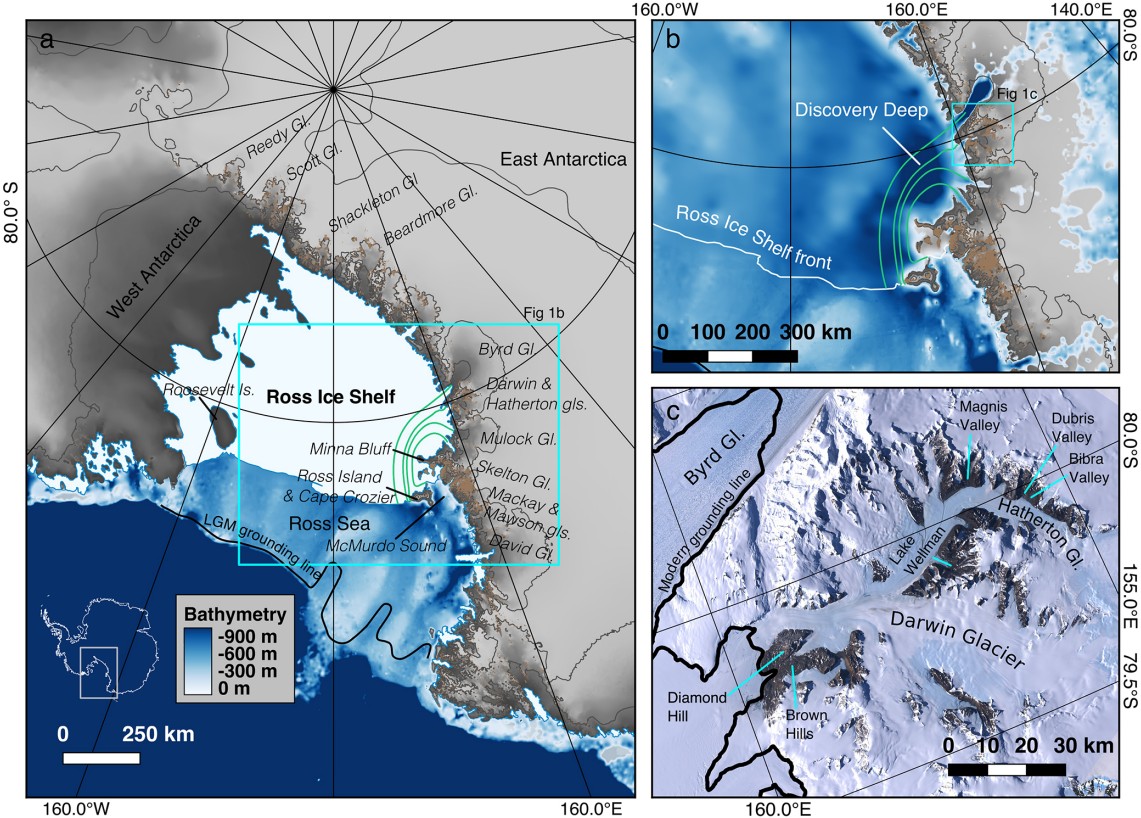

**Figure 1. (a)** The Ross Embayment of Antarctica, with regional and study locations noted. The modern ice sheet surface is shown in gray, the modern ice shelf is shown in white, and exposed rock is shown in brown. Bathymetry beyond the Ross Ice Shelf front is shown in blues (Fretwell et al., 2013). LGM grounding line from Halberstadt et al. (2016). Flow lines through the ice shelf from Byrd, Darwin, Mulock, and Skelton glaciers are shown in green and calculated using the software of Greene et al. (2017). **(b)** Inset of area outlined with cyan box in panel A, with bathymetry shown in the same color map as **(a)**. The Ross Ice Shelf has been removed to show bathymetry. The combined Byrd–Darwin–Mulock–Skelton flow path covers Discovery Deep, which is the deepest part of the seafloor in the Ross Embayment. **(c)** Landsat imagery of Darwin and Hatherton glaciers, with labels for locations mentioned in the text (Bindschadler et al., 2008). The modern grounding line is shown by the thick black line. Note the prominent blue-ice fields on the glacier surfaces.

including large frozen lakes at the glacier margins, seasonal supraglacial ponds at low elevations, and isolated ponds around Diamond Hill. These various environments host modern blue-green algae (cyanobacterial mats) (Webster-Brown et al., 2010), which are the living equivalents of the freeze-dried algae material preserved within glacial deposits and described by Bockheim et al. (1989) and King et al. (2020).

Farther downstream within the Ross Ice Shelf, ice from the DHGS converges with ice from Byrd and Mulock glaciers (Fig. 1). The convergence occurs over Discovery Deep, the deepest part of the seafloor below the Ross Ice Shelf (up to ∼ 1300 m below sea level). Thomas et al. (1984) inferred positive vertical strain rates along this flow path south of Minna Bluff, which they attributed to the convergence of flow from Byrd, Mulock, Darwin, and Skelton glaciers, combined with the influence of Minna Bluff. Minna Bluff and Cape Crozier provide stabilizing back pressure to the Ross Ice Shelf (Whillans and Merry, 2001; Fürst et al., 2016; Reese et al., 2018), and the presence of rifts at the tip of

Minna Bluff are evidence of stress concentration on the upstream side (Whillans and Merry, 2001).

### 1.3 Late Pleistocene–Holocene growth and retreat of the Darwin–Hatherton Glacier System

During the last glaciation, other outlet glaciers throughout the TAM thickened initially at their mouths in response to thickening grounded ice in the Ross Sea, and later upstream due to increased accumulation after the Antarctic LGM (Todd et al., 2010). Lateral moraines formed during this period have been used to interpret ice thicknesses of both the grounded ice sheet in the Ross Sea and the EAIS outlet glaciers (Bockheim et al., 1989; Bromley et al., 2010, 2012; Denton et al., 1989; Hall et al., 2015; Jones et al., 2015; Joy et al., 2014; King et al., 2020; Spector et al., 2017; Todd et al., 2010).

Bockheim et al. (1989) mapped lateral moraines and drift sheets in ice-free valleys alongside Hatherton Glacier and

dated these deposits based on weathering, soil characteristics, and radiocarbon ages of freeze-dried algae. These algae grew in glacier-dammed ponds and died when the ponds disappeared as the glacier retreated downslope; the radiocarbon ages of the algae provide a proxy for the glacier margin position through time (Bockheim et al., 1989; King et al., 2020). Bockheim et al. (1989) distinguished five drift units: Hatherton, Britannia I, Britannia II, Danum, and Isca drifts, in order from youngest to oldest. All subsequent work has adopted these names and confirmed the original mapping. Although we mapped and sampled some material from the three oldest deposits, this paper focuses on the Britannia I and Hatherton drifts.

The chronology and correlation of these deposits along the Hatherton-to-Darwin glacier flow line remains uncertain, and the Britannia II, Britannia I and Hatherton drifts have each been identified with the last maximum glaciation in this sector of the TAM (Bockheim et al., 1989; Storey et al., 2010; Joy et al., 2014; King et al., 2020). In addition to determining which deposits represent the last local maximum, projection of the associated ice surface downstream from sites on Hatherton Glacier to the surface of the grounded ice in the Ross Sea has been complicated by the absence of comparable deposits on Diamond Hill at the Darwin Glacier mouth. The LGM ice surface elevation at the outlet of Darwin Glacier is key to understanding the dynamics of the last deglaciation between Minna Bluff and Beardmore Glacier and is likely indicative of the behavior of the much larger Byrd and Mulock glaciers, which dominate ice flow into this region.

Although Bockheim et al. (1989) tentatively attributed Britannia II drift to the LGM and Britannia I deposits to a Holocene stillstand or readvance, Joy et al. (2014) have since dated Britannia II drift to the penultimate glacial maximum. Younger and fresher Britannia I drift occurs up to ∼ 350 m above the modern margin of Hatherton Glacier in Dubris and Bibra valleys, in Magnis Valley, and at Lake Wellman (Bockheim et al., 1989; Storey et al., 2010; Joy et al., 2014; King et al., 2020). At these sites, the deposit is a boulder- and cobble-rich ablation till, with clasts showing varying degrees of weathering. Large but discontinuous ice-cored moraines mark the limit of the deposit at Lake Wellman and in Magnis Valley and separate it from the older Britannia II drift. At Dubris and Bibra valleys the LGM limit varies from a thin scatter of fresh clasts to a low ridge of cobbles and boulders lacking an ice core. Clasts are distinctly less weathered than those in the underlying Britannia II drift. Similar recessional drift extends from the Britannia I limit towards the present glacier margin, including low moraine ridges and numerous relict lake-ice boulder pavements (King et al., 2020).

The Hatherton drift is a minimally weathered deposit, commonly underlain by relict ice and found within 50–100 m elevation of the modern margin of Hatherton Glacier. While Bockheim et al. (1989), Storey et al. (2010), Joy et al. (2014), and King et al. (2020) agree on the general characteristics of Hatherton drift, their interpretations of its significance differ. Bockheim et al. (1989) posited that Hatherton drift represents the last episode of thinning from the highstand represented by the Britannia deposits. In contrast, Storey et al. (2010) interpreted Hatherton drift to represent the last local advance of Hatherton Glacier, which they dated to 15–19 ka. In making this interpretation, they discarded several younger ages of < 3 ka, attributing these to recent exhumation from the deposit. King et al. (2020) interpreted the Hatherton drift as the youngest recessional deposit along Hatherton Glacier. Differences in morphology and the common presence of an ice core and intermittent moraine along its margin suggest that the deposit marks a brief stillstand during glacial retreat, though there is no indication in the chronologies discussed below of a prolonged pause or readvance hundreds to thousands of years in duration.

Given the absence of recognizable Britannia deposits at Diamond Hill, Bockheim et al. (1989) extrapolated their elevations from sites up-glacier to infer a thickening of 1100 m relative to present at the confluence of Darwin Glacier with the ice sheet in the Ross Sea during the last glaciation. Subsequent numerical modeling experiments (Anderson et al., 2004) yielded a more modest 800 m of thickening relative to present. The inference by Storey et al. (2010) that Hatherton rather than Britannia I drift marks the LGM highstand of Hatherton Glacier led these authors to propose no more than ∼ 50 m of LGM thickening. Joy et al. (2014) showed that the Britannia I drift in fact represents the LGM highstand, with several hundred meters of LGM thickening relative to present. They showed that Britannia II drift is a deposit of the penultimate glaciation and obtained Holocene ages from recessional positions in Britannia I drift. Radiocarbon ages of algae in Lake Wellman show that Hatherton Glacier advanced to the Britannia I limit at 9.5 ka, suggesting exposure ages in the region were overestimates due inheritance from prior exposures (King et al., 2020).

Our goals in this paper are to (i) extend and tie together the chronology of Britannia I and Hatherton drifts with new exposure ages from Dubris, Bibra, and Magnis valleys on upper Hatherton Glacier (Fig. 1); (ii) determine the LGM thickness and recessional history of the ice sheet at the mouth of Darwin Glacier, using a combination of surface exposure ages and numerical modeling of Darwin and Hatherton glaciers; and (iii) explore the sensitivity of ice thickness at the mouth of Darwin glacier to grounding-line retreat using an ensemble of ice sheet model simulations. Geochronologic constraints are presented in Sect. 2; glacier flowband modeling is presented in Sect. 3; the ice sheet model ensemble is presented in Sect. 4.

## 2 Records of glacier fluctuations

### 2.1 Geochronological methods

We mapped the limit of the Britannia I deposit to determine LGM ice thickness at Hatherton Glacier following the methodology of King et al. (2020). The Britannia I deposit often terminates in well-defined limits (mapped in blue in Figs. 2, 3, and 4), outboard of which the Britannia II deposit is clearly distinguished by its markedly more weathered appearance. We sampled glacially transported cobbles and small boulders (primarily sandstone and occasional granite erratics, resting on dolerite bedrock, gneissic granite bedrock at Diamond Hill, or older dolerite-dominated glacial deposits) for $^{10}$Be, $^{26}$Al, and $^{14}$C surface exposure dating from the limit of Britannia I deposition, as well as recessional transects back towards the modern glaciers at Diamond Hill (< 10 km from modern grounding line), Magnis Valley (∼ 70 km from modern grounding line), and Dubris and Bibra valleys (∼ 85 km from modern grounding line). We selected the least weathered available samples, targeting rocks with differentially weathered top and bottom surfaces, interpreted as indicating that a sample had remained stable since deposition. Collection of isolated erratics rather than from moraines avoids samples which may have been disturbed, overturned, or temporarily covered during sublimation and collapse of ice in the moraine core (Storey et al., 2010; Joy et al., 2014; King et al., 2020). This process can take thousands of years in Antarctica, where some glacial maximum moraines remain ice-cored up to the present day (including those in Magnis Valley and at Lake Wellman). Where possible, we sampled erratics perched stably on bedrock, such as at Diamond Hill (Fig. S1 in the Supplement) and Danum Platform (Fig. S10 in the Supplement). On the floors of Dubris, Bibra, and Magnis valleys we collected samples from Britannia and Hatherton drifts. We attempted to sample areas of thin or discontinuous drift, targeting rocks resting on underlying older, consolidated drift, though the thickness of Britannia I recessional deposits on the floors of Bibra and Dubris valleys, in particular, made this difficult. A second potential complication for these samples is coverage by proglacial lakes at the ice margin during retreat. The pervasive occurrence of sub-fossil algae (desiccated fragments of lacustrine cyanobacterial mats), discontinuous shorelines, and lake-floor pavements in both valleys indicates that retreating ice was fronted by one or more shallow proglacial lakes throughout deglaciation. Although we sampled to avoid cover by lakes, some valley floor exposure ages may have been affected and therefore underestimate depositional ages. We also noted the relative height of nearby boulders and the dominant orientation of local snow tails so as not to sample rocks that may have been covered by drift snow. In these wind-swept areas adjacent to blue-ice glacier margins, it is unlikely that deep snow cover would persist for long enough to strongly skew the exposure ages. Our sampling strategy

is similar to that of King et al. (2020) and differs from the strategies of Storey et al. (2010) and Joy et al. (2014) in that we account for differential weathering and the possibility of past coverage by ponds and snow tails. These precautions help reduce but do not eliminate the risk of spuriously young or old ages (King et al., 2020). Figures S1–S11 in the Supplement show examples of our sample selection and of the marked differences between the Britannia I and II deposits.

We also measured cosmogenic $^{14}$C produced in situ in quartz along an elevation transect of granitic bedrock from Diamond Hill to constrain the LGM ice thickness in the absence of a clear depositional limit at this location. Because any rock exposed during the few tens of kiloyears prior to the LGM may contain inherited $^{14}$C, apparent $^{14}$C exposure ages provide an upper bound on the timing of exposure since the LGM. However, the short half-life of $^{14}$C makes these exposure ages less sensitive to exposure prior to the LGM than $^{26}$Al or $^{10}$Be ages. Within the range of subaerial erosion rates typical of Antarctic bedrock (< 1 m Myr$^{-1}$), $^{14}$C concentrations reach secular equilibrium in ∼ 30 kyr (Balco et al., 2016). Because of the short half-life of $^{14}$C, 1–3 kyr of ice cover during the LGM will create a detectable signal of burial, depending on production rate and measurement sensitivity, while rock that was not covered by ice in the last 30 kyr will remain at its equilibrium concentration. Our goal at Diamond Hill, where there are no deposits that indicate the upper limit of glacial maximum ice cover, was to bracket LGM ice elevation between the lowest sample that is saturated with respect to $^{14}$C (i.e., not ice-covered during the LGM) and the highest unsaturated sample (i.e., ice-covered during the LGM).

We used conventional purification methods including heavy liquid separation, surfactant separation, and dilute hydrofluoric acid (HF) etching at ∼ 60 °C (Kohl and Nishiizumi, 1992) to isolate pure quartz for exposure dating. Samples analyzed for in situ $^{14}$C were prepared and measured twice. For the first set, quartz purified at the University of Washington Cosmogenic Nuclide Laboratory (UW) was further treated at the Tulane lab with 1 : 1 concentrated HNO$_3$: de-ionized water for 30 min at room temperature prior to loading into the carbon-extraction system. Following a 500 °C preheating in vacuo, $^{14}$C was extracted using methods described by Goehring et al. (2019b). Five of the six samples in this first batch gave $^{14}$C concentrations exceeding theoretical saturation concentrations (by up to ∼ 75 %). We reanalyzed all six samples using the improved protocol of Nichols and Goehring (2019). Results reported in this paper are from this second set of analyses, while both sets of analyses are presented by Nichols and Goehring (2019) without geologic context. As a further check on the validity of the second set of measurements, we also analyzed two erratic samples from Diamond Hill and Danum Platform, which gave exposure ages in agreement with $^{10}$Be and $^{26}$Al results. Complete analytical results are given in the Supplement. We isolated Be and Al for isotopic analysis at the University of

Washington Cosmogenic Nuclide Laboratory following the standard ion-exchange chromatography method of Ditchburn and Whitehead (1994). [14]C cathodes were prepared at Tulane University using the new fully automated carbon extraction
and graphitization system (Goehring et al., 2019b).

We analyzed samples for in situ [10]Be at the Center for Accelerator Mass Spectrometry (CAMS) at Lawrence Livermore National Laboratory. As a further check, we analyzed [26]Al from the same aliquots of quartz at the Purdue Rare Iso-
10 tope Measurement Laboratory (PRIME). [14]C cathodes were analyzed at the National Ocean Sciences Accelerator Mass Spectrometry Laboratory and at the Woods Hole Oceanographic Institution. The exposure ages presented in this paper have been calculated using the latitude–altitude scaling
scheme of Stone (2000) and the production rate calibration dataset of Borchers et al. (2016). While the choice of another scaling scheme changes the individual exposure ages, there is no major impact on the overall results, and we have therefore chosen to present results based on this scheme for sim-
plicity. For the ages presented here (excluding significantly preexposed outliers, defined here as ages > 25 kyr), using the nuclide-dependent scaling scheme of Lifton et al. (2014) would decrease [10]Be ages by $7.2 \pm 0.9\%$, decrease [26]Al ages by $8.3 \pm 0.7\%$, and increase in situ [14]C ages by $0.5 \pm 1.8\%$
on average relative to the Stone (2000) scheme. Sample data and calculated ages for all samples in this study are found in the Supplement and are archived at http://ice-d.org (last access: 3 May 2021).

## 2.2 Chronology of glacial deposits

### 2.2.1 Hatherton Glacier

The Britannia I limit across Dubris and Bibra valleys represents at least 370 m of thickening relative to the present Hatherton Glacier surface and dates to 8–10 ka (Fig. 2). Slight variations in the age of the Britannia I limit between
different sites are to be expected, as changes in local meteorology could allow small-scale fluctuations of the ice margin. After 8 ka, the glacier margin began to retreat steadily towards its present position. Only one exposure age exhibits significant prior exposure to cosmic rays (13-HAT-133-BV;
79 kyr). This sample was the closest to the glacier margin and thus prevents us from better constraining the last 150 m of glacier thinning. The thinning profile on Danum Platform is steeper than in Dubris and Bibra valleys, which could be an effect of either the complex topography or lake cover in
the valleys.

The Britannia I limit in Magnis Valley predates the limit at Dubris and Bibra valleys, with ages spanning 7.8–13.9 kyr on the valley walls and 8.3–12.4 kyr on the valley floor (Fig. 3). A single preexposed sample at the drift limit dates to ∼ 32 ka.
Assuming little to no prior exposure for the other samples, these ages imply that the glacier margin was close to its limit in Magnis Valley for ∼ 6 kyr. At Lake Wellman, [14]C ages

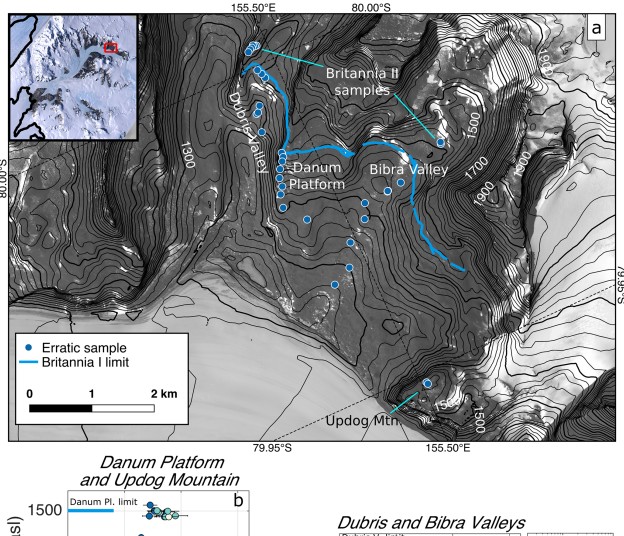

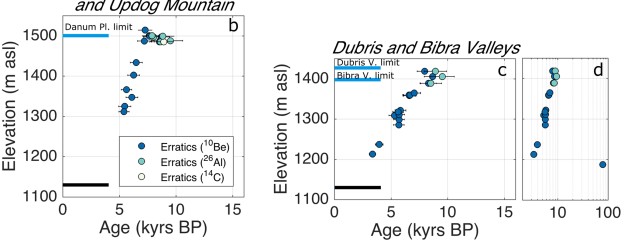

**Figure 2. (a)** Simplified map of Dubris and Bibra valleys, showing sample locations and elevations shown in 20 m contours (100 m contours are in bold). The blue curve represents the Britannia I limit; erratic samples outboard of that limit are taken from the limit of the older Britannia II deposit. Britannia II ages are not presented in this paper but are archived at http://ice-d.org (last access: 3 May 2021). Contours and satellite imagery from LINZ (2010), sourced from NZTopo Database, Crown Copyright Reserved. This product (LINZ Darwin Glacier orthophotos) incorporates data which are © Japan Aerospace Exploration Agency ("JAXA") 2008. Inset imagery from Bindschadler et al. (2008). **(b, c)** Mean surface exposure ages with total uncertainty at 1 standard deviation for glacially transported cobbles and boulders from atop bedrock platforms and valley floors, showing a stable margin until 7–8 ka, after which the glacier thinned steadily towards its modern configuration. The horizontal black line shows the elevation of the present-day glacier margin; horizontal blue lines show approximate elevations of the Britannia I limit at each location. **(d)** All surface exposure ages of the Britannia I deposit in Dubris and Bibra valleys are on a logarithmic scale to include one sample with significant inherited nuclides.

of subfossil algae suggest that the glacier margin there advanced by ∼ 500 m (50 m in altitude) from 13–11 ka (King et
al., 2020). Differences between the apparent ages and durations of maximum advance at Bibra, Dubris and Magnis valleys, and Lake Wellman may reflect small-scale differences in microclimate between the sites or may be an artifact of having dated too few samples at each site to constrain the full span of each maximum advance. We infer that Hather-
ton Glacier was close to maximum thickness from at least

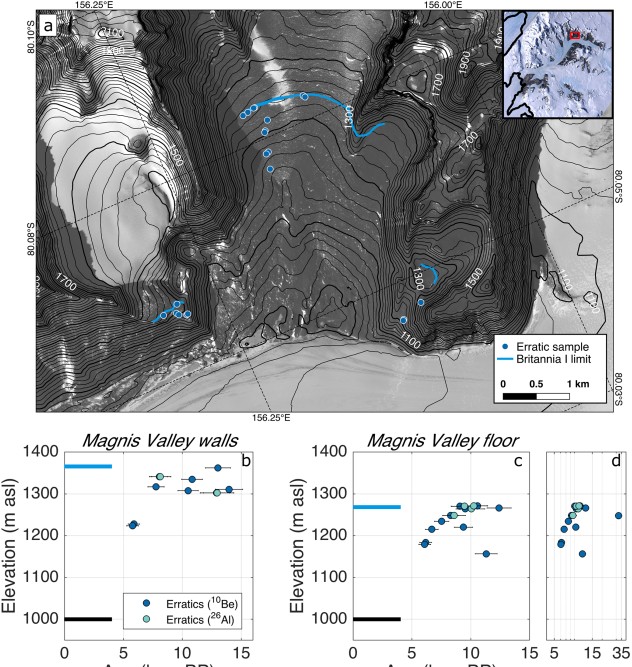

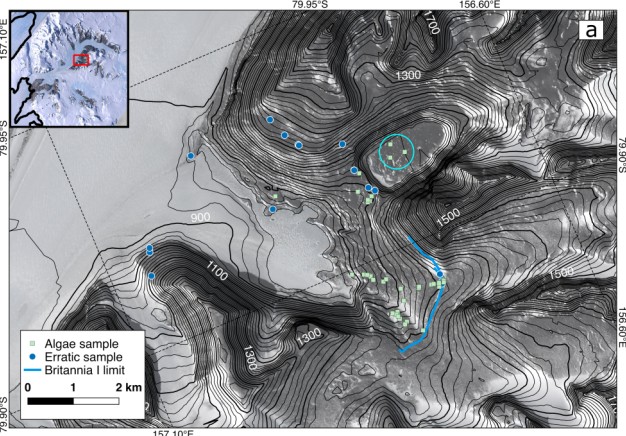

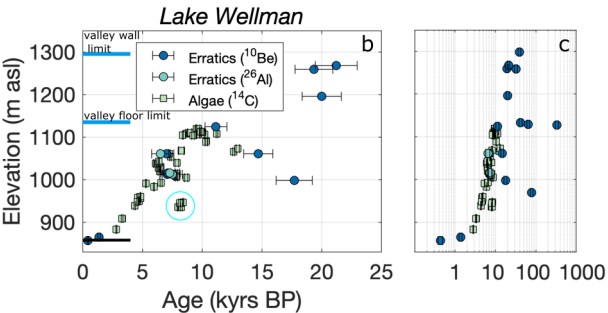

**Figure 3. (a)** Simplified map and sample locations at Magnis Valley. Blue curves indicated the mapped Britannia I (LGM) limit. Contours and satellite imagery from LINZ (2010), sourced from NZ-Topo Database, Crown Copyright Reserved. This product (LINZ Darwin Glacier orthophotos) incorporates data which are ©Japan Aerospace Exploration Agency ("JAXA") 2008. Inset imagery from Bindschadler et al. (2008). **(b, c)** Mean surface exposure ages with total uncertainty at 1 standard deviation from valley walls and valley floor. Hatherton Glacier reached its limit in Magnis Valley $\leq 14$ ka and began to retreat at 7–8 ka. The horizontal black line shows the elevation of the present-day glacier margin; horizontal blue lines show approximate elevations of the Britannia I limit at each location. **(d)** Valley floor ages on a log scale in order to show a single sample with significant prior exposure to cosmic rays.

**Figure 4. (a)** Simplified map of Lake Wellman and sample locations from King et al. (2020); we refer the reader to that paper for more detailed mapping. Hatherton Glacier would have had to override a sill at 1020 m elevation to provide meltwater to samples in the cyan circle. Contours and satellite imagery from LINZ (2010), sourced from NZTopo Database, Crown Copyright Reserved. This product (LINZ Darwin Glacier orthophotos) incorporates data which are ©Japan Aerospace Exploration Agency ("JAXA") 2008. Inset imagery from Bindschadler et al. (2008). **(b)** Mean surface exposure ages with total uncertainty at 1 standard deviation and algae radiocarbon ages at Lake Wellman after King et al. (2020). Samples in cyan circles are the same as in **(a)**. The horizontal black line shows the elevation of the present-day glacier margin; horizontal blue lines show approximate elevations of the Britannia I limit at each location. **(c)** All ages are shown on a logarithmic scale in order to include a significant number of erratics with prior exposure to cosmic rays.

$\sim 14$ until $\sim 8$ ka, although there is considerable variation in timing between sites.

Two erratics from a recessional deposit on the upglacier valley wall record 120 m of glacier thinning by 5.8 ka. Erratics on the valley floor largely agree with this history, showing 100 m of thinning by 6.1 ka. Two of the erratics on the valley floor evidently have had prior exposure to cosmic rays (14-HAT-045-MV and 14-HAT-059-MV). Based on the lack of dated erratics close to the ice margin, we are unable to determine the time at which the glacier margin stabilized at its present position in Magnis Valley. However, if thinning continued at the same rate of $\sim 5\,\mathrm{cm\,yr^{-1}}$, the glacier would have established its modern margin position at $\sim 2$ ka, which is in agreement with the time at which the glacier margin stabilized at Lake Wellman.

For completeness, we include the exposure age and algae radiocarbon chronologies of King et al. (2020) at Lake Well-

man (Fig. 4), which provide key constraints for our flowband modeling in Sect. 3. No new data from Lake Wellman are presented in this paper. The exposure age chronology at Lake Wellman suffers from numerous preexposed cobbles and boulders, resulting in a wide spread of ages (Fig. 4; King et al., 2020; Storey et al., 2010). However, radiocarbon ages of sub-fossil algae from former ice-marginal lakes and ponds show a strong dependence of age on elevation, with a maximum position at 9.5 ka, followed by steady thinning to modern elevations by the Late Holocene (Fig. 4; King et al., 2020). Exposure ages from samples collected along the limit of Britannia I deposition range from 11.5–65 ka

(Fig. 4). Thus, while the cluster of ages at $\sim 20$ ka in Fig. 4b reflect the elevation of the Britannia I limit on the valley walls around Lake Wellman ($\sim 80$–$150$ m higher than the terminal moraine on the valley floor), these ages very likely reflect significant prior exposure to cosmic rays and should be taken only as maximum constraints on the age of the Britannia I limit (King et al., 2020).

### 2.2.2 Darwin Glacier

The Brown Hills lie north of Diamond Hill adjacent to Diamond Glacier – a distributary lobe of Darwin Glacier – and are bordered by the Ross Ice Shelf to the east (Figs. 1c and 5). During glacial periods, Diamond Glacier would likely have crossed the Brown Hills and connected with ice in the Ross Sea. Exposure ages of five erratics from the Brown Hills range from 7–205 ka (Fig. 5). Despite sampling the freshest differentially weathered erratics available, four of these ages are considerably older than the maximum advance at sites on Hatherton Glacier, and they are not ordered by elevation or distance from the Diamond Glacier tongue. We interpret these samples as previously exposed erratics deposited by ice receding from its last maximum position, though it is also possible that they are undisturbed remnants of one or more older deposits. The youngest date from the Brown Hills may also be affected by inherited [10]Be, so we can only conclude that ice retreated from the saddle separating Diamond Glacier from grounded ice in the Ross Sea no earlier than 7 ka.

On the other side of Diamond Hill, at the mouth of Darwin Glacier, deposits are sparse and cannot be correlated with the drift succession described above. On the south slope of the mountain, above Darwin Glacier, fresh glacially worked erratics are scattered over polished and striated bedrock outcrops (Fig. S1). These clasts, stranded by the final stages of recession of Darwin Glacier, extend up to $\sim 135$ m above its present-day surface. Above this, bedrock becomes significantly more weathered, and surfaces are largely devoid of transported boulders and cobbles. The geomorphic transition from surfaces shaped by wet-based glaciation at low altitude to heavily weathered upper slopes resembles transects on other peaks in the TAM and Marie Byrd Land (e.g., Sugden et al., 2005). We attribute the geomorphic transition to differential erosion during the LGM and earlier glaciations. Ice in contact with the low-elevation bedrock was warm-based and erosive, but graded into thinner, cold-based ice cover at higher altitudes. A similar geomorphic gradient occurs in the Brown Hills and on the north slope of Diamond Hill, though bedrock is more weathered everywhere, and there are very few lightly weathered erratics. Till deposits conceal much of the floor of the valley north of Diamond Hill and fill the cirque below and west of the summit at $\sim 800$ m. Exhumation of lightly weathered clasts from these deposits may be a source of erratics with inherited nuclides in this area.

The only relatively fresh, unweathered erratics that we found on Diamond Hill were perched on glacially sculpted bedrock domes overlooking Darwin Glacier, $\sim 10$ km up-glacier from the modern grounding line (Fig. 5). There was no clear limit of deposition, but we did not find any relatively unweathered erratics higher than 135 m above the modern glacier margin. We analyzed eight of these erratics, which yield [10]Be ages spanning the latter half of the Holocene, from $5.2 \pm 0.2$ ka at 135 m above the current glacier margin to $0.3 \pm 0.03$ ka at the current glacier margin (Fig. 5). The rate of thinning matches the record from sub-fossil algae [14]C dates at Lake Wellman (King et al., 2020) (Fig. 4), corroborating our inference that Hatherton Drift represents the last stages of the glacier's recession, separated from the youngest Britannia I deposits by no more than a few hundred years of stillstand or readvance. Thinning during these final stages appears to have been relatively constant between 5.2 and 3.1 ka, after which it slowed.

The lack of fresh deposits above 415 m on Diamond Hill does not necessarily imply ice-free conditions during the last glaciation, as there is no well-defined depositional limit. Based on the preservation of heavily weathered bedrock, we presume that ice above this elevation was cold-based and debris-free and thus did not deposit erratics as it retreated. Steep terrain and the extent of Diamond Hill prevented us from examining all locations, but we climbed to the summit of the mountain via the east (ice-shelf proximal) ridge and descended on the northern (Brown Hills proximal) side, and we did not find evidence of a glacial maximum lateral moraine or recessional deposits above 415 m. Given both potential burial of surfaces by cold-based ice and the lack of erratics that would be suitable for dating with [10]Be, we measured in-situ-produced [14]C in bedrock to determine the maximum thickening and retreat history of Darwin Glacier at Diamond Hill.

On the slopes of Diamond Hill immediately above Darwin Glacier, our highest bedrock sample (14-HAT-026-DH; 472 m a.s.l.) gives an apparent [14]C exposure age of $6.7 \pm 0.7$ ka, extending the retreat history from [10]Be data to 200 m above the modern glacier margin (Fig. 5). Similarly, bedrock sampled < 2 m above the current ice margin (14-HAT-033-DH; 280 m a.s.l.) and adjacent to the 300 year-old ([10]Be age) erratic (14-HAT-032-DH) gives an apparent [14]C exposure age of $500 \pm 200$ years BP. These ages confirm and extend the thinning chronology given by the [10]Be exposure ages. The exposure ages in this transect are too young to record but do not preclude a large Early Holocene deglaciation event, as found at Beardmore, Mackay, Mawson, and David glaciers (Jones et al., 2015, 2020; Spector et al., 2017; Stutz et al., 2020). However, those glaciers only thinned by tens of meters near the glacier mouth following a large drawdown of several hundred meters in the Early Holocene, whereas Darwin Glacier thinned by 200 m since 6.7 ka, a history more similar to that at Reedy Glacier (Todd et al., 2010).

About 500 m above the glacier margin (593 m a.s.l.) on the east ridge of Diamond Hill, the bedrock is at or near saturation concentration with respect to in situ [14]C production–

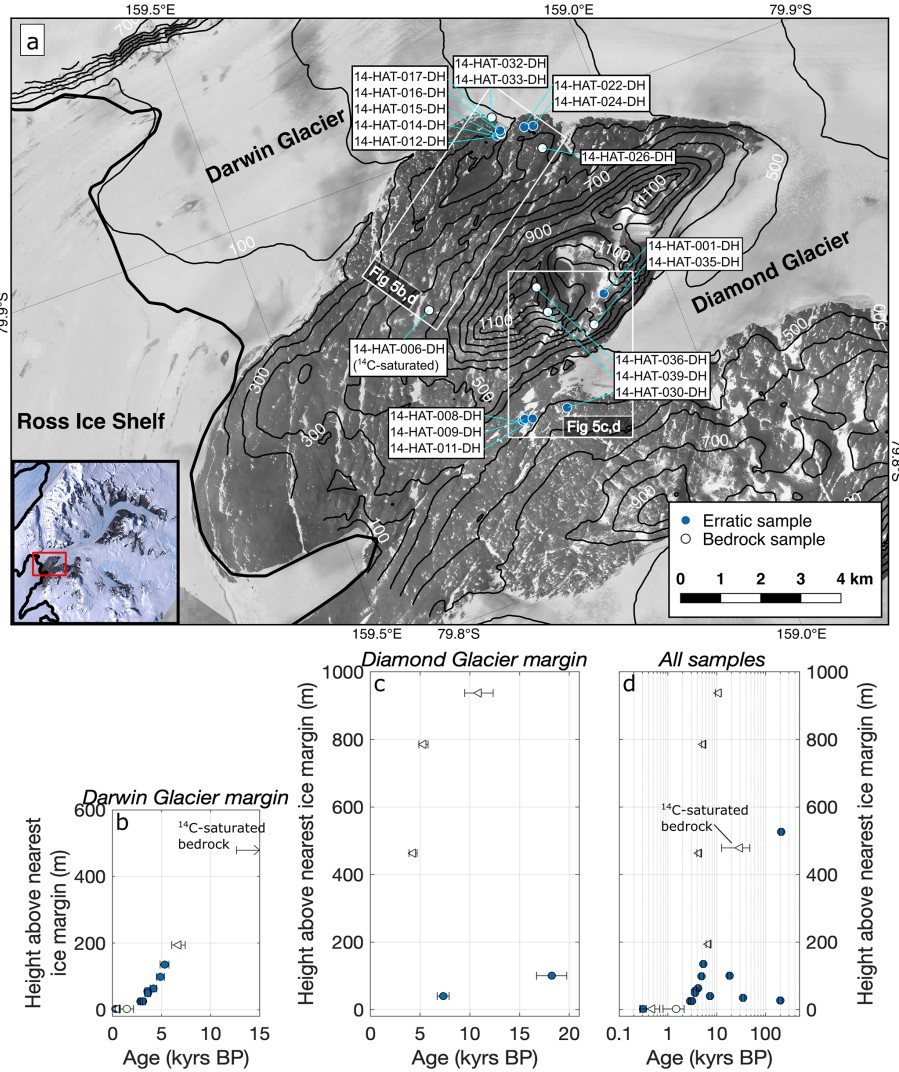

**Figure 5.** TS1 **(a)** Map of Diamond Hill with bedrock and erratic sample locations. The modern grounding line is shown as a bold black line. Sample numbers are included because of the complexity of the topography and exposure age distribution relative to the sample locations in previous figures. Contours and satellite imagery from LINZ (2010), sourced from the NZTopo Database, Crown Copyright Reserved. This product (LINZ Darwin Glacier orthophotos) incorporates data which are © Japan Aerospace Exploration Agency ("JAXA") 2008. Inset imagery from Bindschadler et al. (2008). Missing elevation contours in the lower left corner of the map reflect the edge of the LINZ DEM. White boxes indicate spatial distribution of samples in **(b–d)**. **(b)** Mean exposure ages with total uncertainty at 1 standard deviation from the side of Diamond Hill adjacent to Darwin Glacier. [14]C-saturated bedrock is represented as the lower 1-sigma uncertainty bound and an arrow pointing right in order to keep the recessional transect visible. Triangles and circles indicate bedrock and erratics, respectively. White and blue indicate [14]C and [10]Be, respectively. **(c)** Exposure ages from the Brown Hills and side of Diamond Hill (including summit samples) closer to Diamond Glacier. **(d)** All samples from the Diamond Hill–Brown Hills area on a logarithmic scale to show outliers and [14]C-saturated bedrock. Bedrock [14]C ages give a maximum time of last exposure, because inherited [14]C from exposure before the LGM cannot be quantified. The top of Diamond Hill, > 900 m above the modern glacier surface, was covered during the LGM and was exposed sometime after 11 ka. In contrast, the [14]C-saturated sample ∼ 500 m above present-day Darwin Glacier suggests more modest LGM thickening relative to present. Therefore, it is not straightforward to interpret these data without a glacier flow model, and not all ages can conceivably represent the glacier surface.

decay systematics (14-HAT-006-DH). This sample was either not covered by ice during the last glaciation or was covered for only a brief period (< ∼ 2 kyr). The maximum ice thickness at other locations in the Ross Sea lasted for 3–5 kyr

(Todd et al., 2010; Hall et al., 2015; Spector et al., 2017), so this sample may be an upper bound on LGM ice surface elevation near the modern grounding line. This is a surprising result, given that the LGM ice surface was ∼ 700–900 m

above the modern ice surface at the mouths of other TAM outlet glaciers (Todd et al., 2010; Bromley et al., 2010, 2012; Spector et al., 2017) and 600–700 m above the modern ice shelf at Minna Bluff and on Ross Island (Anderson et al., 2017; Denton and Marchant, 2000). It is at odds with both the extrapolated LGM ice surface 1000 m above present at the mouth of Darwin Glacier of Bockheim et al. (1989), and with the assertion of Storey et al. (2010) that Darwin Glacier did not thicken during the LGM. The estimate of Anderson et al. (2004) of an LGM ice surface 800 m above the modern surface is closer to our potential LGM surface but still appears to be an overestimate.

In contrast, on the summit and north flank of Diamond Hill – above Diamond Glacier – in situ $^{14}$C concentrations are well below saturation. Apparent exposure ages decrease with elevation from $10.8 \pm 1.4$ kyr at the summit of Diamond Hill (1287 m a.s.l.) to $5.3 \pm 0.5$ kyr at 1134 m a.s.l. and $4.3 \pm 0.4$ kyr at 813 m a.s.l. (Fig. 5c). In contrast to the ages from the Darwin Glacier side of Diamond Hill, these ages suggest that the summit was covered by ice or snow for a long period during the last glaciation. Today, Diamond Glacier terminates below these samples in a bedrock saddle between Diamond Hill and the Brown Hills at $\sim 350$ m a.s.l. While Darwin Glacier had thinned to within 135 m of its modern thickness by $\sim 5.1$ ka, these ages seem to suggest that ice north of Diamond Hill was still at least 785 m thicker than present at this time, which is not glaciologically plausible.

We hypothesize that the $^{14}$C-saturated sample (14-HAT-006-DH; 593 m a.s.l.) and the other samples near Darwin Glacier on the south side of Diamond Hill (Fig. 5b) are more representative of large-scale ice fluctuations than the undersaturated samples from higher elevations on Diamond Hill (Fig. 5c) for three reasons. (i) There is no evidence of glacial erosion during the last glacial cycle for the high-elevation samples, and therefore there are no mechanisms for producing older cosmogenic $^{14}$C ages at low elevation than at high elevation, except for local ice or snow cover at those higher elevations. (ii) Two of the three highest-elevation cosmogenic $^{14}$C ages on Diamond Hill are younger than several lower-elevation bedrock and erratic ages closer to Darwin Glacier (even excluding the $^{14}$C-saturated sample), which cannot be explained if all samples are interpreted to represent changes of the large-scale glacier surface. (iii) The summit of Diamond Hill (1287 m a.s.l.) is as high as the local LGM deposits reported by King et al. (2020) 50 km upglacier at Lake Wellman (1130–1298 m a.s.l.; Fig. 4). This interpretation suggests that the exposure ages higher on Diamond Hill are the result of the disappearance of local ice or snow fields. However, given the limited number of in situ $^{14}$C exposure ages at Diamond Hill, these results are equivocal.

### 2.2.3 Summary of geochronologic constraints

We have presented geochronologic data that constrain ice thickness along a flow line from near the head of Hather-ton Glacier to near the modern Darwin Glacier grounding line, based on (i) $^{10}$Be and $^{26}$Al exposure ages of deposits from Bibra, Dubris, and Magnis valleys; (ii) published radiocarbon dates of algae and $^{10}$Be and $^{26}$Al exposure ages from Lake Wellman (King et al., 2020); and (iii) $^{10}$Be ages of glacially transported cobbles and in situ $^{14}$C exposure ages of bedrock from the south side of Diamond Hill adjacent to Darwin Glacier. Collectively, these data suggest glacier advance and/or fluctuations at close to maximum thickness from $\sim$ 14–8 ka. At Lake Wellman, Magnis Valley, and Bibra–Dubris valleys ice was at its maximum limit at 9.5 ka, a time when large outlet glaciers in the southern TAM were thinning at their mouths (Todd et al., 2010; Spector et al., 2017), and the grounding line in the Ross Sea was retreating southwards (Hall et al., 2013; McKay et al., 2016; Goehring et al., 2019a; Prothro et al., 2020). Dates of maximum thickness are not demonstrably different between the three sites on Hatherton Glacier, which also differs from the diachronous behavior of upper and middle Reedy Glacier (Todd et al., 2010), although the sizes of these glaciers and the distance between sites are very different. Our data from Diamond Hill show that Darwin Glacier thinned steadily by 200 m since 6.7 ka. However, the data are equivocal about the magnitude and rate of ice thickness changes near the grounding line prior to 6.7 ka. Data from closest to Darwin Glacier on Diamond Hill suggest slow and steady thinning of $\sim 500$ m through the Holocene, while data at the summit and distal side of Diamond Hill suggest $\sim 900$ m of thinning, with a rapid pulse of $\geq 600$ m thinning in the Early to Mid-Holocene. We prefer the former interpretation, but we do not have enough data to rule out the latter. If the former interpretation is correct, then there is no record of a pulse of rapid thinning at Darwin and Hatherton glaciers, in contrast to some other TAM outlet glaciers both to the north and south (Jones et al., 2015, 2020, 2021; Spector et al., 2017; Stutz et al., 2020); if the latter interpretation is correct, then the DHGS experienced the largest and most rapid drawdown yet recorded in the TAM. However, in our preferred interpretation there are no constraints near the grounding line between 6.7 ka and the LGM (interpreted to be $\sim 9.5$ ka), and a rapid thinning episode of $< 300$ m could conceivably have occurred at the glacier mouth in this data gap. These interpretations are tested in the next section using a numerical glacier flowband model.

## 3  Numerical modeling of glacier fluctuations

In the preceding section, we described the two different deglaciation scenarios suggested by the data from Diamond Hill, near the modern grounding line of Darwin Glacier. To determine which of these scenarios most likely reflects the glaciers' history and evolution, we use a flowband model that simulates how these deglaciation scenarios at the mouth of Darwin Glacier would be reflected in the ice elevation chronologies from upstream at Hatherton Glacier. We set up

three experiments to evaluate interpretations of our data. In Experiments 1 and 2, we test the hypothesis that the [14]C-saturated bedrock at Diamond Hill represents a maximum bound on the LGM ice thickness of Darwin Glacier ($\sim 500$ m thicker than present) and that bedrock [14]C ages from the summit of Diamond Hill reflect changes in local snow or ice fields. In Experiment 1, ice at the mouth of Darwin Glacier thins steadily through the Holocene. In Experiment 2, we include a rapid drawdown in the Early Holocene in a gap in our exposure age chronology at Diamond Hill. This history is consistent with records from other TAM outlet glaciers (e.g., Jones et al., 2015, 2020; Spector et al., 2017). In Experiment 3, we test the hypothesis that the [14]C-saturated bedrock is an outlier, that Darwin Glacier was 950 m thicker than present at the LGM and that a rapid pulse of thinning occurred in the Mid-Holocene. Because climate forcing and boundary conditions are poorly known through time, we use a simple model that minimizes the number of unknown or assumed parameters.

## 3.1 Flowband model description

We modeled Darwin and Hatherton glacier evolution since the LGM using a 1.5-D shallow ice glacier flowband model to evaluate possible deglaciation scenarios consistent with our geochronological data. Applying a relatively simple model to constrain ice-flow conditions and deglaciation behavior has been done for other TAM outlets (e.g., Anderson et al., 2004; Golledge et al., 2014; Jones et al., 2016). The model we apply solves the mass conservation equation to calculate ice thickness evolution using the finite-volume method at 1 km resolution, with 75 grid points for Hatherton Glacier and 141 grid points for Darwin Glacier. This model is computationally inexpensive and reduces the number of poorly constrained ice-flow parameters and boundary conditions. The model domain starts near Diamond Hill at a point 10 km upstream from the modern grounding line where we have geochronological data and extends to the head of the modern glacier catchments. We do not model grounding-line evolution; instead, we use geochronological data to prescribe ice thickness change at a location that is always upstream of the grounding line over the past 20 kyr. This avoids the use of the shallow ice approximation near the grounding line, where the inherent assumptions in the model do not apply. We are thus able to use data to constrain aspects of the model that would otherwise require more complexity, but where a more complex model would introduce additional parameters that are poorly known or completely unknown.

Ice flow in a flowband (1.5-D) is described by the time-evolving mass conservation equation (Cuffey and Paterson, 2010):

$$\frac{\partial H(x,t)}{\partial t} = -\frac{1}{W(x)}\left(\frac{\partial q(x,t)}{\partial x}\right) + \dot{b}(x,t), \tag{1}$$

where $H(x,t)$ is the ice thickness, $q(x,t)$ is the volumetric ice flux, $W(x)$ is the glacier width, and $\dot{b}(x,t)$ is the surface mass balance. Mass balance effects of melting or freezing at the glacier bed are neglected due to a lack of data constraints on these processes. The total ice velocity ($U$) is the sum of contributions from internal deformation ($U_d$) and basal sliding ($U_s$) taken to be of the following form:

$$U = U_d + U_s = f_d H \tau_d^n + f_s \frac{\tau_b^m}{H}, \tag{2}$$

where $f_d$ is the deformation factor, $f_s$ is the sliding factor, $n = 3$ is the flow-law exponent, $m = 3$ is the sliding exponent, $\tau_d$ is the driving stress, and $\tau_b$ is the basal shear stress. Cross-sectional area is accounted for at each grid point. The modern fluxes from small tributary glaciers contributing to Darwin and Hatherton glaciers are estimated using a flux gate calculation, and we make the assumption that these tributary flux contributions are constant through time.

We use the RACMO2.1 5.5 km resolution model (Lenaerts et al., 2012) of the modern surface mass balance, as this is the only data product to include the significant surface ablation that is observed. While these ablation areas do not occur in the same locations observed from satellite imagery, we argue that this is the best choice of surface mass balance because it matches the overall flux out of the glacier system reasonably well (Gillespie et al., 2017). Furthermore, our flowband model is a function of the integral of the surface mass balance and insensitive to the specific pattern of accumulation and ablation at any given location. The blue-ice areas on Darwin Glacier are likely very near their maximum extent, and only small climate fluctuations are required to greatly reduce their area (Brown and Scambos, 2004). Thus, we use the RACMO2.3 (van Wessem et al., 2014) 27 km resolution product – which does not include blue-ice areas – to simulate feasible LGM SMB scenarios. We apply spatially uniform scaling factors of 0.6, 1.0, and 2.0 to explore a range of LGM SMB scenarios. The scaled LGM SMB is then linearly interpolated in time to the modern RACMO2.1 SMB field, which we do not scale. The two width-averaged RACMO SMB inputs, bed topography profiles (Gillespie et al., 2017), and model domains used for Hatherton and Darwin glaciers shown in Figs. S12–S14 in the Supplement.

We first optimize a steady-state version of the flowband model to estimate poorly constrained ice-flow parameter values by minimizing the mismatch between the modeled and observed modern glacier surface elevation and the modern surface velocity at each model grid point. This strategy has been applied in related problems (see Golledge et al., 2014). We vary the basal sliding factor ($f_s$) and the ice-flow deformation factor ($f_d$) defined at each model grid point within a plausible range of values ($f_s$: $10^{-12}$ to $10^{-10}$ m$^2$ Pa$^{-3}$ yr$^{-1}$ (Anderson et al., 2004); $f_d$: $3.65 \times 10^{-18}$ to $1.1 \times 10^{-15}$ Pa$^{-3}$ yr$^{-1}$). While we have limited information available to constrain the values of the sliding and deformation parameters, the inferred patterns of these parameters

generate surface elevation and velocity profiles that match modern values within their uncertainties. The deformation factor and basal sliding parameter vary spatially over the model grid, but we assume that the spatial patterns that op-
5 timize the fit to modern data have not changed over the past 20 kyr because there is not enough information available to constrain a different choice for the parameter values in the past. However, we are using this simple model to constrain the time evolution of the glacier system that is consistent
with our geochronological observations and not to interpret inferred parameter values and patterns of controls on ice flow.

The downstream boundary condition for Darwin Glacier is a prescribed surface elevation through time based on our glacial geologic data. The surface elevation at the down-
15 stream boundary of the major tributary Hatherton Glacier is prescribed at each time step by the value at the adjacent node along the Darwin Glacier flow line where the glaciers intersect, similar to what was done by Anderson et al. (2004). At each time step, the flux from Hatherton Glacier is added to
20 Darwin Glacier at that node, the surface of Darwin Glacier is recalculated, and the surface elevation at the mouth of Hatherton Glacier is reset to match the updated surface of Darwin Glacier. While the model domains are set up to contain the modern drainages of Darwin and Hatherton glaciers,
these catchment boundaries may have changed in the past. Thus, we evaluate the influence of time-varying flux that enters the domain from the upstream boundary, as required by a flux-balance calculation. We are also able to modulate flux entering the glacier through the upstream boundary. Be-
cause our model domain boundary is at the modern catchment boundaries, any flux added or removed at the upstream boundary represents a change in catchment size. We use this adjustment of flux as a way to investigate the role of catchment size changes in the evolution of the glacier profiles.

In order to compare the model output with our chronologies from each location, we must account for the fact that our geochronology data record the elevation of the glacier margin through time, rather than the glacier centerline that is calculated in the flowband model. Thus, the exposure and ra-
diocarbon ages represent a minimum elevation for the glacier centerline through time. While the glacier centerline generally lies 100 m above its margin today alongside our study sites, this would not necessarily have been the case at the LGM. We determine the possible range of glacier centerline
elevations at the LGM by calculating a linear best-fit curve to the LGM deposits in the valley floor (far from the modern glacier) and on the valley walls (close to the modern glacier). By extrapolating this linear fit out to the center of the glacier, we obtain a maximum constraint on the LGM
glacier centerline elevation because the glacier should not have a concave-up profile in the transverse direction. We use the elevation of the highest LGM deposits as a minimum constraint because the glacier margin can never be higher than the centerline. In the absence of further constraints, we use
the mean of these elevations as the estimated LGM centerline

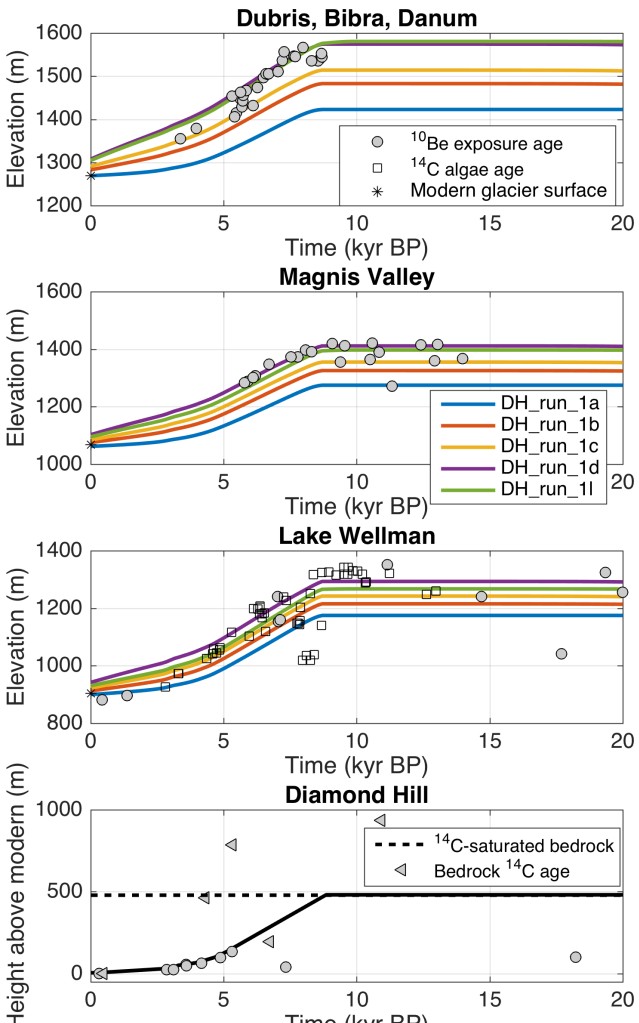

**Figure 6.** Flowband model results from Experiment 1 compared with surface exposure ages from this study and surface exposure and radiocarbon ages from King et al. (2020), which have been projected to centerline elevations using the method described in Sect. 3.1. The two best-fitting scenarios require either a larger catchment for both glaciers at the LGM (green curve; DH_run_1l) or an accumulation rate far higher than modern (purple curve; DH_run_1d). The black curve in the Diamond Hill panel represents the prescribed ice surface history at the downstream boundary.

elevation. We estimate LGM glacier centerline elevations of 1355 ± 67 m at Lake Wellman, 1402 ± 56 m at Magnis Valley, and 1550 ± 45 m at Dubris–Bibra valleys. Elevations from recessional deposits were then projected to estimate the centerline elevation by determining the percentage of total thin- 60 ning each sample elevation represents and multiplying this by the height of the estimated LGM surface above the modern glacier elevation. Model results are compared against these projected elevations in Figs. 6–8.

At Diamond Hill, we lack the clear limit of LGM deposits 65 necessary to project samples to a glacier centerline eleva-

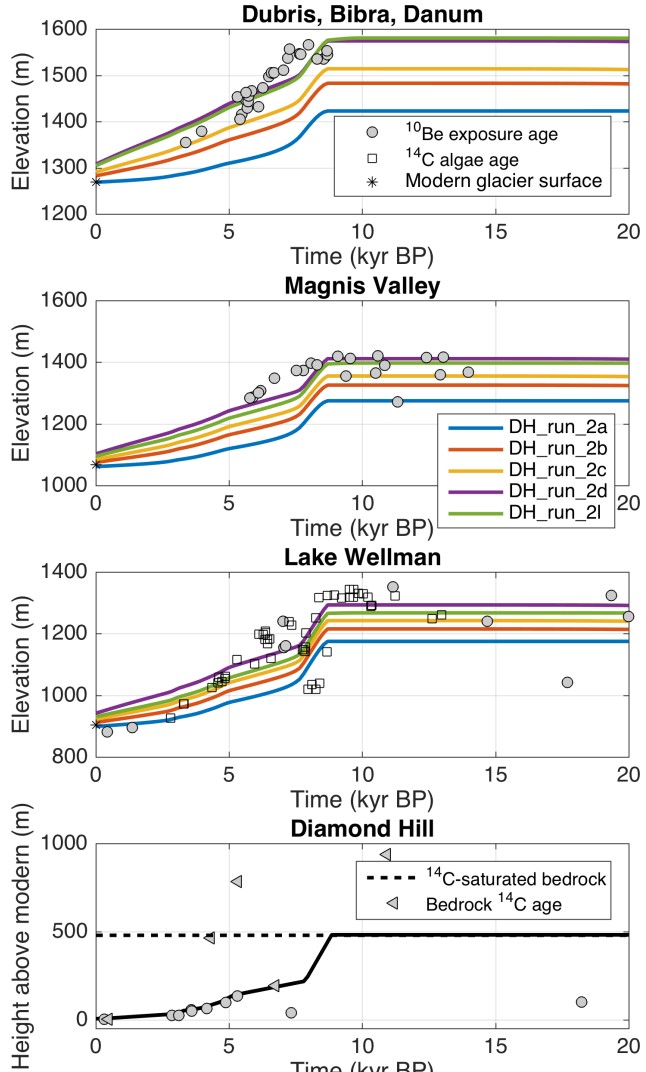

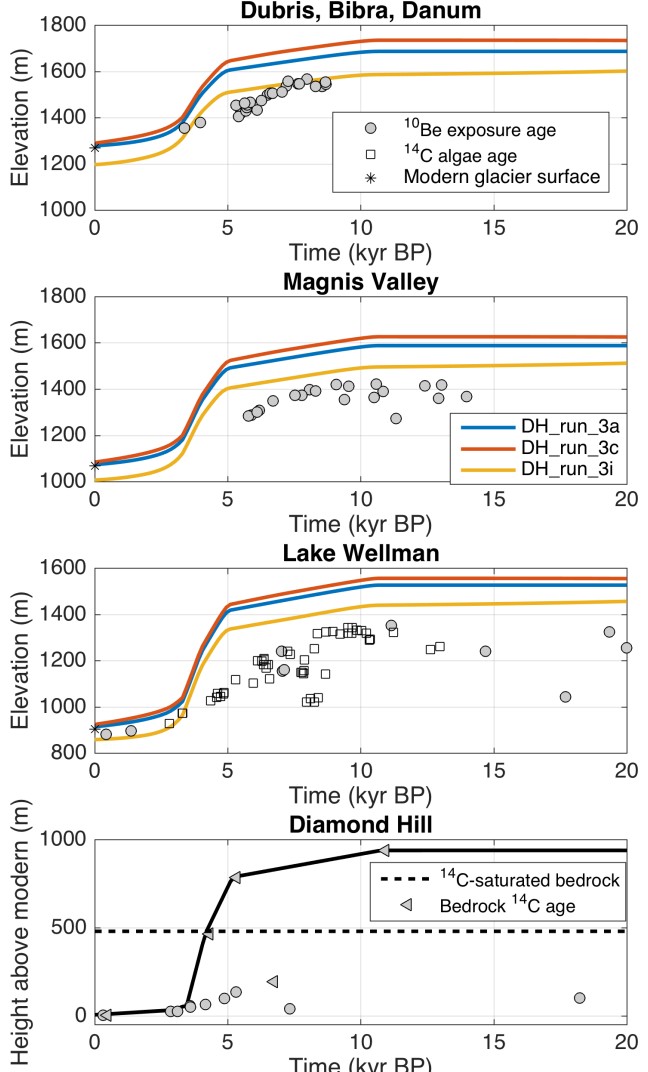

**Figure 7.** Flowband model results for Experiment 2. All inputs are the same as for Experiment 1, except for the downstream boundary condition, which contains a period of rapid deglaciation at 9–8 ka. It is evident that this event would have been recorded in the deposits at Hatherton Glacier, but there is no clear evidence of this in our chronologies. Thus, the case of slow thinning in Experiment 1 provides a better fit to the data.

**Figure 8.** Flowband model results for Experiment 3, in which we assume that the Holocene exposure ages high on Diamond Hill represent thinning of the main trunk of Darwin Glacier. Fitting the LGM thickness of Hatherton Glacier requires a much smaller Darwin Glacier catchment at the LGM, but we cannot fit the shape of the chronologies with any combination of simple assumptions about surface mass balance or catchment area. Thus, we rule out this scenario as unlikely and conclude that the young exposure ages at high elevation on Diamond Hill may reflect more local ice fluctuations.

tion. Instead, we convert the sample elevations to a height above the nearest ice margin and use those values as constraints for glacier fluctuations and for the experiments discussed in Sect. 3.2. Likewise, we do not have a reliable means of estimating a range of possible LGM centerline elevations. However, this uncertainty is greatly outweighed by the uncertainty in which samples represent the elevation of Darwin Glacier versus local snow and ice fields (i.e., the distinction between the scenarios in Experiments 1 and 3).

## 3.2 Transient experiments

Input values for all experiments in this section are summarized in Table 1.

**Table 1.** Inputs for flowband model runs.

| Run code | Modern SMB | LGM SMB | External flux Darwin ($m^3 \, yr^{-1}$) | External flux Hatherton ($m^3 \, yr^{-1}$) |
|---|---|---|---|---|
| DH_run_1a | RACMO2.1 | RACMO2.1 | 0 | 0 |
| DH_run_1b | RACMO2.1 | 60 % RACMO2.3 | 0 | 0 |
| DH_run_1c | RACMO2.1 | RACMO2.3 | 0 | 0 |
| DH_run_1d | RACMO2.1 | 200 % RACMO2.3 | 0 | 0 |
| DH_run_1l | RACMO2.1 | 60 % RACMO2.3 | $1.2 \times 10^8$ | $6.4 \times 10^7$ |
| DH_run_2a | RACMO2.1 | RACMO2.1 | 0 | 0 |
| DH_run_2b | RACMO2.1 | 60 % RACMO2.3 | 0 | 0 |
| DH_run_2c | RACMO2.1 | RACMO2.3 | 0 | 0 |
| DH_run_2d | RACMO2.1 | 200 % RACMO2.3 | 0 | 0 |
| DH_run_2l | RACMO2.1 | 60 % RACMO2.3 | $1.2 \times 10^8$ | $6.4 \times 10^7$ |
| DH_run_3a | RACMO2.1 | RACMO2.1 | 0 | 0 |
| DH_run_3c | RACMO2.1 | 60 % RACMO2.3 | 0 | 0 |
| DH_run_3i | RACMO2.1 | 44 % RACMO2.1 (scaled to Taylor Dome record) | $-1.4 \times 10^8$ | 0 |

### 3.2.1 Experiment 1: 500 m of gradual thinning at the mouth of Darwin Glacier

**Modern surface mass balance, constant catchment size**

Since we have no constraint on changes in accumulation and ablation patterns through time for the DHGS, we first set up a transient run in which the modern surface mass balance pattern is kept constant in time. The glacier surface at the outlet of Darwin Glacier is held at 500 m above the modern surface until 9 ka to match the onset of thinning at Hatherton Glacier and then lowered according to the geochronologic constraints from Diamond Hill (Fig. 5b), using the assumption that the $^{14}$C-saturated bedrock sample represents an upper bound on the LGM ice surface of the main trunk of Darwin Glacier. This is based on the interpretation that the Holocene $^{14}$C exposure ages higher up on Diamond Hill represent cover by local ice or snow fields. The results of this experiment (DH_run_1a) are shown in Fig. 6.

There is essentially no lag time between the application of the elevation change at the mouth of Darwin and the response of Hatherton Glacier in this scenario. Thus, the onset of thinning at each location along the profile of Hatherton Glacier occurs almost simultaneously with the onset of thinning at Diamond Hill. This is consistent with our chronologies from the valleys alongside Hatherton Glacier, each of which indicates that retreat began at ∼ 8–9 ka. However, using the modern surface mass balance at the LGM leads to a thinner modeled Hatherton Glacier than is indicated by the glacial deposits. This discrepancy is more pronounced further up-glacier: at Lake Wellman (close to the confluence of Darwin and Hatherton), the model predicts an LGM ice surface ∼ 75 m below the LGM deposits at the entrance to the valley; at Magnis Valley, the model underpredicts LGM thickness by

∼ 100 m; at Dubris and Bibra valleys, the modeled glacier is ∼ 170 m too thin.

There are three possible explanations for the underprediction of LGM ice thickness in this experiment. First, it is possible that the current size of the glacier catchments is smaller than it was during the LGM. Observations of striated bedrock near the head of Hatherton Glacier indicate that at some unknown time in the past, ice likely flowed over into the Hatherton valley from what is now the catchment of Byrd Glacier (Bockheim et al., 1989). Likewise, there are no constraints on the former size of the Darwin Glacier catchment. It is thus possible that more ice fed into the DHGS due to larger catchment size during the LGM. Second, our assumption of a constant pattern of surface mass balance through time is too simple. A small climate perturbation would greatly decrease the size of the ablation zones on the modern glaciers, leading to a more positive surface mass balance (Brown and Scambos, 2004). However, Hatherton Glacier must have had ablation areas (both wind scouring and surface melt) at its margins during the local LGM in order to provide the meltwater necessary to create the ponds that hosted algae, and in order to deposit material in the valleys. Furthermore, accumulation rates tend to increase as the atmosphere warms during a glacial termination; the Taylor Dome ice core record shows a roughly 100 % increase in accumulation rate between 12 and 0.7 ka (Monnin et al., 2004). The presence of blue-ice areas due to scouring by katabatic winds would likely lead to a more complicated change in surface mass balance, but the reduction in snow accumulation during glacial periods would likely have been a common feature throughout the Transantarctic Mountains. Third, our records at the mouth of Darwin Glacier (Fig. 5) are equivocal as discussed above. The fact that the complicated and spatially sparse bedrock $^{14}$C record is difficult to interpret leaves open the possibility that the average LGM ice thickness at Diamond Hill was

higher than we have tentatively concluded. We next explore the effects of time-evolving surface mass balance and variable flux into the glacier canyons.

**Time-evolving surface mass balance**

We next attempt to determine the magnitude of glacial–interglacial change needed to match the records from Hatherton Glacier using only variations in SMB. The modern surface mass balance is taken from RACMO 2.1. We defined the LGM surface mass balance by multiplying the RACMO 2.3 surface mass balance – which does not include ablation areas – by a scaling factor. We use values of 60 %, 100 %, and 200 %. The choice of 60 % is made to match the accumulation rate change in the Vostok ice core between the LGM and present (Petit et al., 1999). This is of course a crude estimation, but it results in reasonable LGM accumulation rates of 2–10 cm yr$^{-1}$. The surface mass balance is varied linearly in time between the modern and LGM states. After a 5 kyr spin-up to allow the model to equilibrate with LGM climate, the LGM SMB is held constant until 15 kyr TS2, and then varied linearly to the modern SMB.

In this scenario, 60 % and 100 % SMB scalings lead to underprediction of LGM ice thickness by about 50–100 m at all three Hatherton Glacier sites (DH_run_1b and DH_run_1c in Fig. 6). Using a 200 % scaling of RACMO2.3 SMB at the LGM leads to excellent agreement between the glacier model and most of the records from Hatherton Glacier (DH_run_1d in Fig. 6). LGM surface elevations are matched to within a few tens of meters at all three locations, the timing and rate of thinning agree very well, and the modern surface elevations are reproduced to within 40 m at all three locations.

While the 200 % scaling results in a reasonable match between the model and surface-exposure age data, it produces an unlikely surface mass balance history. This requires LGM accumulation rates of 7–30 cm yr$^{-1}$, compared to ∼ 3 cm yr$^{-1}$ accumulation at Taylor Dome during the same time period (Monnin et al., 2004). The fact that at least some surface ablation had to occur to create ice-marginal ponds means total surface mass balance may have been even lower than at Taylor Dome. This scenario also requires the overall surface mass balance to decrease during the termination of the glacial period, which is unlikely. Although this scenario fits our data quite well, we search for a more reasonable explanation.

**Added flux to account for changing catchment area**

As noted above, there are no constraints on the size of the glacier catchments at the LGM. While the modern glacier catchments are kept small by their proximity to Byrd and Mulock glaciers, they could have been larger at the LGM. Due to the low surface slopes of the EAIS and the small size of the DHGS catchment relative to those of Byrd and Mulock glaciers, just a moderate amount of localized thick-

ening or thinning could drastically increase or decrease the drainage area contributing to the DHGS. To account for this uncertainty, we take the simple approach of adding ice flux to the upstream boundary of the Darwin and Hatherton glacier model domains. A challenge of this approach is to define a time series of flux change across the upstream boundaries. Because this value would have varied in time and would not necessarily have been the same for both glaciers, any solution is likely to be non-unique. However, the goal of this exercise is not to calculate the magnitude of the flux entering the glaciers at the LGM, but rather to determine whether changing catchment areas could be a reasonable means of achieving the LGM ice thickness of Hatherton Glacier.

By adding a moderate amount of flux to both Darwin and Hatherton glaciers and using the scaled SMB of 60 % modern for the LGM, we achieve a good fit to the LGM limits and retreat histories at all three locations along Hatherton Glacier (DH_run_1l in Fig. 6). Given the uncertainties in bed properties, accumulation rate, and tributary fluxes through time, we consider this to be a satisfactory fit to the data. If the catchment boundaries stabilized before the grounding line of Darwin Glacier stopped retreating, this could also explain the slowdown of thinning at Dubris and Bibra valleys several kiloyears prior to the slowdown of thinning at Lake Wellman, ∼ 30 km closer to the grounding line.

We now consider whether the amount of additional incoming flux required to match the LGM limits is a physically reasonable quantity. For the additional $6.4 \times 10^7$ m$^3$ yr$^{-1}$ added to Hatherton and the additional $1.2 \times 10^8$ m$^3$ yr$^{-1}$ added to Darwin and a reasonable LGM accumulation rate of 3 cm yr$^{-1}$ over the East Antarctic Plateau, this would require 2100 and 4100 km$^2$ of additional LGM catchment area for Hatherton and Darwin glaciers, respectively. This represents a ∼ 75 % increase over the modern catchment area (Gillespie et al., 2017) but only a 5 % decrease in the Mulock Glacier catchment or a 0.5 % decrease in the Byrd Glacier catchment areas based on values calculated by Stearns (2011). Because this total areal change is roughly equivalent to the reported uncertainty in the Mulock Glacier catchment area, and only about 11 % of the uncertainty in the Byrd Glacier catchment area, we consider this amount of divide migration over a glacial–interglacial cycle to be reasonable.

### 3.2.2 Experiment 2: rapid Early Holocene thinning at the mouth of Darwin Glacier

This experiment, like Experiment 1, assumes the [14]C-saturated bedrock at Diamond Hill represents the upper limit of the LGM ice surface. Other records from the western Ross Embayment show a rapid drawdown event at 9–8 ka, presumably indicating widespread deglaciation of the region in the Early Holocene (Spector et al., 2017). There is no record of abrupt thinning at Hatherton Glacier, but such an event could have occurred in the data gap between the [14]C-saturated and the 6.7 kyr old bedrock (separated by 300 m of elevation) at

Please note the remarks at the end of the manuscript.

the mouth of Darwin Glacier. We explore this possibility by imposing a rapid thinning event of 275 m (similar to thinning at Beardmore and Mackay glaciers (Jones et al., 2015; Spector et al., 2017)) from 9–8 ka, followed by gradual thinning consistent with the glacial geologic constraints. Surface mass balance and additional flux are the same as the scenarios from Experiment 1. The results of this experiment are shown in Fig. 7.

In all model runs, the rapid thinning imposed at the mouth of Darwin Glacier propagates upglacier in the model with no significant lag. While the amplitude of the signal decays with distance upglacier, it is still readily detectable in the modeled glacier changes at Dubris and Bibra valleys. This leads to a much poorer fit to the data than the situation with gradual thinning at the Darwin Glacier mouth in Experiment 1. This shows that the records from Hatherton Glacier are not consistent with an episode of rapid thinning at the mouth of Darwin Glacier, which implies that the DHGS did not respond to the last deglaciation in the same way as other outlet glaciers to the north and south (e.g., Jones et al., 2015; Spector et al., 2017).

### 3.2.3   Experiment 3: 950 m of thinning at the mouth of Darwin Glacier, with a rapid pulse of thinning in the Mid-Holocene

This experiment, unlike Experiments 1 and 2, assumes the $^{14}$C-saturated bedrock at Diamond Hill is merely an outlier and that the exposure ages at higher elevations represent the fluctuations of Darwin Glacier. While we have argued that the higher-elevation ages are not as representative of the surface of Darwin Glacier as they are of more local ice configuration (see Sect. 2.2.2), we do not have enough data to prove this assertion. Thus, in this experiment we evaluate a scenario in which Darwin Glacier was 950 m thicker than present at the LGM and covered the top of Diamond Hill, after which it thinned through the Holocene, with a rapid pulse initiating around 5 ka. This thinning history is defined by the bedrock samples in Fig. 5c. Results are shown in Fig. 8.

For RACMO2.1 accumulation rates and modern catchment boundaries, this deglaciation scenario leads to a modeled Hatherton Glacier that is ∼ 160–200 m too thick at all locations on Hatherton Glacier (DH_run_3a in Fig. 8). Scaling the RACMO 2.3 SMB by 60 % leads to a worse fit due to the lack of ablation zones (DH_run_3c in Fig. 8). The fit to Hatherton Glacier LGM ice thickness can be improved somewhat by moving the Darwin Glacier catchment boundary ∼ 25 km into the model domain (i.e., decreasing the catchment area) and scaling the RACMO 2.1 SMB to the Taylor Dome records, which is the lowest of all our explored SMB scenarios (DH_run_3i in Fig. 8). There is no geologic evidence for this decrease in catchment area at the LGM, but it may not be an unreasonable amount of change, given the enormous catchment areas of Byrd and Mulock glaciers. However, the imposed rapid thinning at Diamond Hill prop-

agates upglacier in all three model runs, leading to a very poor fit to the records of Hatherton Glacier fluctuations, and the modern Hatherton Glacier surface is underestimated by 50–100 m. Thus, the pattern of thinning cannot be matched using simple assumptions about catchment size or surface mass balance, and thus we conclude that this is an unlikely scenario. This supports our earlier hypothesis that the elevation transect nearest the modern margin of Darwin Glacier at Diamond Hill (Fig. 5b) most closely represents the major ice thickness fluctuations since the LGM, while the chronology from the Brown Hills and high on Diamond Hill represents more local ice or snowfield fluctuations (Fig. 5c).

## 4   Ice sheet model ensemble

### 4.1   Ice sheet model description

Our flowband model is only applied to the grounded portions of Darwin and Hatherton glaciers and thus cannot be used to directly examine the effect of grounding-line retreat on the thickness of the DHGS. To address this, we ran a 48-member ensemble of model simulations using the Pennsylvania State University 3-D ice sheet model (PSUICE) (e.g., Pollard and DeConto, 2012a) to examine the effect of grounding-line retreat in the Ross Sea on ice thickness at the mouth of Darwin Glacier. PSUICE uses a combination of the shallow ice and shallow shelf approximations along with a parameterization of grounding-line flux (Schoof, 2007) to allow for full continent-scale simulations on kiloyear to megayear timescales, as well as an optimized model parameter set that is tuned to match data from around West Antarctica over the past 20 kyr (Pollard et al., 2016). As in previous work with this model (e.g., Pollard and DeConto, 2012a; Pollard et al., 2016), ocean temperatures are taken from the global 22 kyr simulation of Liu et al. (2009), and modern atmospheric forcing is from the ALBMAP v1 dataset (Le Brocq et al., 2010). Modern atmospheric forcing is scaled through time based on benthic $^{18}$O records, and melt rates are calculated from ocean temperatures as described by Pollard and DeConto (2012a). Iceberg calving is based on the crevasse depth formulation of Nick et al. (2010), as described by Pollard et al. (2015). We do not include ice shelf hydrofracture or cliff failure mechanisms (e.g., Pollard et al., 2015; DeConto and Pollard, 2016), but these are very unlikely to have any effect during climates colder than today. Modern bed topography is from Fretwell et al. (2013) and is adjusted for ice loading through time as described by Pollard and DeConto (2012a) and Pollard et al. (2016). For grounded ice, we use the same basal sliding coefficient field as Pollard et al. (2016), which is obtained via the inverse method described by Pollard and DeConto (2012b).

Because we are only interested here in the ice sheet in the Ross Sea sector, the optimized parameter set (Pollard et al., 2016) may not be the most suitable for our regional investigation. So, we investigate the model sensitivity to the choice

of key parameter values. First, we run the model at 40 km resolution over the whole continent from 125 to 25 ka, using two sea-level forcings (Lisiecki and Raymo, 2005; Spratt and Lisiecki, 2016), the optimized parameter set of Pollard et al. (2016), and a modern ice sheet configuration as the initial condition to approximate the last interglacial. We then increase the model resolution to 20 km and run the model over the whole continent from 25 ka to present, still using the optimized parameter set. This 20 km resolution run provides initial conditions and boundary conditions for the ensemble of 48 nested model simulations at 10 km resolution since 20 ka in which we vary model parameters around the values optimized from Pollard et al. (2016). The modern ice discharge from the DHGS ($\sim 0.2\,\mathrm{Gt\,yr^{-1}}$ at present; Gillespie et al., 2017) is very small compared to that from Byrd and Mulock glaciers ($\sim 27.5\,\mathrm{Gt\,yr^{-1}}$ at present; Stearns, 2011), so resolving the DHGS is likely not necessary to model large-scale and long-term evolution of the ice sheet in the Ross Sea sector. We are interested in the relationship between ice thickness at the TAM front and the position of the grounding line and thus do not need to model the individual glaciers at high resolution. The 10 km resolution was chosen as a trade-off between computing time and the ability to resolve pinning points that may have a large effect on grounding-line evolution. While very high resolution ($\leq 1$ km; unfeasible for these simulations) is required to very accurately model grounding-line retreat (Gladstone et al., 2012), the response of grounded ice to grounding-line retreat is less resolution-dependent. So, even though the 10 km resolution likely leads to inaccuracies in the grounding-line migration, the response of grounded ice should be reasonably well captured.

We vary four model parameters that were examined by Pollard et al. (2016): a basal sliding coefficient on the modern seafloor, isostatic rebound rate, ice shelf melt sensitivity to ocean temperatures, and a calving rate factor. We also use two different sea-level curves to explore the effect of this external forcing (Lisiecki and Raymo, 2005; Spratt and Lisiecki, 2016). Parameter values used in the ensemble are listed in Sect. 4.2. Because our geochronologic results show more gradual and recent deglaciation of the DHGS than at other locations in the TAM (Spector et al., 2017), our choice of parameter values is intended to slow down grounding-line retreat to Darwin Glacier relative to the optimized parameter set of Pollard et al. (2016), while still leading to Early Holocene deglaciation of much of the western Ross Embayment (Spector et al., 2017). However, we also recognize that the parameters could vary in space and time, for example due to heterogeneities in ice shelf strength, mantle viscosity, till properties on the seafloor, and sub-shelf ocean circulation. Furthermore, higher-order ice physics and adaptive mesh refinement to resolve grounding-line evolution may be required to fully capture the processes involved in the retreat but require too much computational expense to be included in an ensemble of model runs over timescales of tens of thousands of years.

## 4.2 Ice sheet model results

Results from our ice sheet model ensemble are shown in Table 2. We find that no combination of the parameters we explored reproduces the slow and steady drawdown through the Holocene that we infer from the flowband model and glacial geologic data at Diamond Hill or at other locations along Hatherton Glacier. The basal sliding coefficient leads to a difference of $\sim 600$ m ice thickness between the fastest sliding value ($10^{-5}\,\mathrm{m\,yr^{-1}\,Pa^{-2}}$) and the slowest ($10^{-7}\,\mathrm{m\,yr^{-1}\,Pa^{-2}}$). While the ice at the mouth of Darwin Glacier begins thinning $\sim 1$ kyr earlier for the faster sliding scenarios, modern ice thickness is achieved by 5–6 ka in all model runs, which is much earlier than our glacial chronologies show ($\sim 2$–3 ka). While none of the model runs in this ensemble achieve what we consider to be a good fit to our data from Diamond Hill, the three clusters of model runs defined by the three basal sliding coefficients display internally consistent behavior characteristics that are worth examining in order to better understand the sensitivity of the ice sheet near the mouth of Darwin Glacier.

Each group of model runs shows a complex relationship between rates of thickness changes near Darwin Glacier and the rate of grounding-line retreat (Fig. 9). When grouped by the value of basal sliding coefficient, all three groups of model runs predict slow, almost linear thinning at the mouth of Darwin Glacier as the grounding line retreats from its LGM position to Cape Crozier. Once the grounding line is upstream of Cape Crozier, the model behavior diverges based on the basal sliding parameter. While all runs see a drastic decrease in the rate of grounding-line retreat, the runs with the largest sliding coefficient ($10^{-5}\,\mathrm{m\,yr^{-1}\,Pa^{-2}}$) exhibit a slower rate of thinning at Darwin Glacier, while the thinning rates in the runs with lower sliding values continue to increase before they suddenly slow. Once the grounding line retreats behind Minna Bluff, grounding-line retreat and thinning at Darwin Glacier accelerate in all runs; however, the largest acceleration in grounding-line retreat corresponds to the lowest response in thinning rates at Darwin Glacier. The acceleration of grounding-line retreat as it reaches Discovery Deep is not generally reflected in thinning rates at Darwin Glacier. This suggests that some pulses of rapid grounding-line retreat may not have left a strong signal of thinning at the mouth of Darwin Glacier. Given the complex relationship the model predicts between ice thinning rates at the mouth of Darwin Glacier and the rate of grounding-line retreat, we refrain from interpreting our glacier chronologies in terms of the rate of grounding-line retreat. However, it is still valid to interpret the cessation of major changes in glacier thickness $\sim 3$ ka (Fig. 5) as indicating that the grounding line reached close to its modern position at the mouth of Darwin Glacier around that time.

**Table 2.** Parameter choices for ice sheet model ensemble.

| Parameter description | Name in Pollard et al. (2016) | Values | Optimal value from Pollard et al. (2016) |
|---|---|---|---|
| Asthenospheric rebound timescale | TAUAST | 1000, 2000 years | 2000 years |
| Sub-shelf melting factor | OCFAC | 0.5, 1 | 1, 3 |
| Basal sliding coefficient on seafloor | CSHELF | $10^{-5}, 10^{-6}, 10^{-7}\,\mathrm{m\,yr^{-1}\,Pa^{-2}}$ | $10^{-5}\,\mathrm{m\,yr^{-1}\,Pa^{-2}}$ |
| Calving factor | CALV | 0.7, 1 | 1 |
| Sea-level curve | – | Lisiecki and Raymo (2005), Spratt and Lisiecki (2016) | Only used Lisiecki and Raymo (2005) |

## 5 Discussion

Darwin and Hatherton glaciers continued to adjust to regional deglaciation until $\sim 3$ ka, which we interpret to indicate that the grounding line likely did not arrive at its present location until about that time. Darwin and Hatherton glaciers lie roughly halfway between McMurdo Sound and Beardmore Glacier, which both deglaciated in the Early to Mid-Holocene (Hall et al., 2004; Spector et al., 2017), yet the timing of grounding-line arrival at Darwin Glacier is apparently much more recent. This could imply that a large region extending across Byrd, Darwin, and Mulock glaciers remained grounded for about 4 kyr after the grounding line in the central Ross Embayment had retreated further south. It remains an open question as to how far along the TAM front this grounded ice persisted and whether it comprised a single, grounded ice mass or local piedmont lobes (Lee et al., 2017).

It is unclear whether the chronology from Skelton Glacier supports this interpretation, as there are no deposits at the grounding line that date to the last deglaciation (Jones et al., 2017). A small number of samples $\sim 16$–50 km from the modern grounding line suggest that most thinning in the Skelton Névé had occurred by 6 ka (Jones et al., 2017; Anderson et al., 2020), but the number of samples is too small to determine the rate of thinning or the timing of its onset. More data from Skelton Glacier are needed for thorough comparison with the chronology at Darwin and Hatherton glaciers.

While Darwin and Hatherton glaciers are not necessarily representative of glaciers with small catchments throughout the TAM, it is worth noting that the factors complicating the interpretation of the records presented here may apply at other TAM outlet glaciers as well. The two potentially complicating factors relevant to the DHGS are possible catchment boundary migration and convergent flow with larger neighboring glaciers. Our flowband modeling suggests the possibility of a larger catchment at the LGM than at present, which has since been pirated by the much larger Byrd and Mulock glaciers. This change in catchment size would have been essentially negligible for Byrd and Mulock glaciers, while representing a $\sim 75\%$ increase relative to the modern DHGS catchment area. We have investigated this effect only through a simple parameterization of added flux through the upstream boundary, and our simple model setup is not able to distinguish between possible causes of catchment boundary migration. Whether this may have been caused by changing basal conditions, changing flow speeds of Byrd and/or Mulock glaciers, changing patterns of surface mass balance, or some other process remains an open question. More sophisticated modeling in two or three dimensions may elucidate the controls on potential catchment migration since the LGM. The effect of convergent flow with much larger neighboring glaciers may be more important to the DHGS than to other TAM outlet glaciers because its neighbors are two of the largest glaciers in the TAM and the presence of Minna Bluff ensures convergent flow. However, convergent flow could still have played a role in controlling glacier response to grounding-line retreat at other TAM outlets. We suggest that these two factors – catchment boundary migration and flow convergence downstream – should be taken into account when planning fieldwork on or analyzing chronologies from other relatively small TAM outlet glaciers.

The 1.5-D flowband model is an effective tool for investigating these glacier histories because it minimizes the number of unknown parameters and assumptions. However, the model we employ is simple and has a number of limitations.

1. The shallow-ice model assumes that vertical shear stresses are dominant and does not account for lateral and longitudinal stresses.

2. The model also does not account for the transition from grounded to floating ice, and this restricts application of the model to locations that are always upstream of the modern grounding line and does not allow us to explore mechanisms for grounding-line retreat. This means that the model will likely be less accurate for situations in which the grounding line is close to its modern position. However, as this is only likely to be the case at the end of the model run, we find this an acceptable trade-off.

3. The model domain excludes the ice shelf and grounding line, and we impose an ice thickness boundary condition at the glacier mouth. This enables us to test the scenarios outlined above, but it also means that we cannot distinguish between different causes of thinning at the mouth, e.g., grounding-line retreat or a change in buttressing.

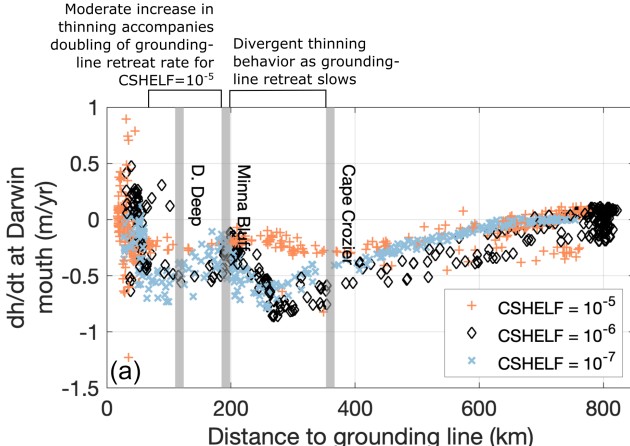

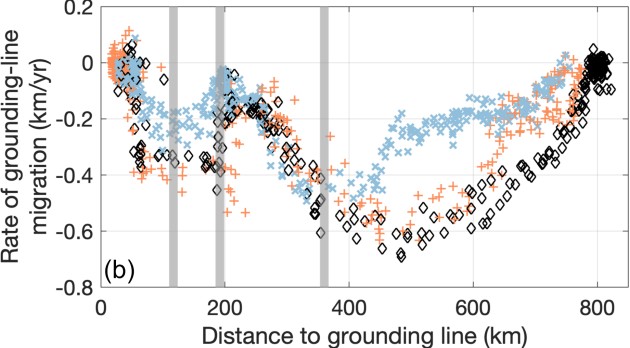

**Figure 9.** Results from the 48-member PSUICE ice sheet model ensemble. **(a)** The relationship between rate of ice thickness change at the mouth of Darwin Glacier and the distance to the grounding line for all ice sheet model runs. The basal sliding parameter (CSHELF) has by far the largest effect of the parameters we explored in the ensemble. Gray bars indicate the locations of the features we hypothesized to affect the rate of grounding-line retreat: Cape Crozier, Minna Bluff, and the Discovery Deep. **(b)** Grounding-line migration rate as a function of distance to the grounding line from Darwin Glacier. Negative values indicate retreat. Retreat slows after the grounding line retreats past Cape Crozier, but thinning rates at Darwin Glacier increase or slightly decrease, depending on the value of the basal sliding parameter. Grounding-line migration accelerates significantly after the grounding line has retreated past Minna Bluff. Perhaps counterintuitively, the acceleration of grounding-line retreat into the back-sloping Discovery Deep is not always accompanied by increased thinning rates at the mouth of Darwin Glacier. This indicates that there might not be a large thinning signal in glacial deposits near the mouth of Darwin Glacier, even for a very rapid pulse of grounding-line retreat.

4. Our tuned basal sliding and enhancement factor fields do not change in time because we have no constraints on how these values would change. However, this means that we might be missing effects of changes in bed properties or ice temperature that influence the rate of sliding and ice deformation over time. Additionally, the lack of an explicit treatment of lateral drag means that this

is implicitly included in our optimized values for the basal sliding parameter. While that is a simplification, changes in lateral drag due to changing ice geometry over time are likely to be small, given the small changes in ice thickness in comparison to the length of the modern ice–bed interface.

5. While ice flux can be added or subtracted at the upstream end of the flow line model domain to represent changes in the upstream catchment area, this is a simple representation of how large-scale changes in glacier geometry may influence glacier evolution. We also do not explicitly model tributaries other than Hatherton Glacier, but instead we represent these by an addition of mass flux to the values along our primary glacier centerline.

We view these simplifications as justified given the long timescales of interest, the large uncertainties in parameter values and environmental forcings, and the starkly different scenarios between which we seek to discriminate. More sophisticated modeling that incorporates the ice shelf and grounding line may better elucidate the dynamics of these glaciers during the last deglaciation (e.g., Jones et al., 2021).

Steady thinning of the DHGS through the Holocene supports a later arrival of the grounding line relative to glaciers to the north and south (Hall et al., 2015; Jones et al., 2015; Spector et al., 2017). In our flowband model, thinning at the mouth of Darwin Glacier propagates rapidly upglacier; thus, the deposits alongside Hatherton Glacier can be interpreted as recording the relative rates of ice thinning at the mouth of Darwin Glacier. Unlike at some other TAM outlet glaciers, there is no record of an exceptionally fast period of thinning on the order of hundreds of meters in $< 2\,\mathrm{kyr}$, as at Mackay, Beardmore, Mawson, and David glaciers (Jones et al., 2015, 2020; Spector et al., 2017; Stutz et al., 2020) at Hatherton Glacier, in agreement with our interpretation of the sparse data near the modern grounding line. However, our ice sheet model ensemble analysis suggests that periods of exceptionally fast grounding-line retreat – such as across the Discovery Deep region – would not necessarily correspond to noticeably faster ice thinning rates at the mouth of Darwin Glacier. A 2- to 4-fold increase in the modeled rate of grounding-line retreat as ice in Discovery Deep goes afloat does not cause faster drawdown at Darwin Glacier for most of the parameter values we investigated. In the scenarios with the fastest basal sliding, there is an increase in the rate of thinning when the grounding line retreats into Discovery Deep. However, this lags the change in the grounding-line retreat rate and reaches a maximum thinning rate after the grounding line has already slowed. This suggests that the time at which the grounding line arrived at the glacier mouth may be deducible from geologic data but that there are significant complications in drawing conclusions about relative rates of grounding-line retreat from glacier-elevation proxies.

While we are not able to determine the rate of grounding-line retreat, we can make a first-order estimate of the sensitivity of Darwin and Hatherton glaciers to grounding-line position using records from elsewhere in the Ross Sea. The grounding line retreated past Ross Island at > 8.6 ka (McKay et al., 2016), and McMurdo Sound became free of grounded ice between 6.5 and 9 ka (Hall et al., 2004; Jones et al., 2015; Anderson et al., 2017; Christ and Bierman, 2020). This is similar to the time at which Hatherton Glacier began to retreat from its last highstand (8–9 ka), and our glacier modeling suggests Darwin likely began thinning around this same time as well. Thus, Darwin and Hatherton glaciers may be strongly dependent on the ice configuration near Ross Island. The results of our ice sheet model ensemble suggest that this relationship is complex, as different parameter choices lead to opposing behavior as the grounding line retreats past Cape Crozier. This discrepancy should be addressed with further modeling studies of outlet glacier response to grounding-line retreat and/or more data from near the modern grounding line of Darwin Glacier. The more rapid thinning of Mackay, Beardmore, Mawson, and David glaciers could be due to the steeply back-sloping bed topography in the immediate vicinity of the glacier mouths (Jones et al., 2015, 2020, 2021; Spector et al., 2017; Stutz et al., 2020).

Our estimated LGM ice surface elevation of ∼ 600 m a.s.l. at the mouth of Darwin Glacier from our flowband modeling needs to be reconciled with the elevations of the LGM limit of 637 m a.s.l. at Minna Bluff (Denton and Marchant, 2000) and 520 m a.s.l. at Mt. Discovery (Anderson et al., 2017). The flowband of Darwin Glacier likely would have passed between these sites at the LGM (Denton and Hughes, 2000), and with our inferred ice surface elevation the glacier surface profile would have been very flat at the LGM, resulting in lower driving stresses than we would expect in comparison to modern glaciers and ice streams (see Cuffey and Paterson, 2010, p. 297). We identify three factors that could reconcile these elevation differences and lead to a sufficient LGM surface slope between Darwin Glacier and Minna Bluff to Mt. Discovery.

First, there is significant uncertainty in the large-scale ice surface elevation that corresponds to the [14]C-saturated bedrock elevation at Diamond Hill. Ablation by katabatic winds can cause a large difference in surface elevations around a nunatak (Bintanja, 1999). The downglacier side of Diamond Hill may have been subject to especially vigorous wind-driven ablation at the LGM and thus could have been significantly lower than the large-scale ice surface. As we have shown, the fit of our flowband model results to the data at Hatherton Glacier is more sensitive to rates of thickness change at the glacier mouth than to the absolute LGM ice thickness. Therefore, the possibility that the glacier centerline LGM ice surface could have been significantly higher (perhaps ∼ 100–200 m) than the [14]C-saturated bedrock sample at Diamond Hill is not necessarily in conflict

with our preferred deglaciation scenario, as long as thinning was smooth, steady, and initiated around 9 ka.

Second, none of the reported LGM surface elevations include the effects of glacial isostatic adjustment, which could have varied between Darwin Glacier and Minna Bluff. Geothermal flux south of Ross Island may be 58 %–125 % higher than at Diamond Hill (Morin et al., 2010; An et al., 2015). If the crustal densities around Minna Bluff and Ross Island are correspondingly lower than at the mouth of Darwin Glacier, the loading of the crust by grounded ice would have caused greater isostatic depression at Minna Bluff than at Darwin Glacier, potentially helping to reconcile these elevation differences. However, the ICE6G model (Argus et al., 2014; Peltier et al., 2014) predicts only small amounts of isostatic depression during the LGM, with slightly more depression at Diamond Hill (78 m) than at Minna Bluff (63 m), so this is not our preferred explanation. But we note that ICE6G assumes 100 % areal ice cover over the cell containing Darwin Glacier, which we know to be incorrect, and so this may be an overestimate of isostatic depression at the LGM.

Third, the majority of the Ross Sea drift on Minna Bluff is found significantly lower than the reported LGM maximum of 637 m a.s.l. reported by Denton and Marchant (2000). Only one small patch occurs higher than 500 m a.s.l.; this defines the reported 637 m a.s.l. LGM limit (see Plate 1 in Christ and Bierman, 2020). As there are no numeric age constraints at this location, it is possible that this represents a short-lived (∼ 1 kyr) transient maximum that could have also occurred at Diamond Hill but would not be detectable by our in situ [14]C measurement. The short-lived maximum at Lake Wellman reported by King et al. (2020) may support this hypothesis.

Ice sheet model simulations of the last deglaciation – including the ensemble presented here, as well as those of Pollard et al. (2016), Kingslake et al. (2018), Lowry et al. (2019, 2020), and Albrecht et al. (2020) – predict rapid grounding-line retreat to the mouth of Darwin Glacier early in the last deglaciation (around 10 ka) as the grounding line retreats between Minna Bluff and the TAM. The models generally agree poorly with glacial geologic data in the McMurdo Sound region as well, predicting deglaciation thousands of years earlier than glacial geologic data suggest. This is not surprising, given that ice sheet model response to climate forcing is strongly sensitive to a number of poorly known parameters (Pollard et al., 2016; Lowry et al., 2020), climate forcing itself is poorly known, and the topography beneath the modern-day Ross Ice Shelf used by these models is coarsely resolved (Fretwell et al., 2013). Including new constraints on bed topography (Tinto et al., 2019; Morlighem et al., 2020) could lead to different model behavior. Additionally, the large ensemble experiments that are tuned to glacial geologic data (Pollard et al., 2016; Albrecht et al., 2020) attempt to fit data from around West Antarctica or the whole continent and thus might sacrifice some fit to data in the Ross Embayment to improve the overall ensemble score when tuning

spatially homogeneous parameter values. Tuning parameters to fit geochronologic constraints using a large ensemble over the Ross Embayment alone could lead to a better fit between models and data.

We suggest that the relatively recent and slow deglaciation of the DHGS is likely due to the convergence of Byrd and Mulock glaciers near the mouth of Darwin Glacier and/or lateral drag past Minna Bluff and Cape Crozier, which would both have led to dynamic ice thickening that opposed grounding-line retreat. Convergent flow can counteract the acceleration of dynamic thinning as the grounding line retreats down a reverse bed slope (Gudmundsson, 2013), and positive strain rates measured south of Minna Bluff show that convergent flow and compression are causing dynamic thickening of the ice shelf in this region today (Thomas et al., 1984). Whillans and Merry (2001) identified Cape Crozier and Minna Bluff as controlling obstacles to the flow of the modern Ross Ice Shelf, and we expect they would have had a similar effect during the last deglaciation. At 8 ka, grounded ice likely covered the Pennell and Ross banks as ice rises connected by an extension of the Ross Ice Shelf, and an ice shelf may have extended far north and east of Ross Island until ~ 3 ka (Prothro et al., 2020). These features may have contributed additional buttressing to the Byrd–Darwin–Mulock flowband. Together, the effects of convergent flow, lateral drag, and buttressing by ephemeral ice rises could have created a protected embayment that resisted grounding-line retreat for longer than glaciers farther to the south, even given the steeply back-sloping bed topography of Discovery Deep.

## 6 Conclusions

We have dated deposits of the DHGS in order to help constrain the timing and pattern of grounding-line retreat in the Ross Embayment since the last glacial maximum. The data suggest a later and slower deglaciation than that experienced by glaciers both farther south (Spector et al., 2017) and farther north (Jones et al., 2015, 2020; Stutz et al., 2020). The scarcity of glacial deposits near the modern grounding line of Darwin Glacier makes it difficult to directly interpret the thickness and timing of the LGM at the mouth of Darwin Glacier. We used a 1.5-D glacier flowband model to evaluate possible deglaciation scenarios for Darwin Glacier that are consistent with our new data, and we used a 3-D ice sheet model to evaluate the sensitivity of model reconstructions of regional deglaciation to poorly known model parameter values.

We find that LGM glacial deposits in ice-free valleys alongside Hatherton Glacier record up to 350 m thickening relative to present at Magnis Valley and 300 m at Dubris and Bibra valleys, in agreement with Joy et al. (2014) and King et al. (2020), and in contrast to the minimal thickening inferred by Storey et al. (2010). The glacier margin extended several kilometers into each valley during its maximum before re-

ceding slowly and steadily from ~ 9 to ≤ 2.8 ka. Exposure ages of bedrock and glacially transported cobbles < 10 km upstream of the modern grounding line at Diamond Hill record 190 m of thinning of Darwin Glacier between 6.7 ka and 300 years BP, with most of that thinning complete by 3 ka. Prior to 6.7 ka, the Darwin Glacier chronology is equivocal. Maximum bedrock $^{14}$C exposure ages indicate two possible, but conflicting, thinning histories at the mouth of Darwin Glacier: the LGM surface of Darwin Glacier was either 190–500 m above the modern glacier and thinned steadily through the Mid-Holocene, or it was ~ 950 m thicker than present with ~ 600–800 m of rapid thinning in the Mid-Holocene.

We used a glacier flowband model to test the two conflicting deglaciation scenarios suggested by the data at the mouth of Darwin Glacier. Flowband model results show that our data are most consistent with ~ 500 m slow and steady thinning at the mouth of Darwin Glacier between ~ 9 and ~ 3 ka, accompanied by a large decrease in catchment area through the Holocene. Rapid deglaciation at the glacier mouth in the Early to Mid-Holocene – like that recorded in deposits at several other outlet glaciers in the TAM (Jones et al., 2015, 2020; Spector et al., 2017; Stutz et al., 2020) – is not consistent with our geochronologic data. Our finding that the model requires a larger DHGS catchment at the LGM to fit the geochronologic data suggests that records from outlet glaciers with small catchments like the DHGS may be at least partially influenced by the dynamics of their larger neighbors. The required catchment area changes are essentially negligible for Byrd and Mulock glaciers, while representing a 75 % larger DHGS catchment at the LGM.

The apparent lack of an episode of rapid thinning at the DHGS is surprising, given that grounding-line retreat into the Discovery Deep would presumably have caused acceleration and drawdown of upstream grounded ice. An ensemble of 48 runs using a 3-D ice sheet model indicates that using glacial geologic data of ice elevation change to constrain even relative rates of grounding-line retreat is not straightforward. Counter to expectation, periods of more rapid grounding-line migration may correspond to relatively constant thinning rates at outlet glacier mouths. Conversely, the presence of pinning points such as Cape Crozier reduces the rate of modeled grounding-line retreat, but not necessarily the rate of thinning at Darwin Glacier. Thus, rates of thinning from glacial geologic records at Darwin and Hatherton glaciers should not be directly interpreted as reflecting rates of grounding-line retreat. This may also be true of other glaciers that we have not examined here, although we do not suggest that this will be the case in general. However, the arrival of the grounding line close to its modern configuration must be reflected by the slowing of upstream thinning. Thus, we interpret the timing of grounding-line arrival at the mouth of Darwin Glacier to be ~ 3 ka. The relatively slow thinning of Darwin and Hatherton glaciers through the last deglaciation could be the result of convergent flow of Darwin Glacier

with Byrd, Mulock, and Skelton glaciers, along with lateral drag due to the flow past Minna Bluff and Cape Crozier.

*Code and data availability.* Cosmogenic isotope data are archived on the Informal Cosmogenic-Nuclide Exposure-Age Database at http://ice-d.org (last access: 3 May 2021). Flowband model code is available at https://github.com/mkoutnik/DH_code (last access: 23 May 2019). Penn State Ice Sheet model code is not publicly available, as it is the intellectual property of David Pollard. Flowband model output is archived at https://doi.org/10.5281/zenodo.4735623 (Hillebrand et al., 2021). Ice sheet model output is archived at https://doi.org/10.5281/zenodo.4734615 (Hillebrand, 2021).

*Supplement.* The supplement related to this article is available online at: https://doi.org/10.5194/tc-15-1-2021-supplement.

*Author contributions.* TRH, JOS, BH, and CK conducted fieldwork and sample collection. TRH, JOS, and CK prepared and analyzed samples for $^{10}$Be and $^{26}$Al. KN and BG prepared and analyzed rock samples for $^{14}$C. MK and TRH developed the glacier flowband model code. MK, TRH, and HC designed, performed, and analyzed glacier flowband model experiments. TRH designed, performed, and analyzed PSU ice sheet model experiments. MKG provided gridded ice thickness and bed topography products for the flowband model experiments. TRH wrote the manuscript with input from all authors.

*Competing interests.* The authors declare that they have no conflict of interest.

*Acknowledgements.* In situ $^{14}$C measurements were generously funded by awards from the University of Washington Department of Earth and Space Sciences and Quaternary Research Center. Jan Lenaerts and Melchior van Wessem provided RACMO data. David Pollard provided the PSU ice sheet model code. Knut Christianson provided access to the computer cluster used for the ice sheet model ensemble, and Ed Mulligan provided invaluable IT support. We thank Lauren Simkins and Richard Selwyn Jones for insightful and constructive reviews that greatly improved this paper. Chris Stokes also provided helpful comments. We also thank the Antarctic Support Contractor, the US Air National Guard, Ken Borek Air, and Petroleum Helicopters Inc. for logistical support.

*Financial support.* This research has been supported by the National Science Foundation (grant nos. 1246110, 1542756, and 1246170).

*Review statement.* This paper was edited by Chris R. Stokes and reviewed by Richard Selwyn Jones and Lauren Simkins.

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
