# Peer review of "Holocene thinning of Darwin and Hatherton glaciers, Antarctica and implications for grounding-line retreat in the Ross Sea"

_The Cryosphere, 2020_

## Referee Comment (RC1) · Lauren Simkins (Referee) · 18 Jan 2021

Review summary This study integrates cosmogenic exposure ages and numerical modeling to investigate the history of Darwin and Hatherton glaciers. The major conclusion is that the two glaciers are in-sync and experienced 500 m of thinning between 9,000 and 3,000 year before present. While the geochronology is convincing enough to support the major conclusion, the addition of some of the model brings novel insight to SMB and ice flux that sheds light on the geochronology. However, the flowline modeling work beyond experiment 1 is not supported by the geochronology (which is the backbone of this study) and, in my mind, unnecessary.

Very little motivation and broader significance is presented, leaving the reader either

not convinced of the importance of the study or having to do extra work to make those leaps on their own. In both the abstract and introduction, the problem, questions, and broader significance are completely lacking. I urge the authors to explicitly address "why is this study needed/timely/significant?" and "what can we learn from small catchments of the TAM during/since the LGM?" The text is dominated by results, some of which seem unnecessary to present which end up shrouding the important results/interpretations. Careful consideration is needed regarding what results and methodological details to divulge and to what degree they are integral to the story. The text would benefit greatly from revision of the introduction, results, and conclusion.

In summary, the study has significance for understanding the history of Darwin and Hatherton glaciers. I, however, think the impact of this work is underdeveloped as it is currently written. Explicit hypotheses or questions would go a long way in boosting the importance of the study to better understand (a)synchroneity of neighboring glacial systems and relatively small glacial systems of the TAM and EAIS in general.

Line comments Abstract 16: explicitly state what is the Ross Sea Ice Sheet 20: as an example of removing unnecessary words, the phrase "of Darwin and Hatherton glaciers" is not needed, rather implied 21-22: remove "Our modeling shoes that" and start with "The constraints" 23: replace semicolon with period and capitalize "rapid" 29: change to "...convergent ice flow from..." Throughout text when using a series of proper nouns with a shared word, like in reference to more than one drift deposit, valley, or glacier, the correct grammar is to not capitalize the shared word; therefore, you should write e.g., "Darwin and Hatherton glaciers".

Introduction Prior to section 1.1, what was done is mentioned but not why the work was done; therefore, the text in its current state will likely not engage a broader audience who works outside of the Ross Sea/Transantarctic Mountain region. Some of the text from sections 1.2 and 1.3 - sections that would benefit from being more concise - could be reworked and moved up to set the stage for this work. Otherwise, as written, there is a considerable amount of text to read before any motivation/significance for

the study can be gleaned. 35: remove "In this paper," and capitalize "we" 36: first use of "D" in main text; therefore, rephrase to 1.5-dimensional (D) 41: an updated chronology for the offshore record in the context of terrestrial chronology is provided in Prothro et al. (2020) in QSR that would be interesting to consider in this study, which suggest major reorganization of the EAIS grounded in the Ross Sea during the Holocene. 110: replace last glaciation with "LGM" 111: modify to read "...thickening grounded ice..." 146: exhumation by what process? 150: specify by what magnitude of time is considered "prolonged" - hundreds to thousands of years? 157-160: as an example of making the text more concise throughout the manuscript, here is the original text: King et al. (2020) were able to date the advance and retreat of Hatherton Glacier in the Lake Wellman valley using radiocarbon dating of freeze-dried algae. They showed that the glacier advanced to the Britannia-I limit at 9.5 kyr BP, and argued that both the majority of their own exposure ages and those of 160 Storey et al. (2010) were overestimates due to prior exposure of the samples. And here is what I would suggest: Based on radiocarbon dating of algae in Lake Wellman, Hatherton Glacier advanced to the Britannia-I limit at 9.5 kyr BP suggesting exposure ages in the region were overestimates due inheritance from prior exposures (King et al., 2020). 169: Still by this point in the text, motivation, significance, questions, and/or hypotheses are not explicitly addressed. Also it would be useful at some point to address how the study is organized: first present the chronology/thickness changes and interpretations and then modeling to help explain the geological observations.

Records of glacier fluctuations I suggest having one subsection for Dubris, Bidra, and Magnis valleys and Lake Wellman and another for Brown Hills, Diamond Glacier, and Diamond Hill, followed by the summary section 203: Goehring et al. (2019a) I presume 203-208: why mention the failed preparation technique in detail rather than simply referring to the Nichols and Goehring (2019) manuscript? That is six lines of text that could be half a line to one line of text. 221-222: move up to preceding paragraph 247: remove "if the ages at the...of prior exposure," 253: remove "Based on...along Hatherton Glacier". Modeling results have not been presented yet, and I presume you

went into the model with the indication that you could use the chronology from the sites together representing a single system. 276: remove "()" 305-307: Considering modern ice-flow velocities that suggest flow solely due to internal deformation, paleo-evidence of warm-based ice indicates a major change in ice-bed coupling and the potential for major changes in magnitudes of ice flow velocity. I think this is an interesting point that is not taken any further, but could be followed up on in either/both the modeling or/and discussion section. 403-409: suggest removing all text starting with "we prefer…" to the end of the paragraph

Numerical modelling of glacier fluctuations Both "modelling" and "modeling" used in the text . Combine sections 3.1-3.3 and make more concise, particularly information in section 3.2 should be in section 3.1 to explain constraints on variables. 432: remove first sentence 443: rephrase to "...grid point (c.fl., Golledge et al., 2014)." 444: curious what range of basal sliding is used here that is considered plausible 447-450: rephrase to "While we have limited information available to constrain the values of the sliding and deformation, inferred patterns of these parameters generate a surface elevation and velocity profiles that match modern values within their uncertainties." 450-451: this sort of statement would be best at the beginning of section 3.1 to explain why the modeling work was conducted. 461: I would argue that you are not investigating the role of catchment geometry, rather catchment flow change which could result from a number of glaciological changes. Section 3.3 can be much shorter and, like mentioned above, integrated into a combined section with 3.1 and 3.2 473: are you estimating 2-sigma uncertainty? Or calculating it? Not clear what is being done to geologically constrain centerline positions for the model… 480: "Figures 10-12." 481-486: seems out of place For the transient experiments, I would argue that experiments 2 and 3 are distracting and not useful in the scheme of the work presented here. It is interesting to note in the discussion that these small glacial systems show no geological evidence for rapid thinning. Experiment 3 could be retained if authors feel that can be justified given lack of constraints and seemingly arbitrarily chosen LGM ice thickness. I am, however, a proponent of the presentation of negative results, but feel like these extra steps in the

modeling are not supported or justified by the geological observations and thus do the study a disservice by distracting readers with unnecessary negative findings that do the opposite of what the modeling work is supposed to do - which I interpret as aiming to help explain the geological observations.

Ice sheet model ensemble I am largely on board with this work, but think the text could be much more concise (apologies for repeating that point).

Discussion is fine; however, broader significance for other small glacial systems in the TAM and beyond could be integrated here.

Conclusion would benefit greatly from not being written as a list. I suggest that authors have a go at linking the findings and major outcomes between the geochronology and the modeling work. As currently written, the reader has to do a lot of work to piece the listed bits together and come away from this text with take-aways that will invoke further thought, questions, and applications.

Figures The figures look really nice. Figure 1: the bathymetry colormap you have used is not categorized, like the legend suggests. Include Bidra Valley Suggest combining figures 2 and 3, figures 4 and 5, and figures 6 and 7. Make sure consistent reference to figure panels, e.g. figure 3 caption that uses "(a)" and "panel c"

References were not checked.

Supplementary documents Units for ages in the data table are needed.

Please also note the supplement to this comment:
https://tc.copernicus.org/preprints/tc-2020-356/tc-2020-356-RC1-supplement.pdf

---

## Referee Comment (RC2) · Richard Selwyn Jones (Referee) · 3 Feb 2021

**Summary and recommendation**

The paper presents an interesting combination of geochronological data (10Be, 26Al and 14C ages), flowband modelling and 3D ice sheet modelling, focused on Darwin and Hatherton Glaciers. The authors conclude that Darwin Glacier thinned gradually by 500 m during the Holocene, in contrast to other glaciers in the Transantarctic Mountains, and that this behaviour was possibly due to the convergent ice flow and lateral drag in the vicinity of Byrd Glacier.

A lot of work is covered in this paper. The manuscript is generally well structured, written and illustrated, the methods seem sound, and the conclusions are mostly sup-

ported by the results. The study provides new high-quality data for this region, a new understanding about the ice dynamics at Darwin and Hatherton Glaciers, and ideas related to deglaciation of the Ross Sea.

I recommend publication in The Cryosphere following some revision.

There are a few aspects where I think the paper could be improved. First, the rationale for the study needs to be better communicated, particularly in the Abstract and also Introduction. Why should the reader care? Second, it is not always clear what is new and what is "borrowed" from previous studies. This concerns the geochronological constraints and interpretations at Lake Wellman, and the ice sheet model ensemble. Third, the maximum ice limits at 9.5 kyr BP are not clearly supported by this study. There are no saturated in situ 14C samples from Magnis Valley and Bibra/Dubris Valleys, and the interpretation of Lake Wellman is based on another study, with limited description of why older samples are ignored. Fourth, the flowband modelling skips over some limitations in this approach, while the ice sheet model ensemble is short on detail. More details on these points are included below.

**Detailed comments**

Title: 'grounding-line retreat' is included, yet the paper concludes that it is not possible to gauge grounding-line retreat from the thinning data, and the modelling does not include grounding-line retreat of this system. Consider removing.

Line 17, and elsewhere: The Early and Mid/Middle Holocene are now formalised sub-epochs, so should be capitalised.

Line 42, and elsewhere: Grounded ice in the Ross Sea is not a true self-contained ice sheet. Rather it is a sector of the Antarctic ice sheet, or a region comprising ice from the East and West Antarctic ice sheets. Consider rephrasing.

Line 62: Dramatic thinning is also now recorded at Mawson Glacier (Jones et al.,

Geology, 2020) and David Glacier (Stutz et al., TCD, in review).

Lines 68-69: Can you describe Darwin and Hatherton Glaciers in relation to the glaciers mentioned above?

Lines 99-104: It might also be worth mentioning the model buttressing studies of Furst et al. (2016) and Reese et al. (2018), which show that this region has a large buttressing effect on the ice sheet (albeit based on the current ice configuration).

Lines 156-163: Slightly confusing timeline of research. It would probably be better to present in publishing order (Storey et al., 2010, Joy et al., 2014, King et al., 2020).

Lines 167-169: This last point seems a bit lost. Consider making it a third goal.

Line 172 ("erratics"): Technically, if they are erratics then their lithology should differ to the local geology; how they are 'erratic' should be described here. It would also be worth providing a broad description of the erratic size - cobbles, boulders?

Line 172 ("Diamond Hill, as well as from Bibra, Dubris and Magnis Valleys"): It would be useful to state here how far each of these sites are from the modern grounding line.

Line 173: "freshest" should be made clearer. Rocks that appeared least weathered?

Lines 174-180: As previous studies in the area (e.g. Storey et al., 2010; Joy et al., 2014) have highlighted the importance of sampling strategy, particularly regarding reworking by cold-based ice, it would be useful to point out how your approach is similar/different.

Lines 193-195: It would be useful here to clarify what is detectable given uncertainties in erosion rates and measurements.

Lines 218-219: It would be useful to quantify the impact of the scaling scheme on the exposure ages - what percent difference between schemes (e.g. Stone (2000) and Lifton et al. (2014) nuclide-specific scheme)? Also, what production rate dataset is used? This should be described here.

Lines 250-255: It is not clear how the maximum thickness is determined to be from at least 14 kyr. No potentially older samples were collected from higher elevations, and there are no in situ 14C measurements from here to test for saturation.

Lines 275-278: If no new ages from Lake Wellman are presented in this study, then this needs to be stated clearly here, indicating that data from previous studies are used to assist interpretation of ice history in the area. You also need to explain in more detail why the higher elevation erratic ages at 20 kyr are "pre-exposed" and therefore ignored.

Line 394: The text in this section does not convincingly support the claim that ice was at its maximum at 9.5 kyr. There are no saturated in situ 14C samples from these sites, and there is limited justification of why older samples are ignored. Clarify and expand on this point in the above text, or revise this claim.

Line 396: You should probably include a study that directly dates grounding-line retreat in the Ross Embayment (e.g. McKay et al., 2016; Prothro et al., 2020).

Line 414: Jones et al. (2016) is not included in the references.

Lines 416-417: The model flowband/domain needs to be shown somewhere, perhaps in Figure 1c.

Lines 420-423: While this point is largely true, the flowband is a very simple model and so the limitations should be stated. What key stresses are not calculated? What aspects of the deglacial history can versus cannot be tested?

Lines 423-430: As far as I can see, this model does not account for lateral drag, which the conclusions indicate is potentially very important. How might this impact the flow-band results?

Line 428: How many grid points? What is the horizontal resolution used?

Line 438: How is SMB scaled using the scaling factors for LGM scenarios?

Lines 452-453: To what extent does modification of the downstream boundary repre-sent perturbations (e.g. reduce buttressing, grounding-line retreat) downstream of this point? As mentioned above, you should make clear what processes are and are not accounted for, and what can be tested.

Line 468, and elsewhere: What is meant by 'LGM' in the context of these sites? The proposed local ice maximum? The Antarctic or global glacial maximum?

Lines 597-599: This interpretation should be in Section 2.

Lines 646-648: Is this a new set of model simulations, and not output from previously published simulations? This would be a considerable effort and requires more descrip-tion here. No need to describe all the of the model physics, but state that this can be found in a different paper. What ocean and atmosphere forcings are applied? What are the values of other parameters that were not examined? Some readers would expect to see these details, even if described in the Supplementary Material.

Lines 712-714: How might the model resolution impact this finding?

Line 722: McMurdo Sound deglaciated in the Mid Holocene, so "Early to Mid Holocene" is probably more correct.

Lines 728-279: I agree that more data are required from the lower reaches of Skelton Glacier. But, to be picky, the erratic age at 6 kyr is <20 km from the modern grounding line, which is not "far" considering the scale of these glaciers. Consider rephrasing.

Line 735: In what context is there no record of "an exceptionally fast period of thin-ning"? How is this quantified? Is this relative to modern observations or other thinning estimates from TAM?

Lines 748-749: As most thinning at Mackay Glacier to the north occurred at 7.5-6.5 kyr, McMurdo Sound became free of ground ice between 6.5 (not 7.5) and 9 kyr BP.

Line 755: Also the case with Mawson Glacier and David Glacier (see references

above).

Lines 773-777: The argument regarding glacial isostatic adjustment could be clearer. How greater isostatic depression at Minna Bluff could resolve the elevation differences between Darwin Glacier and Minna Bluff needs a little more explanation. Is this just considering the relative lowering of the LGM elevations at Minna Bluff, or also the impact on the age constraints? It is possible to estimate the relative elevation difference?

Lines 796-797: This was done for the Ross Embayment alone in Lowry et al. (2019, 2020), yet still some large differences in the timing of thinning between simulations and geological data.

Lines 805-809: While this is a useful discussion with the suggestion that the confluence of Byrd-Hatherton-Darwin Glaciers may explain gradual thinning at Darwin despite the topography at Discovery Deep, it is a bit of a push to hint that the results from this study support the hypothesis that Byrd Glacier is of fundamental importance to the stability of ice in the Ross Sea sector.

Line 818-819: As mentioned previously, the reported evidence does not convincingly indicate that ice could not have been thicker at the LGM. This conclusion may not be wrong, but it is currently not well-supported.

Figure 1: Label the Ross Sea, McMurdo Sounds and Mackay Glacier, which are mentioned in the text.

Figures 3, 5, 7: It would be worth stating that these points represent the mean ages with total uncertainty at 1 standard deviation. For clarity, perhaps use a different colour for the horizontal blue line; some readers may automatically associate this line with the Britannia I limit in the maps. It is difficult to understand how these ages relate to the samples and deposit limits in the maps. It would help if you showed the elevation of the Britannia I limit in these plots.

Figure 9: The 14C-saturated sample could probably be better represented in the figure.

Is it trying to say that the apparent age is from 11.5 kyr to infinity?

Figure 13: Can the figure be annotated to highlight the main interpretations from these results?

Supplemental Data Table: Units are not included for all columns, on all sheets. The scaling scheme and production rates used for the ages should be clearly stated.

---

## Editor Comment (EC1) · Chris R. Stokes (Editor) · 12 Feb 2021

I would like to thank both reviewers for their thoughtful and constructive reviews. It is clear that both would like to see this paper published, but they also raise a number of potentially important issues, which I would encourage the authors to consider.
* * *

---

## Author Comment (AC1) · 12 Mar 2021

Response to reviewers' comments on "Holocene thinning and grounding-line retreat of Darwin and Hatherton Glaciers, Antarctica" by Trevor R. Hillebrand, John O. Stone, Michelle Koutnik, Courtney King, Howard Conway, Brenda Hall, Keir Nichols, Brent Goehring, and Mette K. Gillespie

We thank both reviewers for their insightful and extremely thorough reviews. Reviewer text is in black, with author responses in blue. We see the reviewers' main comments as falling into three categories, which we will be happy to address:

1) Insufficient motivation is given for the work. Questions and hypotheses should be stated clearly in the Introduction.
2) The glacial geologic constraints (independent of the exposure ages) need to be better explained.
3) The flowband and ice-sheet modeling sections need a more thorough and clear explanation of our approach and motivation, as well as additional information about the limitations of such an approach.

The reviewers have also made excellent recommendations for how to improve portions of the text and several figures, and have pointed out some references that should be added to the paper.

**Response to Dr. Lauren Simkins (Reviewer 1)**
Review summary: This study integrates cosmogenic exposure ages and numerical modeling to investigate the history of Darwin and Hatherton glaciers. The major conclusion is that the two glaciers are in-sync and experienced 500 m of thinning between 9,000 and 3,000 year before present. While the geochronology is convincing enough to support the major conclusion, the addition of some of the model brings novel insight to SMB and ice flux that sheds light on the geochronology. However, the flowline modeling work beyond experiment 1 is not supported by the geochronology (which is the backbone of this study) and, in my mind, unnecessary.
Thank you for this comment. While we disagree with the assessment that the flowband modeling beyond experiment 1 is not supported by the geochronology, this comment shows that we need to do a better job of justifying and explaining our modeling approach at the beginning of the modeling section.

We view Experiment 3 as essential. The geological constraints at Diamond Hill (near the modern grounding line) present two very different possible deglaciation scenarios, which are examined in Experiments 1 and 3. The choice of LGM ice thickness in Experiment 3 is not arbitrary, but is chosen to match the high-elevation, bedrock $^{14}$C ages that seem to be in conflict with the lower elevation ages from both bedrock and erratics. Note how the black curve in the Diamond Hill panel of Fig 12 goes through three bedrock $^{14}$C ages 500–1000 m above modern elevation.

Experiment 2 deals with the somewhat surprising outcome that there is no record of rapid thinning at Darwin and Hatherton Glaciers (with the exception of the high elevation $^{14}$C samples, which our modeling shows to be unlikely to reflect actual fluctuations of Darwin Glacier). Rapid thinning is recorded at other glaciers along the TAM to the north and south of Darwin-Hatherton. So the question is whether a period of rapid thinning could lie unrecorded in the data gap prior to 7 kyr BP. Our modeling shows that this is inconsistent with the data from Hatherton Glacier, and thus it is most likely that there was not a period of rapid thinning at these glaciers. While this experiment is not essential to explaining our data, we view it as a very interesting and unexpected outcome of this project and would prefer to leave it in the manuscript. We believe that leaving this experiment out of the manuscript would leave a very interesting and potentially important feature of the Darwin-Hatherton chronology undiscussed.

We will add a paragraph to the beginning of Section 3 that gives an overview to help guide readers through the modeling section. This will help explain our modeling approach and the motivation for each experiment more clearly.

Very little motivation and broader significance is presented, leaving the reader either not convinced of the importance of the study or having to do extra work to make those leaps on their own. In both the abstract and introduction, the problem, questions, and broader significance are completely lacking. I urge the authors to explicitly address "why is this study needed/timely/significant?" and "what can we learn from small catchments of the TAM during/since the LGM?" The text is dominated by results,some of which seem unnecessary to present which end up shrouding the important results/interpretations. Careful consideration is needed regarding what results and methodological details to divulge and to what degree they are integral to the story. The text would benefit greatly from revision of the introduction, results, and conclusion.

We agree with the assessment that the motivation is under-stated as currently written. We will expand the beginning of the introduction to better state the motivation and the questions addressed in the paper. A summary of this section is as follows:
The radiocarbon chronology of Bockheim et al. (1989) from Hatherton Glacier has been used as a key constraint on grounding-line retreat in the western Ross Sea for more than three decades. King et al. (2020) showed that significant thinning continued for several thousand years after the Bockheim et al. (1989) chronology, calling the accuracy of the previous inferred age of grounding-line arrival (>6.8 kyr) into question. Darwin and Hatherton glaciers thus provide the only high-resolution constraint on LGM-to-present ice thickness between the Minna Bluff-to-McMurdo Sound region and Beardmore Glacier, a span of around 600 km. However, these data were collected alongside the tributary Hatherton Glacier 50 km from the modern grounding line of Darwin Glacier; it is not clear precisely how this chronology constrains the

timing of grounding-line arrival. So this leaves unanswered a first-order question: When did the grounding line reach its modern position at Darwin Glacier? In order to understand the LGM-to-present history of ice thickness fluctuations in this large data gap, we present new chronologies from two additional sites at Hatherton Glacier (Dubris-Bibra Valleys and Magnis Valley), and one site near the grounding line of Darwin Glacier at Diamond Hill.

In summary, the study has significance for understanding the history of Darwin and Hatherton glaciers. I, however, think the impact of this work is underdeveloped as it is currently written. Explicit hypotheses or questions would go a long way in boosting the importance of the study to better understand (a)synchroneity of neighboring glacial systems and relatively small glacial systems of the TAM and EAIS in general.

Thank you for this assessment. We will improve the motivation as discussed in the previous comment. We also agree that this study could hold important information for work done (or planned) at other small glacier systems, and will add a paragraph to the discussion on this topic. We cannot say whether this glacier system is especially representative of small glacier systems in the TAM and EAIS as a whole, but we can use our results to show that chronologies from these systems may be heavily influenced by larger neighboring glaciers, both through catchment boundary migration and downstream ice dynamics.

Line comments
Abstract
16: explicitly state what is the Ross Sea Ice Sheet
Rephrase to grounded ice sheet in the Ross Sea of Antarctica

20: as an example of removing unnecessary words, the phrase "of Darwin and Hatherton glaciers" is not needed, rather implied
Removed

21-22: remove "Our modeling shoes that" and start with "The constraints"
Removed

23: replace semicolon with period and capitalize "rapid"
Replaced

29: change to "...convergent ice flow from..."
Darwin Glacier flowlines converge with Byrd and Mulock flowlines; therefore, the sentence is correct as is.

Throughout text when using a series of proper nouns with a shared word, like in reference to more than one drift deposit,valley, or glacier, the correct grammar is to not capitalize the shared word; therefore,you should write e.g., "Darwin and Hatherton glaciers".
Replaced

Introduction
Prior to section 1.1, what was done is mentioned but not why the work was done; therefore, the text in its current state will likely not engage a broader audience who works outside of the Ross Sea/Transantarctic Mountain region.
Some of the text from sections 1.2 and 1.3 - sections that would benefit from being more concise -could be reworked and moved up to set the stage for this work. Otherwise, as written,there is a considerable amount of text to read before any motivation/significance for the study can be gleaned.
We will enhance this section to state motivation and questions, using appropriate revised text from sections 1.2 and 1.3.

35: remove "In this paper," and capitalize "we"
Removed

36: first use of "D" in main text; therefore, rephrase to 1.5-dimensional (D)
Rephrased

41: an updated chronology for the offshore record in the context of terrestrial chronology is provided in Prothro et al. (2020) in QSR that would be interesting to consider in this study, which suggest major reorganization of the EAIS grounded in the Ross Sea during the Holocene.
We will add this citation and include in the introduction and discussion.

110: replace last glaciation with "LGM"
We think the term "last glaciation" is more accurate since this sentence deals with transient changes before and after the LGM.

111: modify to read "...thickening grounded ice..."
Modified

146: exhumation by what process?
This is not specified by Storey et al. (2010). We would interpret the process to be either the ablation of ice within the deposit or motion due to ice-wedge polygons, but would prefer not to speculate as to the original authors' interpretation.

150: specify by what magnitude of time is considered "prolonged" - hundreds to thousands of years?
Yes, this has been added

157-160: asan example of making the text more concise throughout the manuscript, here is the original text: King et al. (2020) were able to date the advance and retreat of HathertonGlacier in the Lake Wellman valley using radiocarbon dating of freeze-dried algae.They showed that the glacier advanced to the Britannia-I limit at 9.5 kyr BP, and argued that both the majority of their own exposure ages and those of 160 Storey et al. (2010) were overestimates due to prior exposure of the samples. And here is what I would suggest: Based on radiocarbon dating of algae in Lake Wellman, Hatherton Glacieradvanced to the Britannia-I limit at 9.5 kyr BP suggesting exposure ages in the region were overestimates due inheritance from prior exposures (King et al., 2020).
We will take this specific revision and attempt to make the text more concise throughout.

169: Still by this point in the text, motivation, significance, questions, and/or hypotheses are not explicitly addressed. Also it would be useful at some point to address how the study is organized: first present the chronology/thickness changes and interpretations and then modeling to help explain the geological observations.
See previous response about enhancing motivation at the top of the Introduction.
This paragraph is essentially a statement of how the paper is organized, but we will make that more explicit.

Records of glacier fluctuations
I suggest having one subsection for Dubris, Bidra, andMagnis valleys and Lake Wellman and another for Brown Hills, Diamond Glacier, andDiamond Hill, followed by the summary section
This is a good suggestion. We will make that change.

203: Goehring et al. (2019a) I presume
Correct; amended

203-208: why mention the failed preparation technique in detail rather than simply referring to the Nichols and Goehring (2019) manuscript? That is six lines of text that could be half a line to one line of text.
We believe it is necessary to be forthcoming about the failure of our initial set of analyses to avoid sweeping bad results under the rug. Most of these six lines of text are dealing with the technique that was used for both sets of analyses. However, we will make this text more clear and concise.

221-222: move up to preceding paragraph

Done

247: remove "if the ages at the...of prior exposure,"
We think this is an important point to make, as the spread in these ages is larger than found along the drift limits at the other sites. However, we agree that this is poorly worded and will rephrase.

253: remove "Based on...along Hatherton Glacier". Modeling results have not been presented yet, and I presume you went into the model with the indication that you could use the chronology from the sites together representing a single system.
This sentence will be changed to: "We infer that Hatherton Glacier was close to maximum thickness from at least ~14 kyr BP until ~8 kyr BP, although there is considerable variation in timing between sites."

276: remove "()"
These empty parentheses should have contained a reference to Figures 6 and 7. Those will be added.

305-307: Considering modern ice-flow velocities that suggest flow solely due to internal deformation, paleo-evidence of warm-based ice indicates a major change in ice-bed coupling and the potential for major changes in magnitudes of ice flow velocity. I think this is an interesting point that is not taken any further, but could be followed up on in either/both the modeling or/and discussion section.
Modern flow velocities are ~100 m/yr in this area, which very likely indicates that the lower region of Darwin Glacier is wet-based today, so this observation is unlikely to reflect a major change in ice-flow regime. Therefore, our tuning of the basal sliding and flow enhancement factors to match modern velocities already takes wet-based conditions into account in this area.

403-409: suggest removing all text starting with "we prefer..." to the end of the paragraph
We believe this text is necessary both to underline the equivocal nature of the in situ [14]C results and to explain how we proceed with the model to help elucidate which of these interpretations is more likely.

Numerical modelling of glacier fluctuations
Both "modelling" and "modeling" used in the text .
Will change all instances to "modelling"

Combine sections 3.1-3.3 and make more concise, particularly information insection 3.2 should be in section 3.1 to explain constraints on variables.
These sections will be combined.

432: removefirst sentence
Removed

443: rephrase to "...grid point (c.fl., Golledge et al., 2014)."
Rephrased

444: curious what range of basal sliding is used here that is considered plausible
We vary the basal sliding coefficient $f_s$ between $10^{-12}$ and $10^{-10}$ $Pa^{-3}\,yr^{-1}$ —consistent with the values used by Anderson et al. (2004)—and the flow enhancement factor between 0.1 and 30. We will add these values to the text.

447-450: re phrase to "While we have limited information available to constrain the values of the sliding and deformation, inferred patterns of these parameters generate a surface elevation and velocity profiles that match modern values within their uncertainties."
Rephrased

450-451:this sort of statement would be best at the beginning of section 3.1 to explain why the modeling work was conducted.
The analogous statement exists in Lines 404–405. We can move these to the top of section 3

461: I would argue that you are not investigating the role of catchment geometry, rather catchment flow change which could result from a number of glaciological changes.
We model the glaciers to the top of their current respective catchments, and so adding flux to the upstream boundary necessarily means a larger catchment. We recognize that this has not been stated clearly and will revise accordingly.

Section 3.3 can be much shorter and, like mentioned above, integrated into a combined section with 3.1 and 3.2
It is not clear how to make this section shorter, as it deals with important details of how we compare the data to the model. We could instead summarize it in the main text and move the details to the supplement.

473: are you estimating2-sigma uncertainty? Or calculating it? Not clear what is being done to geologically constrain centerline positions for the model…
We have removed the "2-sigma" designation, as these are really minimum and maximum estimates, and we use the mean for lack of other constraints.

480: "Figures 10-12."
Replaced

481-486: seems out of place

These lines are necessary, as they describe how ice-marginal samples are treated with respect to glacier centerline elevation for Diamond Hill. The previous paragraph does this for the Hatherton Glacier sites.

For the transient experiments, I would argue that experiments 2 and 3 are distracting and not useful in the scheme of the work presented here. It is interesting to note in the discussion that these small glacial systems show no geological evidence for rapid thinning. Experiment 3 could be retained if authors feel that can be justified given lack of constraints and seemingly arbitrarily chosen LGM ice thickness. I am, however,a proponent of the presentation of negative results, but feel like these extra steps in the modeling are not supported or justified by the geological observations and thus do thestudy a disservice by distracting readers with unnecessary negative findings that do the opposite of what the modeling work is supposed to do - which I interpret as aiming to help explain the geological observations.

See our reply at the beginning of this document for why we need to keep experiments 2 and 3.

Ice sheet model ensemble I am largely on board with this work, but think the text could be much more concise (apologies for repeating that point).

Will attempt to make this section more concise.

Discussion is fine; however, broader significance for other small glacial systems in theTAM and beyond could be integrated here.

It is not clear to us that Darwin and Hatherton glaciers are especially representative of small glacier systems in the TAM. However, the possibility that Byrd and Mulock glaciers pirated the catchments of Darwin and Hatherton could be useful as a warning to those planning fieldwork on other glaciers of similar size. We will include a statement that the potential for large relative changes in catchment size should be considered when planning fieldwork or interpreting chronologies.

Conclusion would benefit greatly from not being written as a list. I suggest that authors have a go at linking the findings and major outcomes between the geochronology and the modeling work. As currently written, the reader has to do a lot of work to piecethe listed bits together and come away from this text with take-aways that will invoke further thought, questions, and applications.

We appreciate the feedback and will try to make these connections clearer. However, we believe that the conclusions already link our chronology and modeling work. The list starts with what we can glean from the chronology alone (first two bullet points); states the inconsistency in the data that necessitates the modeling (third bullet point); describes how the flowband modeling resolves this inconsistency (fourth bullet point); describes the use of the ice sheet model ensemble to attempt to put the Darwin-Hatherton results in a regional context (fifth point); and finally poses a hypothesis for why our combined geochronology and model results are different from other

TAM glaciers. We will attempt to make these text more clear and tie the conclusions back to the original motivation for the work more explicitly. We had thought that writing this as a list helped clarify the flow, but perhaps paragraph form is needed to better outline the connections, as you suggest.

Figures The figures look really nice. Figure 1: the bathymetry colormap you have used is not categorized, like the legend suggests.
This will be changed

Include Bidra Valley
Added

Suggest combining figures 2 and 3, figures 4 and 5, and figures 6 and 7.
We would be happy to do so, but are concerned that the map plus the multi-panel chronology figures might be difficult to fit on the page. We will defer to the editor for this.

Make sure consistent reference to figure panels, e.g. figure 3 caption that uses "(a)" and "panel c"
Text changed to be consistent.

Supplementary documents Units for ages in the data table are needed.
Units added

**Response to Dr. Richard Selwyn Jones (reviewer 2)**
The paper presents an interesting combination of geochronological data ($10Be$, $26Al$ and $14C$ ages), flowband modelling and 3D ice sheet modelling, focused on Darwin andHatherton Glaciers. The authors conclude that Darwin Glacier thinned gradually by 500m during the Holocene, in contrast to other glaciers in the Transantarctic Mountains,and that this behaviour was possibly due to the convergent ice flow and lateral drag in the vicinity of Byrd Glacier.

A lot of work is covered in this paper. The manuscript is generally well structured,written and illustrated, the methods seem sound, and the conclusions are mostly sup-ported by the results. The study provides new high-quality data for this region, a new understanding about the ice dynamics at Darwin and Hatherton Glaciers, and ideas related to deglaciation of the Ross Sea.

I recommend publication in The Cryosphere following some revision.

There are a few aspects where I think the paper could be improved. First, the rationale for the study needs to be better communicated, particularly in the Abstract andalso Introduction. Why should the reader care?

Thank you for the feedback, and we agree. See response to reviewer 1 on this same issue.

Second, it is not always clear what is new and what is "borrowed" from previous studies. This concerns the geochronological constraints and interpretations at Lake Wellman, and the ice sheet model ensemble.

We do state that the Lake Wellman chronology is from King et al. (2020), and that the ice sheet model ensemble is our own work that builds on the Pollard et al. (2016) ensemble. However, it would be easy to make these distinctions more obvious to readers, and we will do so.

Third, the maximum ice limits at 9.5 kyr BP are not clearly supported by this study.There are no saturated in situ 14C samples from Magnis Valley and Bibra/Dubris Valleys, and the interpretation of Lake Wellman is based on another study, with limited description of why older samples are ignored.

We disagree here. We date well-defined depositional limits in the valleys alongside Hatherton Glacier. It is clear that there was no significant ice advance beyond these limits during the last glaciation because of the sharply delimited boundaries between deposits. Therefore, *in situ* $^{14}$C are not needed for the Hatherton Glacier sites, where we can rely on glacial geology alone to demarcate the boundary of the last glacial advance into these valleys. At Diamond Hill, there is no clearly defined limit of deposition, and so we rely on *in situ* $^{14}$C ages of bedrock and our flowband model to interpret the ice thickness history here. We take this comment to mean that our description of our glacial geologic mapping is lacking, and we will add details and references to make it clear how we determined maximum ice thickness independent of the exposure ages. We will also include photos of these depositional limits in the supplementary material.

Fourth, the flowband modelling skips over some limitations in this approach, while the ice sheet model ensemble is short on detail. More details on these points are included below.

We will add a discussion of the limitations of the flowband model, and more details for the ice sheet model ensemble. See our responses to the more detailed comments below. Thank you for these insightful and helpful comments.

Detailed comments
Title: 'grounding-line retreat' is included, yet the paper concludes that it is not possible to gauge grounding-line retreat from the thinning data, and the modelling does not include grounding-line retreat of this system. Consider removing.

This is a good point. However, Darwin and Hatherton glaciers are important precisely because they have been used for several decades as key constraints on grounding-line retreat in the Ross Sea. Our work shows that the ice thickness history at the mouth of Darwin Glacier is not easily interpretable in terms of rates of grounding-line retreat, but we also conclude that interpreting the timing of grounding-line arrival from the end of the thinning history is still valid. So, while we agree that the title can be clarified, we would like to retain a mention of the grounding line in the title. We suggest the new title: Holocene thinning of Darwin and Hatherton glaciers, Antarctica and implications for grounding-line retreat in the Ross Sea

Line 17, and elsewhere: The Early and Mid/Middle Holocene are now formalised sub-epochs, so should be capitalised.
Changed

Line 42, and elsewhere: Grounded ice in the Ross Sea is not a true self-contained ice sheet. Rather it is a sector of the Antarctic ice sheet, or a region comprising ice from the East and West Antarctic ice sheets. Consider rephrasing.
 Changed to "grounded ice in the Ross Sea"

Line 62: Dramatic thinning is also now recorded at Mawson Glacier (Jones et al.,Geology, 2020) and David Glacier (Stutz et al., TCD, in review).
References added

Lines 68-69: Can you describe Darwin and Hatherton Glaciers in relation to the glaciers mentioned above?
Will add description of location, size, and flow regime relative to Scott, Reedy, Shackleton, Beardmore, Mackay, and David.

Lines 99-104: It might also be worth mentioning the model buttressing studies of Furstet al. (2016) and Reese et al. (2018), which show that this region has a large buttress-ing effect on the ice sheet (albeit based on the current ice configuration).
Added

Lines 156-163: Slightly confusing timeline of research. It would probably be better to present in publishing order (Storey et al., 2010, Joy et al., 2014, King et al., 2020).
Reordered

Lines 167-169: This last point seems a bit lost. Consider making it a third goal.
Added (iii)

Line 172 ("erratics"): Technically, if they are erratics then their lithology should differ to the local geology; how they are 'erratic' should be described here. It would also be worth providing a broad description of the erratic size - cobbles, boulders?

Changed to glacially-transported cobbles and boulders

Line 172 ("Diamond Hill, as well as from Bibra, Dubris and Magnis Valleys"): It would be useful to state here how far each of these sites are from the modern grounding line.

Distances added

Line 173: "freshest" should be made clearer. Rocks that appeared least weathered?

Changed to "least weathered"

Lines 174-180: As previous studies in the area (e.g. Storey et al., 2010; Joy et al.,2014) have highlighted the importance of sampling strategy, particularly regarding re-working by cold-based ice, it would be useful to point out how your approach is simi-lar/different.

We will enhance our description of the glacial geologic mapping and sampling strategies, as this will also address the previous point about how we define LGM ice thickness independent of exposure ages.

Lines 193-195: It would be useful here to clarify what is detectable given uncertainties in erosion rates and measurements.

We already have the text: "a few thousand years of ice cover during the LGM will create a detectable signal of burial". Sub-aerial erosion rates in Antarctica are negligible over the time span in question, whereas subglacial erosion will only bring the apparent exposure age close to a true age by removing nuclides inherited from exposure prior to being ice-covered.

Lines 218-219: It would be useful to quantify the impact of the scaling scheme on the exposure ages - what percent difference between schemes (e.g. Stone (2000) and Lifton et al. (2014) nuclide-specific scheme)? Also, what production rate dataset is used? This should be described here.

Reference to Borchers et al. (2016) production rate dataset added. We will quantify the difference between the most frequently used schemes, which is on the order of a few percent for most samples.

Lines 250-255: It is not clear how the maximum thickness is determined to be from at least 14 kyr. No potentially older samples were collected from higher elevations, and there are no in situ 14C measurements from here to test for saturation.

This will be clarified by text added to address two previous comments regarding the glacial geology. We sampled at the limit of the Britannia I deposit, which is very well defined. The deposit beyond this limit (Britannia II) was shown by Joy et al. (2014) to be much older (~130

kyr). in situ $^{14}$C measurements are not necessary in locations like these, where clearly defined limits of deposits demarcate positions of maximum advance. We do not feel that additional text is necessary to support our assertion here.

Lines 275-278: If no new ages from Lake Wellman are presented in this study, then this needs to be stated clearly here, indicating that data from previous studies are used to assist interpretation of ice history in the area. You also need to explain in more detail why the higher elevation erratic ages at 20 kyr are "pre-exposed" and therefore ignored.
We do state this in the figure captions, but will reiterate in the main text. We will add a short summary of the discussion of the 20 kyr samples from King et al. (2020). Briefly stated, the Lake Wellman area has a great abundance of pre-exposed erratics (see Fig 7b and the widely scattered ages in Story et al., 2010). The 20 kyr ages are inconsistent with the dense algae radiocarbon dataset and the ages of depositional limits elsewhere at Hatherton Glacier, and are therefore considered outliers.

Line 394: The text in this section does not convincingly support the claim that ice wasat its maximum at 9.5 kyr. There are no saturated in situ 14C samples from these sites,and there is limited justification of why older samples are ignored. Clarify and expand on this point in the above text, or revise this claim.
See point above with regards to lines 250–255 and to the reviewer's third point in the summary. These ages are from well-defined depositional limits in many cases, or are the highest-elevation samples that exhibit the characteristics of the Britannia I deposit. The contrast between the Britannia I and Britannia II deposits is stark, as the Britannia II deposit is ~115 kyr older than Britannia I. We will make this point more explicit, including references to previous mapping in the area, further description of our glacial geology method, and photos of deposits in the supplement.

Line 396: You should probably include a study that directly dates grounding-line retreat in the Ross Embayment (e.g. McKay et al., 2016; Prothro et al., 2020).
These references and the dates therein will be added.

Line 414: Jones et al. (2016) is not included in the references.
Sorry about that. This reference has been added.

Lines 416-417: The model flowband/domain needs to be shown somewhere, perhaps in Figure 1c.
Model centerline added to Fig 1c

Lines 420-423: While this point is largely true, the flowband is a very simple model and so the limitations should be stated. What key stresses are not calculated? What aspects of the deglacial history can versus cannot be tested?

We will add a discussion of the limitations, namely that:

1) Our shallow-ice model assumes that vertical shear stresses are dominant and does not account for lateral and longitudinal stresses. Our model also does not account for the transition from grounded to floating ice and this restricts applying the model to locations that are always upstream of the modern grounding line,, which is one reason we cut off the model domain 10 km upstream of the modern grounding line. Therefore, the model will become less accurate as the grounding line gets close to its modern position, but that is only likely to be the case towards the end of the model runs.

2) The goals of this investigation and the lack of information about boundary conditions in the past are the primary reasons we implemented such a simple model, but this means that we cannot distinguish between different causes of thinning at the mouth, e.g., grounding-line retreat or change in buttressing. We saw this as a separate goal, which could be investigated in future work and if specific hypotheses could be tested and an appropriate model and boundary conditions could be applied.

3) Our tuned basal sliding and enhancement factor fields do not change in time because we have no constraints on how these values would change over time. However, this means that we might be missing effects of changes in bed properties or ice temperature that influence the rate of sliding and ice deformation over time.

4) While we can add or subtract flux entering at the upstream end of the flowline model domain to represent changes in the upstream catchment area, that is a simple representation of how large-scale changes in glacier geometry may influence glacier evolution. Also a simplification is that, we only account for glacier tributary contribution to the domain as an addition of mass flux to the values along our primary glacier centerline.

Lines 423-430: As far as I can see, this model does not account for lateral drag, which the conclusions indicate is potentially very important. How might this impact the flow-band results?

The lateral drag that we indicate could be important is far outside the flowband model domain at Minna Bluff and Cape Crozier. But, we appreciate the question because the lower portions of many TAM outlets become more laterally constrained (as you pointed out in Jones et al., 2016 and was important for Skelton Glacier). The importance of lateral drag depends on the cross-sectional geometry of a glacier (how much friction from valley walls) and can be significant if the glacier is narrow, if the flow is channelized, or the ice-flow speeds are high. While we seek to apply the simplest flowline model that is physically meaningful, there has been work to develop parameterizations for higher-order physics that are otherwise ignored in shallow-ice flowline models like we apply here. In the case of lateral drag, Adhikari and Marshall (2012) expanded on previous work to provide a set of correction factors for lateral drag that is due to either glacier geometry or due to variations in glacier slip. So, while we could apply

this parameterization as a correction factor (0.1 to 1) to modify our value of gravitational driving stress, this work showed that lateral drag was most critical for fast flowing ice and if the glacier was narrow or channelized. Given this, and how we apply the flowline model in this work there are two main reasons we chose to not include a lateral-drag factor: 1) We treat the two parameter values already used in our model (the deformation factor and the sliding factor) as tuning parameters. We do not interpret the value of these factors, and so adding an additional factor will only offset the inferred (tuned) value and not offer more physical information. In addition, since we have no constraints on how any of these factors may have changed in time we use the values inferred from matching to modern conditions. 2) Most of the ice in our domain is relatively slow flowing. It is possible that the lower portion of Darwin is flowing fast enough for lateral drag to be more significant, but that would be taken up in our inferred values for the deformation factor. Again, since these values are tuning parameters any actual influence of lateral drag on the modern flow field would be reflected in the inferred values of the tuning parameters.

Line 428: How many grid points? What is the horizontal resolution used?
1km resolution and number of grid points added (75 for Hatherton, 141 for Darwin)

Line 438: How is SMB scaled using the scaling factors for LGM scenarios?
We have added the clarification that scaling is spatially uniform, and added reference to table 1.

Lines 452-453: To what extent does modification of the downstream boundary represent perturbations (e.g. reduce buttressing, grounding-line retreat) downstream of this point? As mentioned above, you should make clear what processes are and are not accounted for, and what can be tested.
When the flowline model is introduced we will make it more clear what processes are accounted for and why certain simplifications were made. The downstream boundary is an elevation value that represents how that point on the lower glacier changed over time in response to ice-shelf buttressing and glacier evolution as the grounding line retreated. However, we are not able to interpret the distribution of processes (i.e., change in ice-shelf buttressing vs change in grounding-line position) that led to our inferred elevation history. Our model domain always begins upstream of the grounding line and the deformation and sliding factors in our model do not change over time since we have no constraints on how they would evolve. While the model is simplified, we feel that it is still appropriate for this application. We appreciate that this point was raised and will add more explanation about the flowline model assumptions.

Line 468, and elsewhere: What is meant by 'LGM' in the context of these sites? The proposed local ice maximum? The Antarctic or global glacial maximum?
We mean the local ice maximum, which is much later than the global or even Antarctic maxima. The usage is frequent enough that it would be ungainly to specify local maximum everywhere in the text, but we will note this important distinction with the first usage of the term.

Lines 597-599: This interpretation should be in Section 2.

We will add this in 2.2.6 (Summary of geochronologic constraints)

Lines 646-648: Is this a new set of model simulations, and not output from previously published simulations? This would be a considerable effort and requires more descrip-tion here. No need to describe all the of the model physics, but state that this can be found in a different paper. What ocean and atmosphere forcings are applied? What are the values of other parameters that were not examined? Some readers would expect to see these details, even if described in the Supplementary Material.

This is our original work, presented here for the first time. We will add more description of the ensemble to the Supplementary Material.

Lines 712-714: How might the model resolution impact this finding?

While very high resolution (≤1 km; unfeasible for these simulations) is required to very accurately model grounding-line retreat, the response of grounded ice to grounding-line retreat is less resolution dependent. So, the 10 km resolution probably leads to inaccuracies in the grounding-line migration, but the response of grounded ice should be reasonably well captured. We will add a statement to this effect.

Line 722: McMurdo Sound deglaciated in the Mid Holocene, so "Early to Mid Holocene" is probably more correct.

Amended

Lines 728-279: I agree that more data are required from the lower reaches of Skelton Glacier. But, to be picky, the erratic age at 6 kyr is <20 km from the modern grounding line, which is not "far" considering the scale of these glaciers. Consider rephrasing.

Good point. Rephrased to "A small number of samples ~16–50 km from the modern grounding line"

Line 735: In what context is there no record of "an exceptionally fast period of thin-ning"? How is this quantified? Is this relative to modern observations or other thinning estimates from TAM?

We mean this in contrast to, for example, the 200 m of thinning in <2000 years at Mackay Glacier or the similar rapid thinning at Beardmore. We don't define a threshold for what counts as "exceptionally fast", but by contrast to records from some other glaciers these records show remarkably steady thinning rates. We will clarify this in the text.

Lines 748-749: As most thinning at Mackay Glacier to the north occurred at 7.5-6.5kyr, McMurdo Sound became free of ground ice between 6.5 (not 7.5) and 9 kyr BP.

Amended

Line 755: Also the case with Mawson Glacier and David Glacier (see references above)
Added

Lines 773-777: The argument regarding glacial isostatic adjustment could be clearer.How greater isostatic depression at Minna Bluff could resolve the elevation differences between Darwin Glacier and Minna Bluff needs a little more explanation. Is this just considering the relative lowering of the LGM elevations at Minna Bluff, or also the im-pact on the age constraints? It is possible to estimate the relative elevation difference?
This is just accounting for relative lowering between the two sites, to attempt to make the ice surface at the mouth of Darwin higher than at Minna Bluff. This would need to be the case regardless of age constraints in order to fit these previous reconstructions. We will attempt to estimate the reasonable range of isostatic adjustment at each location, although the complex LGM ice geometry around Darwin Glacier might make this estimate highly uncertain and overly simple.

Lines 796-797: This was done for the Ross Embayment alone in Lowry et al. (2019,2020), yet still some large differences in the timing of thinning between simulations and geological data.
Text will be revised accordingly.

Lines 805-809: While this is a useful discussion with the suggestion that the confluence of Byrd-Hatherton-Darwin Glaciers may explain gradual thinning at Darwin despite the topography at Discovery Deep, it is a bit of a push to hint that the results from this study support the hypothesis that Byrd Glacier is of fundamental importance to the stability of ice in the Ross Sea sector.
Thank you, and we agree. This statement will be removed.

Line 818-819: As mentioned previously, the reported evidence does not convincingly indicate that ice could not have been thicker at the LGM. This conclusion may not be wrong, but it is currently not well-supported.
We disagree, for the reasons explained above. To reiterate briefly, we dated well-defined depositional limits along Hatherton Glacier. There is no reason to suspect that ice advanced significantly beyond these limits, as there is no mechanism to explain a sudden shift from no deposition to abundant deposition.

Figure 1: Label the Ross Sea, McMurdo Sounds and Mackay Glacier, which are men-tioned in the text.
Labels added

Figures 3, 5, 7: It would be worth stating that these points represent the mean ageswith total uncertainty at 1 standard deviation. For clarity, perhaps use a different colour for the horizontal blue line; some readers may automatically associate this line with theBritannia I limit in the maps. It is difficult to understand how these ages relate to the samples and deposit limits in the maps. It would help if you showed the elevation of the Britannia I limit in these plots.
Horizontal blue line color changed. Britannia I limit elevations added.

Figure 9: The 14C-saturated sample could probably be better represented in the figure. Is it trying to say that the apparent age is from 11.5 kyr to infinity?
Good point. This sample is $^{14}$C saturated, but uncertainties in analysis, production rate, and scaling factor admit the possibility of a much younger age of 11.5 kyr. We will add an arrow pointing right, and explain this in the caption.

Figure 13: Can the figure be annotated to highlight the main interpretations from these results?
Yes. It will be difficult to do without cluttering up the figure, but we could add arrows of different colors and then refer to them in the caption. The main point is that, while there are some broad similarities in structure between rates of thickness change at the mouth of Darwin and the rate of grounding-line migration, there are periods of rapid grounding-line retreat that correspond to periods of slow thinning, and vice versa. Therefore, we should not attempt to interpret our thinning history in terms of grounding-line retreat, only in terms of the timing of grounding-line arrival.

Supplemental Data Table: Units are not included for all columns, on all sheets. The scaling scheme and production rates used for the ages should be clearly stated.
This information has been added.

References
Adhikari S and Marshall SJ (2012) Parameterization of lateral drag in flowline models of glacier dynamics. *Journal of Glaciology* **58**(212), 1119–1132. doi:10.3189/2012JoG12J018.
Anderson BM, Hindmarsh RCA and Lawson WJ (2004) A modelling study of the response of Hatherton Glacier to Ross Ice Sheet grounding line retreat. *Global and Planetary Change* **42**(1–4), 143–153. doi:10.1016/j.gloplacha.2003.11.006.
Bockheim JG, Wilson SC, Denton GH, Andersen BG and Stuiver M (1989) Late Quaternary ice-surface fluctuations of Hatherton Glacier, Transantarctic Mountains. *Quaternary Research* **31**(2), 229–254.

**Borchers B and others** (2016) Geological calibration of spallation production rates in the CRONUS-Earth project. *Quaternary Geochronology* **31**, 188–198. doi:10.1016/j.quageo.2015.01.009.

**Jones RS, Golledge NR, Mackintosh AN and Norton KP** (2016) Past and present dynamics of Skelton Glacier, Transantarctic Mountains. *Antarctic Science* **28**(05), 371–386. doi:10.1017/S0954102016000195.

**Joy K, Fink D, Storey B and Atkins C** (2014) A 2 million year glacial chronology of the Hatherton Glacier, Antarctica and implications for the size of the East Antarctic Ice Sheet at the Last Glacial Maximum. *Quaternary Science Reviews* **83**, 46–57. doi:10.1016/j.quascirev.2013.10.028.

**King C, Hall B, Hillebrand T and Stone J** (2020) Delayed maximum and recession of an East Antarctic outlet glacier. *Geology* **48**(6), 5.

**Pollard D, Chang W, Haran M, Applegate P and DeConto R** (2016) Large ensemble modeling of the last deglacial retreat of the West Antarctic Ice Sheet: comparison of simple and advanced statistical techniques. *Geosci. Model Dev.* **9**(5), 1697–1723. doi:10.5194/gmd-9-1697-2016.

**Storey BC and others** (2010) Cosmogenic nuclide exposure age constraints on the glacial history of the Lake Wellman area, Darwin Mountains, Antarctica. *Antarctic Science* **22**(06), 603–618. doi:10.1017/S0954102010000799.

---

## Author Response (AR1)

Response to reviewers' comments on "Holocene thinning and grounding-line retreat of Darwin and Hatherton Glaciers, Antarctica" by Trevor R. Hillebrand, John O. Stone, Michelle Koutnik, Courtney King, Howard Conway, Brenda Hall, Keir Nichols, Brent Goehring, and Mette K. Gillespie

We thank both reviewers for their insightful and extremely thorough reviews. Reviewer text is in black, with author responses in blue. Any references to figures, sections, or line numbers in the authors' reply are referring to the revised manuscript with tracked changes, while reviewers' comments refer to lines in the previous version.

We see the reviewers' main comments as falling into three categories:
1) Insufficient motivation is given for the work. Questions and hypotheses should be stated clearly in the Introduction.
2) The glacial geologic constraints (independent of the exposure ages) need to be better explained.
3) The flowband and ice-sheet modeling sections need a more thorough and clear explanation of our approach and motivation, as well as additional information about the limitations of such an approach.

The reviewers have also made excellent recommendations for how to improve portions of the text and several figures, and have pointed out some references that should be added to the paper.

**Response to Dr. Lauren Simkins (Reviewer 1)**
Review summary: This study integrates cosmogenic exposure ages and numerical modeling to investigate the history of Darwin and Hatherton glaciers. The major conclusion is that the two glaciers are in-sync and experienced 500 m of thinning between 9,000 and 3,000 year before present. While the geochronology is convincing enough to support the major conclusion, the addition of some of the model brings novel insight to SMB and ice flux that sheds light on the geochronology. However, the flowline modeling work beyond experiment 1 is not supported by the geochronology (which is the backbone of this study) and, in my mind, unnecessary.
Thank you for this comment. While we disagree with the assessment that the flowband modeling beyond Experiment 1 is not necessary, this comment shows that we need to do a better job of justifying and explaining our modeling approach at the beginning of the modeling section. We have added a paragraph at the beginning of Section 3 that reviews the need for each model experiment.

We view Experiment 3 as essential. The geological constraints at Diamond Hill (near the modern grounding line) present two very different possible deglaciation scenarios, which are examined in Experiments 1 and 3. The choice of LGM ice thickness in Experiment 3 is not arbitrary, but is chosen to match the high-elevation, bedrock $^{14}C$ ages that seem to be in conflict with the lower

elevation ages from both bedrock and erratics. Note how the black curve in the Diamond Hill panel of Fig 8 goes through three bedrock $^{14}$C ages 500–1000 m above modern elevation.

Experiment 2 deals with the somewhat surprising outcome that there is no record of rapid thinning at Darwin and Hatherton Glaciers (with the exception of the high elevation $^{14}$C samples, which our modeling shows to be unlikely to reflect actual fluctuations of Darwin Glacier). Rapid thinning is recorded at other glaciers along the TAM to the north and south of Darwin-Hatherton. So the question is whether a period of rapid thinning could lie unrecorded in the data gap prior to 7 kyr BP. Our modeling shows that this is inconsistent with the data from Hatherton Glacier, and thus it is most likely that there was not a period of rapid thinning at these glaciers. While this experiment is not essential to explaining our data, we view it as a very interesting and unexpected outcome of this project and would prefer to leave it in the manuscript. We believe that leaving this experiment out of the manuscript would leave a very interesting and potentially important feature of the Darwin-Hatherton chronology undiscussed.

Very little motivation and broader significance is presented, leaving the reader either not convinced of the importance of the study or having to do extra work to make those leaps on their own. In both the abstract and introduction, the problem, questions, and broader significance are completely lacking. I urge the authors to explicitly address "why is this study needed/timely/significant?" and "what can we learn from small catchments of the TAM during/since the LGM?" The text is dominated by results,some of which seem unnecessary to present which end up shrouding the important results/interpretations. Careful consideration is needed regarding what results and methodological details to divulge and to what degree they are integral to the story. The text would benefit greatly from revision of the introduction, results, and conclusion.

We agree with the assessment that the motivation is under-stated as currently written. We have rewritten Section 1.1 to better state the motivation and the questions addressed in the paper, and have added a similar statement to the abstract.

In summary, the study has significance for understanding the history of Darwin and Hatherton glaciers. I, however, think the impact of this work is underdeveloped as it is currently written. Explicit hypotheses or questions would go a long way in boosting the importance of the study to better understand (a)synchroneity of neighboring glacial systems and relatively small glacial systems of the TAM and EAIS in general.

Thank you for this assessment. We have improved the motivation as discussed in the previous comment. We also agree that this study could hold important information for work done (or planned) at other small glacier systems, and we have added a paragraph to the Discussion (Starting at line 1356) on this topic.

Line comments
Abstract
16: explicitly state what is the Ross Sea Ice Sheet
Throughout the paper, we have rephrased to "grounded ice sheet in the Ross Sea".

20: as an example of removing unnecessary words, the phrase "of Darwin and Hatherton glaciers" is not needed, rather implied
Removed

21-22: remove "Our modeling shoes that" and start with "The constraints"
Removed

23: replace semicolon with period and capitalize "rapid"
Replaced

29: change to "...convergent ice flow from..."
Darwin Glacier flowlines converge with Byrd and Mulock flowlines; therefore, the sentence is correct as is.

Throughout text when using a series of proper nouns with a shared word, like in reference to more than one drift deposit,valley, or glacier, the correct grammar is to not capitalize the shared word; therefore,you should write e.g., "Darwin and Hatherton glaciers".
Replaced

Introduction
Prior to section 1.1, what was done is mentioned but not why the work was done; therefore, the text in its current state will likely not engage a broader audience who works outside of the Ross Sea/Transantarctic Mountain region.
Some of the text from sections 1.2 and 1.3 - sections that would benefit from being more concise -could be reworked and moved up to set the stage for this work. Otherwise, as written,there is a considerable amount of text to read before any motivation/significance for the study can be gleaned.
As stated above, we have rewritten section 1.1 to better state the motivation and questions being addressed.

35: remove "In this paper," and capitalize "we"
Removed

36: first use of "D" in main text; therefore, rephrase to 1.5-dimensional (D)
Rephrased

41: an updated chronology for the offshore record in the context of terrestrial chronology is provided in Prothro et al. (2020) in QSR that would be interesting to consider in this study, which suggest major reorganization of the EAIS grounded in the Ross Sea during the Holocene.
We have added this reference in Sections 1.1, 2.3,  and 5 section.

110: replace last glaciation with "LGM"
We think the term "last glaciation" is more accurate since this sentence deals with transient changes before and after the LGM.

111: modify to read "...thickening grounded ice..."
Modified

146: exhumation by what process?
This is not specified by Storey et al. (2010). We would interpret the process to be either the ablation of ice within the deposit or motion due to ice-wedge polygons, but would prefer not to speculate as to those authors' interpretation.

150: specify by what magnitude of time is considered "prolonged" - hundreds to thousands of years?
Yes, this has been added (Line 289)

157-160: asan example of making the text more concise throughout the manuscript, here is the original text: King et al. (2020) were able to date the advance and retreat of HathertonGlacier in the Lake Wellman valley using radiocarbon dating of freeze-dried algae.They showed that the glacier advanced to the Britannia-I limit at 9.5 kyr BP, and argued that both the majority of their own exposure ages and those of 160 Storey et al. (2010) were overestimates due to prior exposure of the samples. And here is what I would suggest: Based on radiocarbon dating of algae in Lake Wellman, Hatherton Glacieradvanced to the Britannia-I limit at 9.5 kyr BP suggesting exposure ages in the region were overestimates due inheritance from prior exposures (King et al., 2020).
We have re-written this paragraph to make more concise (starts Line 291), and have attempted to make wording more concise throughout.

169: Still by this point in the text, motivation, significance, questions, and/or hypotheses are not explicitly addressed. Also it would be useful at some point to address how the study is organized: first present the chronology/thickness changes and interpretations and then modeling to help explain the geological observations.
See previous response about enhancing motivation at the top of the Introduction.

This paragraph (Line 301) is essentially a statement of how the paper is organized, but we have made that more explicit.

Records of glacier fluctuations
I suggest having one subsection for Dubris, Bidra, andMagnis valleys and Lake Wellman and another for Brown Hills, Diamond Glacier, andDiamond Hill, followed by the summary section
We have made this change.

203: Goehring et al. (2019a) I presume
Correct; amended

203-208: why mention the failed preparation technique in detail rather than simply referring to the Nichols and Goehring (2019) manuscript? That is six lines of text that could be half a line to one line of text.
We believe it is necessary to be forthcoming about the failure of our initial set of analyses to avoid sweeping bad results under the rug. However, we have revised this text to be more clear and concise (Lines 401–405).

221-222: move up to preceding paragraph
Done

247: remove "if the ages at the...of prior exposure,"
We think this is an important point to make, as the spread in these ages is larger than found along the drift limits at the other sites. However, we agree that this is poorly worded have rephrased (Line 458).

253: remove "Based on...along Hatherton Glacier". Modeling results have not been presented yet, and I presume you went into the model with the indication that you could use the chronology from the sites together representing a single system.
This sentence has been changed to: "We infer that Hatherton Glacier was close to maximum thickness from at least ~14 kyr BP until ~8 kyr BP, although there is considerable variation in timing between sites." (Line 463).

276: remove "()"
These empty parentheses should have contained a reference to Figure 4, which has been added.

305-307: Considering modern ice-flow velocities that suggest flow solely due to internal deformation, paleo-evidence of warm-based ice indicates a major change in ice-bed coupling and the potential for major changes in magnitudes of ice flow velocity. I think this is an interesting

point that is not taken any further, but could be followed up on in either/both the modeling or/and discussion section.

Modern flow velocities are ~100 m/yr in this area, which very likely indicates that the lower region of Darwin Glacier is wet-based today, so this observation is unlikely to reflect a major change in ice-flow regime. Therefore, our tuning of the basal sliding and flow enhancement factors to match modern velocities already takes wet-based conditions into account in this area.

403-409: suggest removing all text starting with "we prefer..." to the end of the paragraph
We believe this text is necessary both to underline the equivocal nature of the in situ $^{14}$C results and to explain how we proceed with the model to help elucidate which of these interpretations is more likely. However, we have revised this text (Lines 760–767) to be more explicit about what we can infer from these chronologies and why we need the flowband model in the next section.

Numerical modelling of glacier fluctuations
Both "modelling" and "modeling" used in the text .
We have changed all instances to "modeling"

Combine sections 3.1-3.3 and make more concise, particularly information insection 3.2 should be in section 3.1 to explain constraints on variables.
These sections have combined into Section 3.1 and extensively revised to be more clear and concise. However, as this combined section is very important and not very long to begin with, we have opted not to shorten it drastically.

432: removefirst sentence
Removed

443: rephrase to "...grid point (c.fl., Golledge et al., 2014)."
Rephrased

444: curious what range of basal sliding is used here that is considered plausible
We vary the basal sliding coefficient $f_s$ between $10^{-12}$ and $10^{-10}$ m$^2$ Pa$^{-3}$ yr$^{-1}$ —consistent with the values used by Anderson et al. (2004)—and the deformation factor from 3.65 x $10^{-18}$ to 1.1 x $10^{-15}$ Pa$^{-3}$ yr$^{-1}$. We have added these values to the text in Line 946.

447-450: re phrase to "While we have limited information available to constrain the values of the sliding and deformation, inferred patterns of these parameters generate a surface elevation and velocity profiles that match modern values within their uncertainties."
Rephrased (Line 861)

450-451:this sort of statement would be best at the beginning of section 3.1 to explain why the modeling work was conducted.
We have added an analogous statement in Line 802.

461: I would argue that you are not investigating the role of catchment geometry, rather catchment flow change which could result from a number of glaciological changes.
We model the glaciers to the top of their current respective catchments, and so adding flux to the upstream boundary necessarily means a larger catchment. We have added this statement in Lines 812 and 877.

Section 3.3 can be much shorter and, like mentioned above, integrated into a combined section with 3.1 and 3.2
This section (now the last two paragraphs of Section 3.1) was very short to begin with, but we have removed some extraneous sentences.

473: are you estimating2-sigma uncertainty? Or calculating it? Not clear what is being done to geologically constrain centerline positions for the model…
We have removed the "2-sigma" designation, as these are really minimum and maximum estimates, and we use the mean for lack of other constraints.

480: "Figures 10-12."
Replaced (now Figures 6–8)

481-486: seems out of place
These lines (Lines 918–923) are necessary, as they describe how ice-marginal samples are treated with respect to glacier centerline elevation for Diamond Hill. The previous paragraph does this for the Hatherton Glacier sites.

For the transient experiments, I would argue that experiments 2 and 3 are distracting and not useful in the scheme of the work presented here. It is interesting to note in the discussion that these small glacial systems show no geological evidence for rapid thinning. Experiment 3 could be retained if authors feel that can be justified given lack of constraints and seemingly arbitrarily chosen LGM ice thickness. I am, however,a proponent of the presentation of negative results, but feel like these extra steps in the modeling are not supported or justified by the geological observations and thus do thestudy a disservice by distracting readers with unnecessary negative findings that do the opposite of what the modeling work is supposed to do - which I interpret as aiming to help explain the geological observations.
See our reply at the beginning of this document for why we need to keep experiments 2 and 3. We disagree with the assertion that modeling work should be limited to explaining geological observations; however, this work does just that. Experiment 2 does this by asking the question:

Could a rapid thinning episode (like that recorded at other TAM outlets) be consistent with our geologic constraints? Experiment 3 does this by determining whether the high-elevation bedrock $^{14}$C ages at Diamond Hill reflect changes of Darwin Glacier, or of local snow and ice fields.

Ice sheet model ensemble I am largely on board with this work, but think the text could be much more concise (apologies for repeating that point).
We have removed repetitive text and combined several paragraphs to make this more concise. However, this is an important section of the paper, and the other reviewer requested more information on this work, so it has not been appreciably shortened.

Discussion is fine; however, broader significance for other small glacial systems in theTAM and beyond could be integrated here.
It is not clear to us that Darwin and Hatherton glaciers are especially representative of small glacier systems in the TAM. However, the possibility that larger neighbors might control the catchment size and downstream ice dynamics of small outlet glaciers could be useful as a warning to those planning fieldwork on other glaciers of similar size. We have included a discussion in the paragraph starting at Line 1356.

Conclusion would benefit greatly from not being written as a list. I suggest that authors have a go at linking the findings and major outcomes between the geochronology and the modeling work. As currently written, the reader has to do a lot of work to piecethe listed bits together and come away from this text with take-aways that will invoke further thought, questions, and applications.
We have re-written the conclusions in paragraph form. However, we believe that the conclusions already link our chronology and modeling work. The list starts with what we can glean from the chronology alone (first two bullet points in previous version); states the inconsistency in the data that necessitates the modeling (third bullet point in previous version); describes how the flowband modeling resolves this inconsistency (fourth bullet point in previous version); describes the use of the ice sheet model ensemble to attempt to put the Darwin-Hatherton results in a regional context, and our finding that rates of grounding-line retreat and glacier thinning might not be easy to relate to one another (fifth point in previous version); and finally poses a hypothesis for why our combined geochronology and model results are different from other TAM glaciers.

Figures The figures look really nice. Figure 1: the bathymetry colormap you have used is not categorized, like the legend suggests.
This has been changed

Include Bidra Valley
Added

Suggest combining figures 2 and 3, figures 4 and 5, and figures 6 and 7.
We have combined these (as well as figures 8 and 9 for consistency). We are slightly concerned that this will lead to awkward formatting, since the maps need to be larger than the corresponding age-elevation plots, but we will defer to the editor on this.

Make sure consistent reference to figure panels, e.g. figure 3 caption that uses "(a)" and "panel c"
Text changed to be consistent.

Supplementary documents Units for ages in the data table are needed.
Units added

**Response to Dr. Richard Selwyn Jones (reviewer 2)**
The paper presents an interesting combination of geochronological data (10Be, 26Aland 14C ages), flowband modelling and 3D ice sheet modelling, focused on Darwin andHatherton Glaciers. The authors conclude that Darwin Glacier thinned gradually by 500m during the Holocene, in contrast to other glaciers in the Transantarctic Mountains,and that this behaviour was possibly due to the convergent ice flow and lateral drag in the vicinity of Byrd Glacier.

A lot of work is covered in this paper. The manuscript is generally well structured,written and illustrated, the methods seem sound, and the conclusions are mostly sup-ported by the results. The study provides new high-quality data for this region, a new understanding about the ice dynamics at Darwin and Hatherton Glaciers, and ideas related to deglaciation of the Ross Sea.

I recommend publication in The Cryosphere following some revision.

There are a few aspects where I think the paper could be improved. First, the rationale for the study needs to be better communicated, particularly in the Abstract andalso Introduction. Why should the reader care?
Thank you for the feedback, and we agree. See response to reviewer 1 on this same issue. We have added this to the abstract and Section 1.1.

Second, it is not always clear what is new and what is "borrowed" from previous studies. This concerns the geochronological constraints and interpretations at Lake Wellman, and the ice sheet model ensemble.
We do state that the Lake Wellman chronology is from King et al. (2020), and that the ice sheet model ensemble is our own work that builds on the Pollard et al. (2016) ensemble. However, we have made this much more explicit.

Third, the maximum ice limits at 9.5 kyr BP are not clearly supported by this study. There are no saturated in situ 14C samples from Magnis Valley and Bibra/Dubris Valleys, and the interpretation of Lake Wellman is based on another study, with limited description of why older samples are ignored.

We disagree here, but we take this comment to mean that our description of our glacial geologic mapping is lacking, and we have added explicit statements of this to Section 2.1 to make it clear how we determined maximum ice thickness independent of the exposure ages. We have also added Figures S1–S11 in the supplemental material to illustrate the glacial geology.

Briefly, we date well-defined depositional limits in the valleys alongside Hatherton Glacier. It is clear that there was no significant ice advance beyond these limits during the last glaciation because of the sharply delimited boundaries between deposits. Therefore, *in situ* $^{14}$C are not needed for the Hatherton Glacier sites, where we can rely on glacial geology alone to demarcate the boundary of the last glacial advance into these valleys. At Diamond Hill, there is no clearly defined limit of deposition, and so we rely on *in situ* $^{14}$C ages of bedrock and our flowband model to interpret the ice thickness history here.

Fourth, the flowband modelling skips over some limitations in this approach, while the ice sheet model ensemble is short on detail. More details on these points are included below.

We have added a discussion of the limitations of the flowband model to the Discussion (Lines 1402–1427), and more details for the ice sheet model ensemble (first two paragraphs of Section 4). We could also have added the discussion of flowband model limitations to Section 3.1, but the other reviewer asked us to make that section shorter. See our responses to the more detailed comments below. Thank you for these insightful and helpful comments.

Detailed comments
Title: 'grounding-line retreat' is included, yet the paper concludes that it is not possible to gauge grounding-line retreat from the thinning data, and the modelling does not include grounding-line retreat of this system. Consider removing.

This is a good point. However, Darwin and Hatherton glaciers are important precisely because they have been used for several decades as key constraints on grounding-line retreat in the Ross Sea. Our work shows that the ice thickness history at the mouth of Darwin Glacier is not easily interpretable in terms of rates of grounding-line retreat, but we also conclude that interpreting the timing of grounding-line arrival from the end of the thinning history is still valid. So, while we agree that the title can be clarified, we would like to retain a mention of the grounding line in the title. We have changed the title to: Holocene thinning of Darwin and Hatherton glaciers, Antarctica and implications for grounding-line retreat in the Ross Sea

Line 17, and elsewhere: The Early and Mid/Middle Holocene are now formalised sub-epochs, so should be capitalised.
Changed

Line 42, and elsewhere: Grounded ice in the Ross Sea is not a true self-contained ice sheet. Rather it is a sector of the Antarctic ice sheet, or a region comprising ice from the East and West Antarctic ice sheets. Consider rephrasing.
 Changed to "grounded ice in the Ross Sea" here and elsewhere

Line 62: Dramatic thinning is also now recorded at Mawson Glacier (Jones et al.,Geology, 2020) and David Glacier (Stutz et al., TCD, in review).
References added

Lines 68-69: Can you describe Darwin and Hatherton Glaciers in relation to the glaciers mentioned above?
We have added a brief statement in Lines 137–139

Lines 99-104: It might also be worth mentioning the model buttressing studies of Furstet al. (2016) and Reese et al. (2018), which show that this region has a large buttress-ing effect on the ice sheet (albeit based on the current ice configuration).
Added

Lines 156-163: Slightly confusing timeline of research. It would probably be better to present in publishing order (Storey et al., 2010, Joy et al., 2014, King et al., 2020).
The research timeline has been re-ordered as much as possible. However, King et al. (2020) and Storey et al. (2010) both worked at Lake Wellman, while Joy et al. (2014) worked at Dubris and Bibra Valleys; likewise, Bockheim et al. (1989) and King et al. (2020) primarily used radiocarbon dating, while Story et al. (2010) and Joy et al. (2014) primarily used exposure dating. So there is some need to use a location-based or method-based rather than publication year-based timeline.

Lines 167-169: This last point seems a bit lost. Consider making it a third goal.
Added (iii)

Line 172 ("erratics"): Technically, if they are erratics then their lithology should differto the local geology; how they are 'erratic' should be described here. It would also be worth providing a broad description of the erratic size - cobbles, boulders?
Changed to glacially-transported cobbles and boulders, with an explanation of just how erratic they are.

Line 172 ("Diamond Hill, as well as from Bibra, Dubris and Magnis Valleys"): It would be useful to state here how far each of these sites are from the modern grounding line.
Distances added (Line 353)

Line 173: "freshest" should be made clearer. Rocks that appeared least weathered?
Changed to "least weathered"

Lines 174-180: As previous studies in the area (e.g. Storey et al., 2010; Joy et al.,2014) have highlighted the importance of sampling strategy, particularly regarding re-working by cold-based ice, it would be useful to point out how your approach is simi-lar/different.
We have enhanced our description of the glacial geologic mapping and sampling strategies in section 2.1, and described how this differs from or is similar to previous work in Lines 370–375.

Lines 193-195: It would be useful here to clarify what is detectable given uncertainties in erosion rates and measurements.
Lines 380: "Within the range of subaerial erosion rates typical of Antarctic bedrock (<1 m/Myr), $^{14}$C concentrations reach secular equilibrium in ~30 kyr (Balco et al., 2016). Because of the short half-life of $^{14}$C, 1–3 kyr of ice cover during the LGM will create a detectable signal of burial, depending on production rate and measurement sensitivity, while rock that was not covered by ice in the last 30 kyr will remain at its equilibrium concentration."

Lines 218-219: It would be useful to quantify the impact of the scaling scheme on the exposure ages - what percent difference between schemes (e.g. Stone (2000) and Lifton et al. (2014) nuclide-specific scheme)? Also, what production rate dataset is used? This should be described here.
Reference to Borchers et al. (2016) production rate dataset added Line 415. We have quantified the difference between the most frequently used schemes in Lines 417–420. Lifton et al. (2014) scheme makes $^{26}$Al and $^{10}$Be ages 7–8% younger than presented here using the Stone (2000) scheme, while $^{14}$C ages would increase by $0.5 \pm 1.8\%$. These differences do not change our interpretations, and King et al. (2020) found a good agreement between exposure ages calculated using the Stone (2000) scaling scheme and their algae radiocarbon chronology.

Lines 250-255: It is not clear how the maximum thickness is determined to be from at least 14 kyr. No potentially older samples were collected from higher elevations, and there are no in situ 14C measurements from here to test for saturation.
This has been clarified by text added to Section 2.1 and the glacial geology photos in the Supplement (Figures S1–S11). Briefly, we sampled at the limit of the Britannia I deposit, which is very well defined. The deposit beyond this limit (Britannia II) was shown by Joy et al. (2014) to be much older (~130 kyr). In situ bedrock $^{14}$C measurements are not necessary in locations like these, where clearly defined limits of deposits demarcate positions of maximum advance.

Lines 275-278: If no new ages from Lake Wellman are presented in this study, then this needs to be stated clearly here, indicating that data from previous studies are used to assist interpretation of ice history in the area. You also need to explain in more detail why the higher elevation erratic ages at 20 kyr are "pre-exposed" and therefore ignored.

We do state this in the figure captions, but we have reiterated in the main text (Lines 511–512). In Lines 539–544 we have explained the reason for interpreting the 20 kyr ages as reflecting prior exposure to cosmic rays. Briefly stated, the Lake Wellman area has a great abundance of pre-exposed erratics (see our Fig 4, King et al. [2020] and the widely scattered ages in Story et al. [2010]). The 20 kyr ages are inconsistent with the dense algae radiocarbon dataset and the ages of depositional limits elsewhere at Hatherton Glacier, and are therefore considered outliers.

Line 394: The text in this section does not convincingly support the claim that ice wasat its maximum at 9.5 kyr. There are no saturated in situ 14C samples from these sites,and there is limited justification of why older samples are ignored. Clarify and expand on this point in the above text, or revise this claim.

See point above with regards to lines 250–255 in the previous version and to the reviewer's third point in the summary. These ages are from well-defined depositional limits in many cases, or are the highest-elevation samples that exhibit the characteristics of the Britannia I deposit. The contrast between the Britannia I and Britannia II deposits is stark, as the Britannia II deposit is ~115 kyr older than Britannia I.

Line 396: You should probably include a study that directly dates grounding-line retreat in the Ross Embayment (e.g. McKay et al., 2016; Prothro et al., 2020).

These references have been added (Line 753).

Line 414: Jones et al. (2016) is not included in the references.

Sorry about that. This reference has been added.

Lines 416-417: The model flowband/domain needs to be shown somewhere, perhaps in Figure 1c.

Flowband model domain has been added to the supplemental material (Fig S14) to avoid cluttering Fig. 1.

Lines 420-423: While this point is largely true, the flowband is a very simple model and so the limitations should be stated. What key stresses are not calculated? What aspects of the deglacial history can versus cannot be tested?

We have added this information to Lines 1401–1426 in the Discussion. It could have been added to Section 3.1, but the other reviewer asked us to trim down that section.

Lines 423-430: As far as I can see, this model does not account for lateral drag, which the conclusions indicate is potentially very important. How might this impact the flow-band results?

The lateral drag that we indicate could be important is far outside the flowband model domain at Minna Bluff and Cape Crozier. But, we appreciate the question because the lower portions of many TAM outlets become more laterally constrained (as you pointed out in Jones et al., 2016 and was important for Skelton Glacier). The importance of lateral drag depends on the cross-sectional geometry of a glacier (how much friction from valley walls) and can be significant if the glacier is narrow, if the flow is channelized, or the ice-flow speeds are high. While we seek to apply the simplest flowline model that is physically meaningful, there has been work to develop parameterizations for higher-order physics that are otherwise ignored in shallow-ice flowline models like we apply here. In the case of lateral drag, Adhikari and Marshall (2012) expanded on previous work to provide a set of correction factors for lateral drag that is due to either glacier geometry or due to variations in glacier slip. So, while we could apply this parameterization as a correction factor (0.1 to 1) to modify our value of gravitational driving stress, this work showed that lateral drag was most critical for fast flowing ice and if the glacier was narrow or channelized. Given this, and how we apply the flowline model in this work there are two main reasons we chose to not include a lateral-drag factor: 1) We treat the two parameter values already used in our model (the deformation factor and the sliding factor) as tuning parameters. We do not interpret the value of these factors, and so adding an additional factor will only offset the inferred (tuned) value and not offer more physical information. In addition, since we have no constraints on how any of these factors may have changed in time we use the values inferred from matching to modern conditions. 2) Most of the ice in our domain is relatively slow flowing. It is possible that the lower portion of Darwin is flowing fast enough for lateral drag to be more significant, but that would be taken up in our inferred values for the deformation factor. Again, since these values are tuning parameters any actual influence of lateral drag on the modern flow field would be reflected in the inferred values of the tuning parameters.

Line 428: How many grid points? What is the horizontal resolution used?

Added to Line 809: 1km resolution and number of grid points (75 for Hatherton, 141 for Darwin)

Line 438: How is SMB scaled using the scaling factors for LGM scenarios?

We have added the clarification that scaling is spatially uniform (Line 853).

Lines 452-453: To what extent does modification of the downstream boundary represent perturbations (e.g. reduce buttressing, grounding-line retreat) downstream of this point? As mentioned above, you should make clear what processes are and are not accounted for, and what can be tested.

This information has been added in the Discussion section in Lines 1401–1426. The downstream boundary is an elevation value that represents how that point on the lower glacier changed over time in response to ice-shelf buttressing and glacier evolution as the grounding line retreated.

However, we are not able to interpret the distribution of processes (i.e., change in ice-shelf buttressing vs change in grounding-line position) that led to our inferred elevation history. Our model domain always begins upstream of the grounding line and the deformation and sliding factors in our model do not change over time since we have no constraints on how they would evolve.

Line 468, and elsewhere: What is meant by 'LGM' in the context of these sites? The proposed local ice maximum? The Antarctic or global glacial maximum?
We mean the local ice maximum, which is much later than the global or even Antarctic maxima. The usage is frequent enough that it would be ungainly to specify local maximum everywhere in the text, but have specified this in footnote to Line 75.

Lines 597-599: This interpretation should be in Section 2.
We have added this to Section 2.3 (Summary of geochronologic constraints)

Lines 646-648: Is this a new set of model simulations, and not output from previously published simulations? This would be a considerable effort and requires more descrip-tion here. No need to describe all the of the model physics, but state that this can be found in a different paper. What ocean and atmosphere forcings are applied? What are the values of other parameters that were not examined? Some readers would expect to see these details, even if described in the Supplementary Material.
This is our original work, presented here for the first time. We have added more description and references to the first two paragraphs of Section 4.

Lines 712-714: How might the model resolution impact this finding?
While very high resolution (≤1 km; unfeasible for these simulations) is required to very accurately model grounding-line retreat, the response of grounded ice to grounding-line retreat is less resolution dependent. So, the 10 km resolution probably leads to inaccuracies in the grounding-line migration, but the response of grounded ice should be reasonably well captured. We have added this statement to Line 1225.

Line 722: McMurdo Sound deglaciated in the Mid Holocene, so "Early to Mid Holocene"is probably more correct.
Amended

Lines 728-279: I agree that more data are required from the lower reaches of SkeltonGlacier. But, to be picky, the erratic age at 6 kyr is <20 km from the modern grounding line, which is not "far" considering the scale of these glaciers. Consider rephrasing.
Good point. Rephrased to "A small number of samples ~16–50 km from the modern grounding line"

Line 735: In what context is there no record of "an exceptionally fast period of thin-ning"? How is this quantified? Is this relative to modern observations or other thinning estimates from TAM?
We mean this in contrast to, for example, the 200 m of thinning in <2000 years at Mackay Glacier or the similar rapid thinning at Beardmore. We don't define a threshold for what counts as "exceptionally fast", but by contrast to records from some other glaciers these records show remarkably steady thinning rates. We have clarified this in Line 1431.

Lines 748-749: As most thinning at Mackay Glacier to the north occurred at 7.5-6.5kyr, McMurdo Sound became free of ground ice between 6.5 (not 7.5) and 9 kyr BP.
Amended

Line 755: Also the case with Mawson Glacier and David Glacier (see references above)
Added

Lines 773-777: The argument regarding glacial isostatic adjustment could be clearer.How greater isostatic depression at Minna Bluff could resolve the elevation differences between Darwin Glacier and Minna Bluff needs a little more explanation. Is this just considering the relative lowering of the LGM elevations at Minna Bluff, or also the im-pact on the age constraints? It is possible to estimate the relative elevation difference?
This is just accounting for relative lowering between the two sites, to attempt to make the ice surface at the mouth of Darwin higher than at Minna Bluff. This would need to be the case regardless of age constraints in order to fit these previous reconstructions. We have added estimates based on the ICE6G model (Argus et al., 2014; Peltier et al., 2014), which actually disagrees with our argument, so we have noted that this is not our preferred explanation. However, we also note that the 100% areal ice cover assumed in ICE6G at this location is incorrect, and so isostatic depression at Diamond Hill is likely overestimated in that model. This is found on Line 1478.

Lines 796-797: This was done for the Ross Embayment alone in Lowry et al. (2019,2020), yet still some large differences in the timing of thinning between simulations and geological data.
Lowry et al. (2019) state that they tune their regional domain to the present day ice thickness and velocity observations. There is no mention of an optimization using geochronologic constraints, although their results are compared to exposure ages. We have amended the text to read: "Tuning parameters to fit geochronologic constraints using a large ensemble over the Ross Embayment alone could lead to a better fit between models and data." (Line 1510)

Lines 805-809: While this is a useful discussion with the suggestion that the confluence of Byrd-Hatherton-Darwin Glaciers may explain gradual thinning at Darwin despite the topography at Discovery Deep, it is a bit of a push to hint that the results from this study support the

hypothesis that Byrd Glacier is of fundamental importance to the stability of ice in the Ross Sea sector.

Thank you, and we agree. This statement has been removed.

Line 818-819: As mentioned previously, the reported evidence does not convincingly indicate that ice could not have been thicker at the LGM. This conclusion may not be wrong, but it is currently not well-supported.

We disagree, for the reasons explained above. To reiterate briefly, we dated well-defined depositional limits along Hatherton Glacier. There is no reason to suspect that ice advanced significantly beyond these limits, as there is no mechanism to explain a sudden shift from no deposition to abundant deposition.

Figure 1: Label the Ross Sea, McMurdo Sounds and Mackay Glacier, which are men-tioned in the text.

Labels added

Figures 3, 5, 7: It would be worth stating that these points represent the mean ageswith total uncertainty at 1 standard deviation. For clarity, perhaps use a different colour for the horizontal blue line; some readers may automatically associate this line with theBritannia I limit in the maps. It is difficult to understand how these ages relate to the samples and deposit limits in the maps. It would help if you showed the elevation of the Britannia I limit in these plots.

Horizontal blue line color changed. Britannia I limit elevations added.

Figure 9: The 14C-saturated sample could probably be better represented in the figure. Is it trying to say that the apparent age is from 11.5 kyr to infinity?

Good point. This sample is $^{14}$C saturated, but uncertainties in analysis, production rate, and scaling factor admit the possibility of a much younger age of ~12 kyr. We have added an arrow to the right in this panel, with explanation in the caption, and added all ages to panel d, which shows ages on a log-scale. This is now Figure 5.

Figure 13: Can the figure be annotated to highlight the main interpretations from these results?

We have added several text annotations. This is now Figure 9.

Supplemental Data Table: Units are not included for all columns, on all sheets. The scaling scheme and production rates used for the ages should be clearly stated.

This information has been added.

**Response to Dr. Chris Stokes  (editor)**
Thank you for your comment regarding the structure of the manuscript. We do understand that this is a somewhat unorthodox structure, but we feel an understanding of the geochronologic

results and the problems they present are essential to understanding why we need to use the two modeling approaches. For example, it would be awkward to try to explain our flowband modeling approach in a single methods section, before the reader understands that the exposure ages from near the grounding line suggest multiple conflicting histories. So, we have opted to keep the original structure. We have tried to make the manuscript structure much more clear to the reader by adding subheadings to Section 4 and giving a brief outline of the paper at the end of Section 1.

References

**Adhikari S and Marshall SJ** (2012) Parameterization of lateral drag in flowline models of glacier dynamics. *Journal of Glaciology* **58**(212), 1119–1132. doi:10.3189/2012JoG12J018.

**Anderson BM, Hindmarsh RCA and Lawson WJ** (2004) A modelling study of the response of Hatherton Glacier to Ross Ice Sheet grounding line retreat. *Global and Planetary Change* **42**(1–4), 143–153. doi:10.1016/j.gloplacha.2003.11.006.

**Bockheim JG, Wilson SC, Denton GH, Andersen BG and Stuiver M** (1989) Late Quaternary ice-surface fluctuations of Hatherton Glacier, Transantarctic Mountains. *Quaternary Research* **31**(2), 229–254.

**Borchers B and others** (2016) Geological calibration of spallation production rates in the CRONUS-Earth project. *Quaternary Geochronology* **31**, 188–198. doi:10.1016/j.quageo.2015.01.009.

**Jones RS, Golledge NR, Mackintosh AN and Norton KP** (2016) Past and present dynamics of Skelton Glacier, Transantarctic Mountains. *Antarctic Science* **28**(05), 371–386. doi:10.1017/S0954102016000195.

**Joy K, Fink D, Storey B and Atkins C** (2014) A 2 million year glacial chronology of the Hatherton Glacier, Antarctica and implications for the size of the East Antarctic Ice Sheet at the Last Glacial Maximum. *Quaternary Science Reviews* **83**, 46–57. doi:10.1016/j.quascirev.2013.10.028.

**Lifton, N., Sato, T., and Dunai, T. J.** (2014): Scaling in situ cosmogenic nuclide production rates using analytical approximations to atmospheric cosmic-ray fluxes, Earth and Planetary Science Letters 386, 149–160, https://doi.org/10.1016/j.epsl.2013.10.052.

**King C, Hall B, Hillebrand T and Stone J** (2020) Delayed maximum and recession of an East Antarctic outlet glacier. *Geology* **48**(6), 5.

**Pollard D, Chang W, Haran M, Applegate P and DeConto R** (2016) Large ensemble modeling of the last deglacial retreat of the West Antarctic Ice Sheet: comparison of

simple and advanced statistical techniques. *Geosci. Model Dev.* **9**(5), 1697–1723. doi:10.5194/gmd-9-1697-2016.

**Storey BC and others** (2010) Cosmogenic nuclide exposure age constraints on the glacial history of the Lake Wellman area, Darwin Mountains, Antarctica. *Antarctic Science* **22**(06), 603–618. doi:10.1017/S0954102010000799.